# Framing COVID-19: How we conceptualize and discuss the pandemic on Twitter

**Philipp Wicke** [1]*, **Marianna M. Bolognesi** [2]

**1** Department of Computer Science, University College Dublin, Dublin, Ireland, **2** Department of Modern Languages, Literatures, and Cultures, University Bologna, Bologna, Italy

* philipp.wicke@ucdconnect.ie

## Abstract

Doctors and nurses in these weeks and months are busy in the trenches, fighting against a new invisible enemy: Covid-19. Cities are locked down and civilians are besieged in their own homes, to prevent the spreading of the virus. War-related terminology is commonly used to frame the discourse around epidemics and diseases. The discourse around the current epidemic makes use of war-related metaphors too, not only in public discourse and in the media, but also in the tweets written by non-experts of mass communication. We hereby present an analysis of the discourse around #Covid-19, based on a large corpus tweets posted on Twitter during March and April 2020. Using topic modelling we first analyze the topics around which the discourse can be classified. Then, we show that the WAR framing is used to talk about specific topics, such as the virus treatment, but not others, such as the effects of social distancing on the population. We then measure and compare the popularity of the WAR frame to three alternative figurative frames (MONSTER, STORM and TSU-NAMI) and a literal frame used as control (FAMILY). The results show that while the FAMILY frame covers a wider portion of the corpus, among the figurative frames WAR, a highly conventional one, is the frame used most frequently. Yet, this frame does not seem to be apt to elaborate the discourse around some aspects involved in the current situation. Therefore, we conclude, in line with previous suggestions, a plethora of framing options—or a meta-phor menu—may facilitate the communication of various aspects involved in the Covid-19-related discourse on the social media, and thus support civilians in the expression of their feelings, opinions and beliefs during the current pandemic.

## Introduction

On December 31, 2019, Chinese authorities alerted the World Health Organization of pneumonia cases in Wuhan City, within the Hubei province in China. The cause, they initially said, was unknown, and the disease was first referred to as 2019-nCoV and then named COVID-19. The next day, the Huanan seafood market was closed, because it was suspected to be the source of the unknown disease, as some of the patients presenting with the pneumonia-like illness were dealers or vendors at that market. Since then, the disease has spread quickly throughout China, and from there to the rest of the world. SARS-CoV-2 is the name of the virus

(UCD) provides the funding for the publication fees.

**Competing interests:** The authors have declared that no competing interests exist.

responsible for this coronavirus pandemic that we are experiencing while the present article was in writing. The virus has so far spread throughout all the inhabited continents and affected millions of people, killing thousands of individuals. Schools have been shut down, kids are still at home in many countries, many citizens are now working remotely, locked down in their houses, and leaving only for reasons of primary necessity, such as shopping for groceries and going to medical appointments.

With many countries implementing lock downs and promoting quarantines, suggesting or forcing citizens to stay inside their homes in order to avoid spreading the virus, millions of people are experiencing a global pandemic for the first time in their lives. The social distancing enforced by various governments stimulated many internet users to use social media to communicate and express their own concerns, opinions, beliefs and feelings in relation to this new reality. On Twitter, tweets with hashtags such as #coronavirus, #Covid-19 or #Covid pile up quickly (for instance, we accumulated around 16,000 tweets per hour). A variety of issues are debated on a daily basis on Twitter, in relation to the pandemic. These include, but are not limited to, the political and social consequences of various governmental decisions, the situations in the hospitals getting increasingly more crowded every day, the interpretation of the numbers associated with the spreading of the pandemic, the problems that families face with homeschooling their children while working from home, and so forth. Among these issues, the discussion around the treatment and containment of the virus is surely a central topic.

The present article aims at describing how the discourse around Covid-19 is framed on Twitter. In particular, we present a study that elucidates what the main topics are related to the discourse around Covid-19 on Twitter and to what extent the treatment of the disease is framed figuratively. Because previous research has shown that various social and political issues addressed in public discourse are framed in terms of wars [1], we assumed that this tendency may emerge also on Twitter, in relation to the discourse around Covid-19. Although Twitter contains messages written by journalists and other experts in mass communication, most tweets are provided by non-expert communicators. We investigated to what extent Twitter users, and therefore non-expert communicators, frame Covid-19 in terms of a war, and whether other figurative framings arise from automated analyses of our corpus of Tweets containing virus-related hashtags.

In particular, we addressed the following research questions:

1. What type of topics are discussed on Twitter, in relation to Covid-19?

2. To what extent is the WAR figurative frame and the conventional metaphor DISEASE TREATMENT IS WAR used to talk about Covid-19 on Twitter? Which lexical units are used within this metaphorical frame and which lexical units are not?

3. Are there alternative figurative frames used to talk about Covid-19 on Twitter? And how does their use compare to the use of the WAR frame?

These three questions are addressed in the remainder of this paper in this same order. For each question we present methods, results and discussion of specific corpus-based analyses.

The innovative aspect of this paper lies in the quantitative nature of our observations and analyses of figurative framings used in pandemic-related discourse, and in the methods used: topic modelling. In particular, the WAR frame, which previous studies have identified as pervasive in many crisis-related texts, is hereby investigated by means of automated methods (topic modelling) applied to real-world data. This is a new approach in cognitive linguistics and metaphor studies, where the analysis of figurative frames is typically based on qualitative observations or small-scale corpus analyses. By answering the research questions outlined above our study opens the path to further investigations that may take a longitudinal

perspective on the current reality, to investigate how the discourse around Covid-19 changes, with the development of new phases of the pandemic. These present results and the future developments provide important information in the field of opinion mining and can be used to understand the current state of mind, beliefs and feelings of various communities.

## Theoretical background

Mining the information encoded by private internet users in the short texts posted on Twitter (the tweets) is becoming an increasingly fruitful field of research. In relation to health discourse, tweets have been used by epidemiologists to access supplementary data about epidemics. For example, tweets about particular diseases have been compared to gold-standard incidence data, showing that there are positive correlations between the number of tweets discussing flu-symptoms and official statistics about the virus spread such as those published by the Centers for Disease Control and Prevention and the Health Protection Agency [2]. Already a decade ago, in Brazil, tweets have been used to track the spreading of the dengue fever, a mosquito-transmitted virus [3]. More recently, Pruss and colleagues [4] used a topic model applied to a large corpus of tweets to automatically identify and extract the key topics of discussion about the Zika disease, a virus that spread mainly in the Americas in early 2015. The authors also found that rises in tweeting activity tended to follow major events related to the disease, and Zika-related discussions were moderately correlated with the virus incidence. Moreover, it has been demonstrated that the combination of data collected from hospitals about specific diseases, and data collected from social media, can improve surveillance and forecasting about the disease more effectively [5, 6].

Besides providing a valuable tool for tracking the spread of epidemics, and thus helping experts to make more effective decisions, social media have been used to investigate public awareness, attitudes and reactions about specific diseases [7, 8]. As Pruss and colleagues report in their review [4], the 2013 measles outbreak in the Netherlands, for example, has been analyzed in this perspective by Mollema and colleagues [9], who compared the number of tweets (and other messages posted on social media) with the number of online news articles as well as with the number of reported measles cases and found a strong correlation between social media messages and news articles and a mild correlation between number of tweets and number of reported measles cases. Moreover, through a topic analysis and a sentiment analysis of the tweets, they found that the most common opinion expressed in the tweets was frustration regarding people who do not vaccinate because of religious reasons (the measles outbreak in the Netherlands began among Orthodox Protestants who often refuse vaccination for religious reasons).

The 2014 Ebola outbreak in Africa was also used as a case study to mine the attitudes, concerns and opinions of the public, expressed on Twitter. For example, Lazard and colleagues [10] analyzed user-generated tweets to understand what were the main topics that concerned the American public, when widespread panic ensued on US soil after one case of Ebola was detected. The authors found that the main topics of concern for the American public were the symptoms and lifespan of the virus, the disease transfer and contraction, whether it was safe to travel, and how they could protect their body from the disease. In relation to the same outbreak, Tran and Lee [11] built Ebola-related information propagation models to mine the Ebola related tweets and the information encoded therein, focusing on the distribution over six topics, broadly defined as: 1. Ebola cases in the US, 2. Ebola outbreak in the world, 3. fear and prayer, 4. Ebola spread and warning, 5. jokes, swearing and disapproval of jokes and 6. impact of Ebola on daily life. The authors found that the second topic had the lowest focus, while the fifth and sixth had the highest.

More recently, tweets have been mined to understand the discussion around the Zika epidemics. Miller and colleagues [12] used a combination of natural language processing and machine learning techniques to determine the distribution of topics in relation to four characteristics of Zika: symptoms, transmission, prevention, and treatment. The authors managed to outline the most persistent concerns or misconceptions regarding the Zika virus, and provided a complex map of topics emerged from the tweets posted within each of the four categories. For example, in relation to the issue of prevention they observed the emergence of the following topics: need for control and prevention of spread, need for money, ways to prevent spread, bill to get funds, and research. Vijaykumar and colleagues [13] analyzed how content related to Zika disease spreads on Twitter, thanks to tweet amplifiers and retweets. The authors found that, of the 12 themes taken into account, Zika transmission was the most frequently talked about on Twitter. Finally, Pruss and colleagues [4] mined a corpus of tweets in three different languages (Spanish, Portuguese and English) with a multilingual topic model and identified key topics of discussion across the languages. The authors reported that the Zika outbreak was discussed differently around the world, and the topics identified were distributed in different ways across the three languages.

In cognitive linguistics, and in particular in metaphor studies, public discourse is often analyzed in relation to different figurative and literal communicative frames. We "frame" a topic when we "select some aspects of a perceived reality and make them more salient in a communicating text, in such a way as to promote a particular problem definition, causal interpretation, moral evaluation, and/or treatment recommendation for the item described" [14, p.53]. Metaphors are often used to talk about different aspects of diseases, such as their treatment, their outbreak and their symptoms. The framing power of metaphor is particularly relevant in health-related discourse, because it has been shown that it can impact patients' general well-being. For example, in a seminal study Sontag [15] criticized the popular use of war metaphors to talk about cancer, a topic of research recently investigated also by Semino and colleagues [16]. As these authors explain, the military metaphor that we tend to use to talk about the development, spreading and cure of cancer inside the human body has been repeatedly rejected by cancer patients as well as by many relatives and doctors, who indicate that such framing provokes anxiety and a sense of helplessness that can have negative implications for cancer patients. In a series of experiments, for example, Hendricks and colleagues [17] found that framing a person's cancer situation within the war metaphor, and therefore as a battle, has the consequence of making people believe that the patient may feel guilty in the case that the treatment does not succeed. Conversely, framing the cancer situation as a journey encourages the inference that the patient will experience less anxiety about her health condition.

The military metaphor commonly used to talk about diseases such as cancer is a very common one to be found in public discourse [1]. According to Karlberg and Buell [18] 17% of all articles in Time Magazine published between 1981 and 2000, contained at least one war metaphor. The war metaphor is not used solely to frame the discourse around diseases, but also the discussion around political campaigns, crime, drugs and poverty. As explained in [1], war metaphors are pervasive in public discourse and span a wide range of topics because they provide a very effective structural framework for communicating and thinking about abstract and complex topics. Moreover, this frame is characterized by a strong negative emotional valence. In the special case of the diseases, the war metaphor is typically used to frame the situation relatively to the treatment of the disease. As indicated in MetaNet, the Berkeley-based structured repository of conceptual metaphors and frames [19], the metaphor can be formalized as DISEASE TREATMENT IS WAR, or TREATING DISEASE IS WAGING WAR (https://metaphor.icsi.berkeley.edu/pub/en/index.php/Metaphor:DISEASE_TREATMENT_IS_WAR). Within this metaphor, a variety of mappings can be identified, including: the diseased

cells are enemy combatants, medical professionals are the army of allies, the body is the battle-field, medical tools are weapons, and applying a treatment is fighting.

The figurative frame of WAR, used in discourses around diseases, is certainly a conventional one, frequently used, often unconsciously. As argued by [1], such a frame is handy and frequently used because it draws on basic knowledge that everyone has, even though for most people this is not knowledge coming from first-hand experience. Moreover, this frame expressed in an exemplary way the urgency associated with a very negative situation, and the necessity for actions to be taken, in order to achieve a final outcome quickly. The outcome can be either positive or negative, in a rather categorical way. The inner structure of the frame is also relatively simple, with opposing forces clearly labelled as in-groups and out-groups, or allies and enemies. Each force has a strategy to achieve a goal, which involves risks and can potentially be lethal. For these reasons, this frame is arguably very well suited to appear in the discourse around Covid-19, as previously observed in relation to other diseases. The adversarial relationship between doctors and the virus, the different goals afforded by the two forces and the human body as the battlefield for this operation, are possible mappings that we seek to trace down, with our analysis on Covid-19 related tweets.

Despite the undebatable frequency by which public discourse around diseases uses war metaphors, this frame is sometimes not well received, as mentioned above, and war-related metaphors can be opposed for various reasons. In the last weeks an increasingly large amount of blog posts and articles for the large public confronted and opposed the use of military language to talk about the pandemic, providing different arguments that range from the blindness that war metaphors generate toward alternative ways to solve problems, to the rise of xenophobia and the increase of fear and anxiety in the population that these metaphors generate. For example, [20] argued that "to adopt a wartime mentality is fundamentally to allow for an all-bets-are-off, anything-goes approach to emerging victorious. And while there may very well be a time for slapdash tactics in the course of weaponized encounters on the physical battlefield, this is never how one should endeavor to practice medicine." [21] claimed that "using a war narrative to talk about COVID-19 plays into the hands of white supremacist groups. U.S. officials and the media should stop it." [22] explained that using a WAR frame breeds fear and anxiety, divides communities, compromises democracies and may legitimize the use of actual military actions. [23] foreshadowed "shifts towards dangerous authoritarian power-grabs, as in Hungary, where Prime Minister Viktor Orbán seized wide-ranging emergency powers and the ability to rule by decree", as a consequence of war-related language in the current situation.

As we will further elaborate in the Discussion section, in some cases the press opposes deliberately the war frame, advancing alternative figurative frames. Tracking down alternative frames to the war one in a qualitative manner has been a recent endeavor initiated by scholars in cognitive linguistics and corpus analysis on Twitter. The hashtag #ReframeCovid (first proposed, to the best of our knowledge, by two Spanish scholars, Inés Olza and Paula Sobrino) has been recently used to harvest texts such as articles, advertisements and notes showing how the virus has been opposed and framed in alternative ways by a few journalists and writers. Notably, the discourse has been reframed using lexical units related to the domain of FOOTBALL, of GAMES, of STORMS and so forth. In this paper, we explored the structure and functioning of alternative frames too, in a corpus-based analysis of tweets about Covid-19, and compared them to the WAR frame as well as a literal frame, that is the FAMILY frame. In this case, it should be mentioned that although FAMILY may be used as a metaphorical frame to talk, for example, about nations ("founding fathers", "daughters of the American revolution", "sending our sons to war" and so forth, see [24] for an extensive discussion), in the discourse around Covid-19, family-related words are typically used in their literal meaning, to talk for example

about family members affected by the virus and family-dynamics being disrupted by the measures taken in response to the pandemic (e.g., the lock down).

## Study design

To address our three research questions, first we explored the range of topics addressed in the discourse on Covid-19 on Twitter using a topic modelling technique. Consequently, we explored the actual usage of the WAR frame, and explored which topics (among the topics identified in the first part of the study) are more frequently framed within the metaphor of WAR. To do so, we compiled a list of war-related lexical units and ran it against our corpus of Covid-19 tweets, observing and discussing which lexical units of each frame were used in the tweets, and within which topics. Finally, we explored alternative frames that could be used to frame the discourse around Covid-19 on Twitter. To do so, we compiled lists of lexical units for selected alternative frames (three figurative frames and one literal one) and compared the percentages by which they appear to be used in the corpus of tweets, against the percentages by which the WAR frame is used. To conclude, we replicated our analysis on a new corpus of tweets collected in the weeks that followed the collection of the first corpus, following the same criteria, as well as on an existing resource "Coronavirus Tweets Dataset" by Lamsal [25], which became available during the revision process. Lamsal's dataset is a constantly updated repository of twitter IDs. The collection of those IDs is based on English tweets that include 90+ hashtags and keywords that are commonly used while referencing the pandemic.

### Constructing the corpus of Covid-19 tweets

In order to identify tweets that relate to the Covid-19 epidemic, we defined a set of relevant hashtags used to talk about the virus: #covid19, #coronavirus, #ncov2019, #2019ncov, #nCoV, #nCoV2019, #2019nCoV, #COVID19. Using Twitter's official API in combination with the *Tweepy* python library (tweepy.org) for 14 days we collected 25.000 tweets per day that contain at least one of the hashtags and no retweets. The tweets were collected in accordance with the Twitter terms of service. Two main restrictions of those terms and service motivated our decision to limit the extent of our corpus: Firstly, the free streaming API only allows access up to one week of Twitter's history. Secondly, there is a limit of 180 requests per 15-minutes.

To balance our corpus, we needed to consider how a single tweet or a single user weights on the overall corpus. For example, a scientific analysis of fake news spread during the 2016 US presidential election showed that about 1% of users accounted for 80% of fake news and report that other research suggests that 80% of all tweets can be linked to the top 10% of most tweeting users [26]. In other words, there are Twitter users who tweet a lot, and Twitter users who tweet seldomly. This may be problematic when looking for the frequency distribution of word uses. For example, a specific Twitter user who is very fond of sci-fi issues might use the MONSTER framing very frequently in their tweets, also when tweeting about Covid-19. If we kept all the tweets by this user, this might have biased the frequency distribution of the MONSTER related words. For the purpose of our study, we were interested in exploring the relative uses of different frames in the discourse around Covid-19 on Twitter, rather than in the absolute percentages of use of the different frames in Twitter. Therefore, we constructed our corpus to be representative and balanced, as well as manageable from a computational perspective. To do so, we retained only one tweet per user and dropped retweets. Keeping only one tweet per user allowed us to balance compulsive tweeters and less involved Twitter users. Table 1 accumulates for each day how many tweets have been collected and the number of filtered tweets

**Table 1. Dates of collection for tweets containing hashtags related to the Covid-19 epidemic.**

| Date | 20.03.2020 | 21.03.2020 | 22.03.2020 | 23.03.2020 | 24.03.2020 | 25.03.2020 | 26.03.2020 |
|---|---|---|---|---|---|---|---|
| Filtered / Collected | 20,316/25,000 | 39,284/50,000 | 57,073/75,000 | 73,346/100,000 | 89,785/125,000 | 103,614/150,000 | 118,866/175,000 |
| Date | 27.03.2020 | 28.03.2020 | 29.03.2020 | 30.03.2020 | 31.03.2020 | 01.04.2020 | 02.04.2020 |
| Filtered / Collected | 132,995/200,000 | 146,654/225,000 | 156,775/250,000 | 167,847/275,000 | 180,234/300,000 | 191,278/325,000 | 203,756/350,000 |

after discarding tweets from users that have already tweeted. This implies that only the first contribution of each user was retained: we have kept the first tweet on that day by users whose tweets we have not collected yet. Consequently, all 203,756 tweets are from unique tweeters. For example, Table 1 data column 3 shows that on 22.03.2020, we collected 75,000 tweets in total and from those 57,073 were from unique tweeters.

The filtered tweets are single tweets per user, the total number of collected tweets is 25k per day.

As the streaming API starts collecting tweets from 23:59 CET of each day and has been limited to English, our corpus encompasses mainly tweets produced by users residing in the USA, where the time of data collection corresponds to awake hours, and the targeted language corresponds to the first language of most US residents, according to the American Community Survey (ACS). The total number of collected tweets from unique tweeters over 14 days (20.03.2020 to 02.04.2020) is 203,756. This resulted in 41.78% of the collected tweets being tweets of a user tweeting more than once, thus being filtered out.

Given our research questions and our aims, we did not analyze the dynamics involved in retweetings and mentions on Twitter and neither did we provide an analysis of usernames, hashtags or URLs.

In compliance with the privacy rights of Twitter [27], we have only collected the tweet along with a timestamp. In order to comply with Twitter's content redistribution policy, it is not possible to make any information other than the Tweet IDs publicly available. We have therefore stored our data as tweet IDs. This dataset is publicly available in the online repository on OSF, retrievable at the following url: https://osf.io/bj5a6/?view_only=1644595a66dd4adebeeb6b2bb0449c89.

## General corpus analytics

The corpus encompassed 203,756 tweets, in which the 30 most common words, excluding stopwords and online tags (e.g. "&", "https") are:

> people (19153), us (13368), get (11270), like (10451), time (10263), help (10091), need (9993), cases (9205), home (9044), stay (8788), new (8752), one (8725), friends (8465), please (8232), pandemic (7614), support (7255), know (6931), going (6788), realdonaldtrump (6659), times (6462), world (6451), health (6449), day (6153), family (6010), go (5986), trump (5967), work (5862), would (5705), today (5602), take (5532)

In this list, the number in brackets represents the frequency of occurrence, also visualized in Fig 1 (in which greater size of the word indicates greater occurrence in the corpus). In this word cloud, the larger the word print, the more frequent its occurrence in the corpus. The most frequent word is "people" with about 19k occurrences, followed by "us" with about 13k occurrences. It should be mentioned that we cannot distinguish whether "us" means "United States" or the pronoun "us".

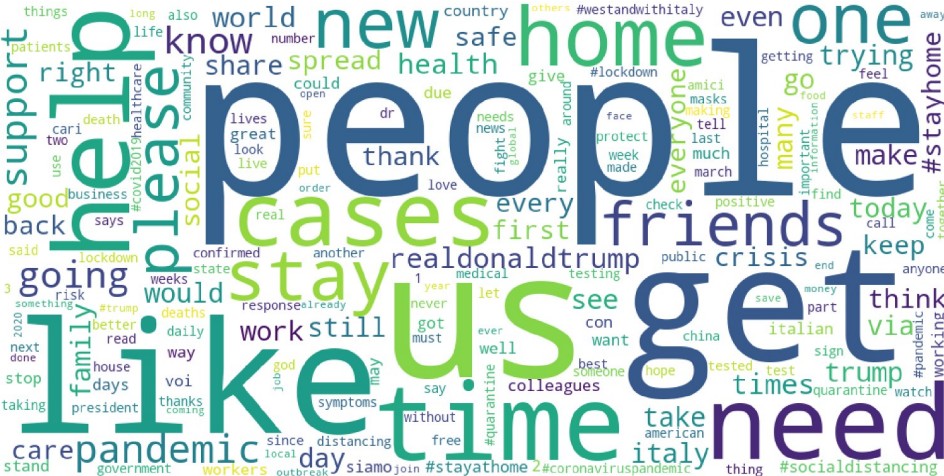

**Fig 1. Word cloud of the most common words in the corpus of over 200k collected tweets with at least one hashtag relating to the covid19 epidemic.**

## What type of topics are discussed on Twitter, in relation to Covid-19?

### Identifying topics in Covid-19 discourse on Twitter through topic modeling

A topic model is a generative statistical model for discovering "topics" that occur in a collection of documents. The topics identified by topic modelling are based on the occurrences of semantically related words. For example, "ball", "strike", "bat", "catcher", "hitter" and "diamond" and "fastball" are likely to appear in documents about baseball. A document typically concerns multiple topics in different proportions, and such proportions are reflected in the probability of the words related to each topic. The topics produced by topic modeling techniques are therefore probability clusters of related words, which are unlabeled (the model returns the cluster of words, not the name of such cluster and thus of the topic). Topics therefore need to be interpreted and labeled by the analysts.

Conversely, in communication sciences a frame is typically defined as consisting of two elements [28]: elements in a text such as words, used as framing devices, and (latent) information used as reasoning devices, through which a problem, cause, evaluation, and/or treatment is implied. A topic operationalized by topic modelling, in this sense, corresponds to the first of these two components of a frame, that is, a list of semantically coherent words used as framing devices).

In order to extract and identify topics from the corpus, we used a Latent Dirichlet Allocation algorithm (henceforth, LDA) [29]. LDA is an unsupervised machine learning algorithm that aims to describe samples of data in terms of heterogeneous categories. It is mostly used to identify categories in documents of text and thus appropriate to identify topics within the Covid-19 corpus of tweets. The study reported by Pruss and colleagues on the corpus of tweets related to the Zika epidemics [4], for example, used the same algorithm to identify topics within the corpus. For the purpose of our study we used the *Gensim* LDA-Multicore algorithm, which allowed us to parallelize the training of our data on multiple CPUs. As an unsupervised learner, LDA needs to be given the number of topics that it will try to divide the data into. Our exploratory approach included the search space for several different amounts of topics, thus

varying in the level of granularity represented within each topic. We hereby reported the results obtained from the division of the data into a relatively small number of topics (N = 4) and a relatively large number of topics (N = 16), to show and compare a less granular and a more granular division of the data. We expected to find broader and more generic concepts listed in the first analysis (N = 4) and more specific concepts in the fine-grained topic analysis (N = 16). These two numbers of clusters were chosen by investigating the data and are backed up by our post-hoc analysis of the LDA coherence measures. The preprocessing phase encompassed the six following steps:

- converting each tweet into a list of tokens (using *Gensim*'s simple_preprocess function)

- removing tokens with less than 3 characters (e.g. "aa", "fo", "#o")

- removing stopwords from the list of tokens (including updated stopwords from Stone et al. [30] and twitter specific stopwords.)

- removing Covid-19 words from the list of tokens (e.g. "covid", "nCov", "coronavirus" etc)

- turning the tokens into a bag-of-words, i.e. a list of tuples with the token and its number of occurrences in the corpus

We excluded terms like "coronavirus", "covid", "corona", "virus" or "nCov19" from the topic modeling because these do not add information about the topics themselves. The preprocessing resulted in a list of 415,329 tokens, that is, inflected word forms. We did not lemmatize the corpus, nor pos-tagged it for the purpose of our study, because different forms of a lemma can express different metaphor scenarios and therefore, they shall be preserved. Hence, for example, gerundive forms of verbs, as well as plural forms of nouns are present in the corpus, and the list of frame-related words is also composed by inflected word forms and not simple lemmas. Additionally, we trained another LDA model with a tf-idf (term frequency-inverse document frequency) version of the tokens. The tf-idf assigns a statistical relevance to each token based on how many times the token occurs and the inverse document frequency (a measure of whether the token is rare or common in the corpus) of that token. As its results did not add any further insight to our research, we included it in the online repository but do not discuss this model further in the current paper.

## Topic model analysis

Dividing the corpus into four topics through the LDA, we obtained a list of words for each topic and the weightage (importance). Fig 2 shows the word clouds with greater words signaling greater significance. Except for topic #II, all of the other topics included the word "pandemic" among their most important words and show a strong overlap. The weights (importance) and words for each topic allocated by the LDA model with N = 4 topics were the following:

LDA (N = 4, 6 passes):

**Topic #I**: 0.008 pandemic, 0.005 news, 0.004 data, 0.004 update, 0.004 world, 0.004 youtube, 0.003 information, 0.003 latest, 0.003 today, 0.003 april

**Topic #II**: 0.029 times, 0.028 friends, 0.027 family, 0.022 share, 0.021 italy, 0.021 trying, 0.014 support, 0.014 sign, 0.013 stand, 0.011 colleagues

**Topic #III**: 0.014 people, 0.013 cases, 0.011 trump, 0.009 realdonaldtrump, 0.008 like, 0.006 china, 0.006 world, 0.005 deaths, 0.005 pandemic, 0.004 going

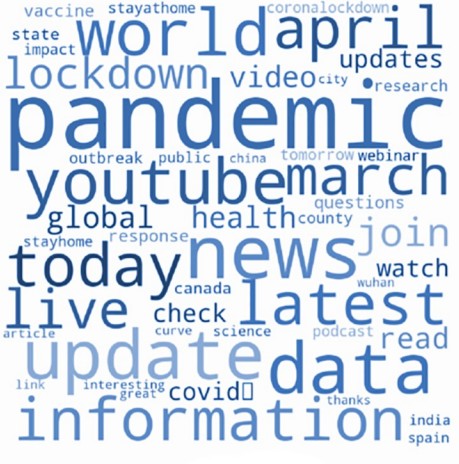

Topic #I: Communications and Reporting

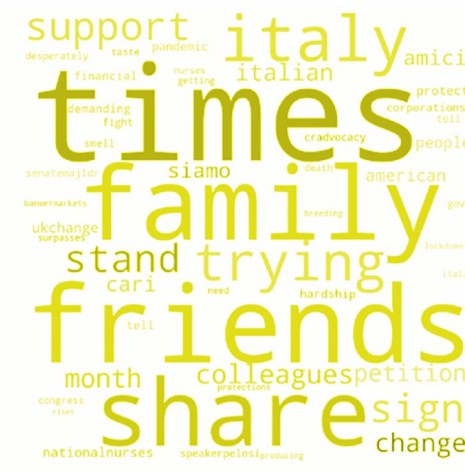

Topic #II: Community and Social Compassion

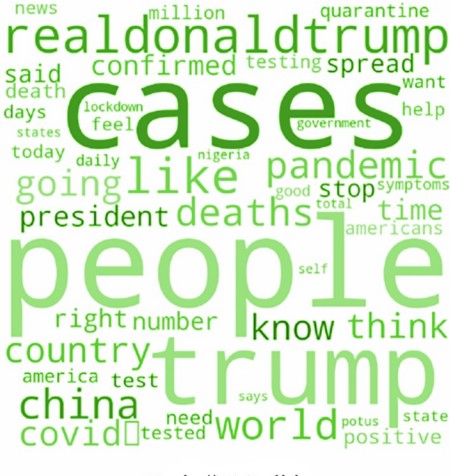

Topic #III: Politics

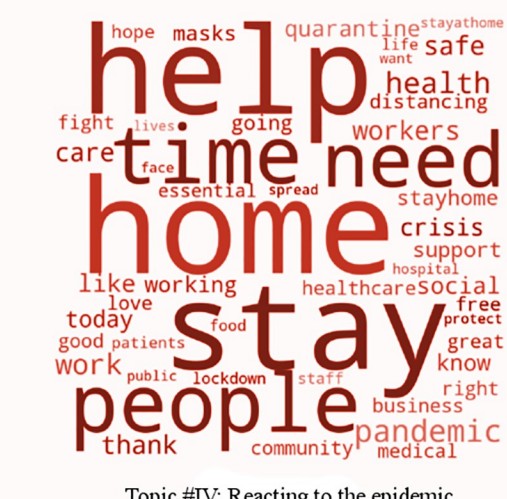

Topic #IV: Reacting to the epidemic

**Fig 2. Word clouds form N = 4 LDA topic modeling with greater words signaling greater significance.**

**Topic #IV**: 0.010 home, 0.008 help, 0.008 stay, 0.007 people, 0.007 time, 0.007 need, 0.007 pandemic, 0.006 health, 0.006 work, 0.005 safe

The results for 16 topics, displayed in Fig 3, showed a much greater diversity among the classes.

## Discussion

As previously mentioned, the LDA algorithm does not provide labels for the topics. The interpretation of the topics is left to the analysts. The topics identified by LDA analyzed above and visualized in Fig 2 can be labeled as follows:

**Topic #I**: *Communications and Reporting.*

**Topic #II**: *Community and Social Compassion.*

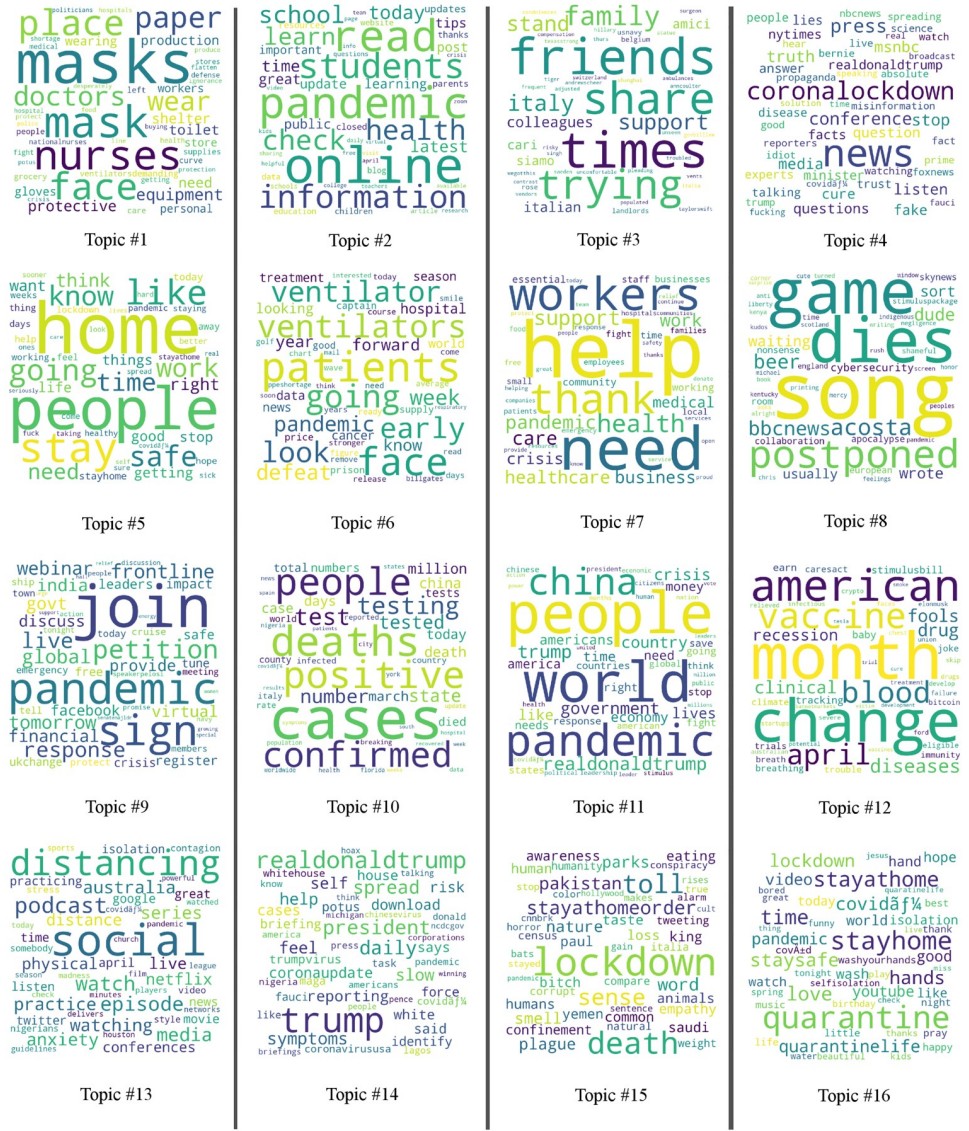

**Fig 3. Depiction of the word clouds for each of the 16 topics clustered by the LDA.**

**Topic #III**: *Politics.*

**Topic #IV**: *Reacting to the epidemic.*

The sixteen topic LDA model provided a more fine-grained view of topics that could be related to the 4 general topics. In the field of *Communication and Reporting*, we observed finer distinctions in topics #4, #11 and some in #15. Topic #11, in particular, is more focused on "World", "Trump" and "China", while topic #4 specifically encompasses "News", "Lockdown", "Press", and "Media". In the domain of *Community and Social Compassion*, topic #3 is very close to topic #II. Whereas, topic #13, #16, #5 relate to topics around the quarantine, self-isolation and in general *Reacting to the Epidemic* (#IV).

There are also some novel topics around treatment and medical needs (#1, #6, #7), around testing (#10) or working/studying from home (#2, #9 and parts of #12). Rather unrelated to the whole epidemic, a conglomerate of words can be found in topic #8 and parts in #12.

Finally, we provide an interactive online tool to explore the results of the LDA models for the 4 topic LDA model (https://bit.ly/3dCczfr) and for the 16 topic LDA model (https://bit.ly/3gUx5tU). These renderings have been produced using pyLDAvis [31].

## To what extent is the WAR figurative frame used to talk about Covid-19 on Twitter?

### Determining lexical units associated with the WAR frame

To investigate to what extent users use the WAR frame to talk about Covid-19, we needed to assess the amount of tweets that use war language in our corpus. To explore the lexical units associated with the WAR frame we took a double approach, using two tools. The first tool was the web-service *relatedwords.org*. This web-service provides a list of words (inflected word forms, not lemmas) related to a target word. This list is ranked through competing scores by several algorithms, one of which finds similar words in a word embedding [32] and another one queries *ConceptNet* [33] to find words with meaningful relationships to the target word. Choi and Lee [34] used the same web-service to expand the list of categories used to model conceptual representations for crisis-related tweets [34]. The list of words retrieved on *relatedwords* was adapted to our purpose. As a matter of fact, the list featured words such as "franco-prussian war" or "aggression". The former is a specific type of war and it includes the term "war" itself. We dropped any kind of specific war or terms that include a compound of war, e.g. "state of war". The latter term "aggression" is too broad and not closely related to the target word. Additionally, in case of doubt, we checked the term in an online dictionary to verify its relation to the war framing. The second tool used to prepare the list of lexical units related to the WAR frame was the MetaNet repository of conceptual metaphors and frames housed at the International Computer Science Institute in Berkeley, California [18]. Here, from the WAR frame (https://metaphor.icsi.berkeley.edu/pub/en/index.php/Frame:War) we selected the 12 words that were not yet included in the selection of lexical units based on *relatedwords*. Moreover, we dropped compound units that included words that we had already included in the list (e.g., "combat zone", because we featured already the word "combat") and two mis-spelled units ("seige" instead of "siege" and "beseige" instead of "besiege"). The total number of lexical units for the WAR framing was 91:

> **WAR (91)**: allied, allies, armed, armies, army, attack, attacks, battle, battlefield, battle-ground, battles, belligerent, bloodshed, bomb, captured, casualties, combat, combatant, combative, conflict, conquer, conquering, conquest, crusade, defeat, defend, defenses, destruction, disarmament, enemies, enemy, escalation, fight, fighter, fighting, foe, fortify, fought, grenade, guerrilla, gunfight, holocaust, homeland, hostilities, hostility, insurgency, invaded, invader, invaders, invasion, liberation, military, peace, peacetime, raider, rebellion, resist, resistance, riot, siege, soldier, soldiers, struggle, tank, threat, treaty, trench, trenches, troops, uprising, victory, violence, war, warfare, warrior, wars, wartime, warzone, weapon, alliance, ally, arsenal, blitzkrieg, bombard, front, line, minefield, troop, vanquish, vanquishment.

A methodological clarification shall be made explicit regarding the identification of lexical units used metaphorically in our corpus. In cognitive linguistics and metaphor studies, the procedure usually adopted for the reliable identification of words used metaphorically in linguistic corpora is MIPVU [37]. This procedure is applied manually, by multiple annotators, in content analyses where analysts make decisions about the metaphoricity of each lexical unit encountered in the text, based on information retrieved from dictionaries. Despite the high

reliability of this method, because it is performed manually, it cannot be applied to large corpora, like the corpus of tweets on which the current study is based. For this reason, we opted for the following procedure: we assumed that war-related lexical entries would be used metaphorically rather than literally, within tweets about Covid-19, and we qualitatively confirmed our intuition by manually looking at a subsample of tweets. We acknowledge the fact that in the tweets that we haven't manually checked it could be the case that some war-related words were used literally, for example to talk about soldiers getting infected with Covid-19 while being in service. However, we believe that this phenomenon may characterize a negligible amount of tweets, compared to the number of tweets in which war-related lexical entries are used metaphorically. Nonetheless, we acknowledge this limitation to our approach and we wish further studies to account for this possibility, and possibly to compare the metaphorical use to the literal use of war-related terms in Covid-19 discourse.

In order to understand where in relation to our predicted LDA topics the WAR frame was located, we collected all tweets that mentioned at least one term of the WAR frame and asked the LDA model to predict its topic. This way, we could identify the topics with the most or the least terms related to WAR.

## WAR framing results

Analyzing all tweets from the database, a total of 10,846 tweets contained at least one term from the WAR framing, which is 5.32% of all tweets. Of these, 1,253 tweets had more than one war-related term. The 20 most common war terms found in our database are hereby reported with relative percentage to all war terms and number of occurrences:

> **WAR**: fight (29.76%, 3228), fighting (10.65%, 1155), war (10.08%, 1093), combat (5.89%, 639), threat (5.13%, 556), battle (4.19%, 454), front line (3.82%, 414), military (3.61%, 392), peace (3.43%, 372), attack (2.95%, 320), enemy (2.61%, 283), defeat (2.51%, 273), violence (2.12%, 230), attacks (1.44%, 156), struggle (1,34%, 145), resist (1.23%, 133), soldiers (1.23%, 133), weapon (1,20%, 130), victory (0.95%, 103), wars (0.95%, 103)

Words that were virtually absent (or had very limited usage) in the context of Covid-19 on Twitter were: combatant (2x), combative (2x), disarmament (2x), gunfight (2x), invader (2x), treaty (2x), bombard (2x), minefield (2x), belligerent (1x), guerilla (1x), insurgency (1x), vanquish (1x), conquest (0x), blitzkrieg (0x), vanquishment (0x).

## LDA topic prediction of WAR tweets

The LDA model can predict the probability that a document belongs to a certain topic of the corpus. We therefore used this prediction method to investigate what topics are relevant for those tweets that feature WAR terms. For this, we tokenized all tweets that contain at least one WAR term and used both of our LDA models to suggest which of the four and sixteen topics those WAR related tweets most likely belong to. For the four-ways topic model we report a distribution in Fig 4.

For the sixteen-ways topic model, we report a distribution in Fig 5. This image shows that lexical units belonging to the WAR domain and therefore tweets that relate to the WAR frame are most likely to be found in tweets that belong to topics IV and I, and partly III (in the macro distinction of topics) and in tweets that belong to topics 2, 7 and 10 in the fine-grained distinction.

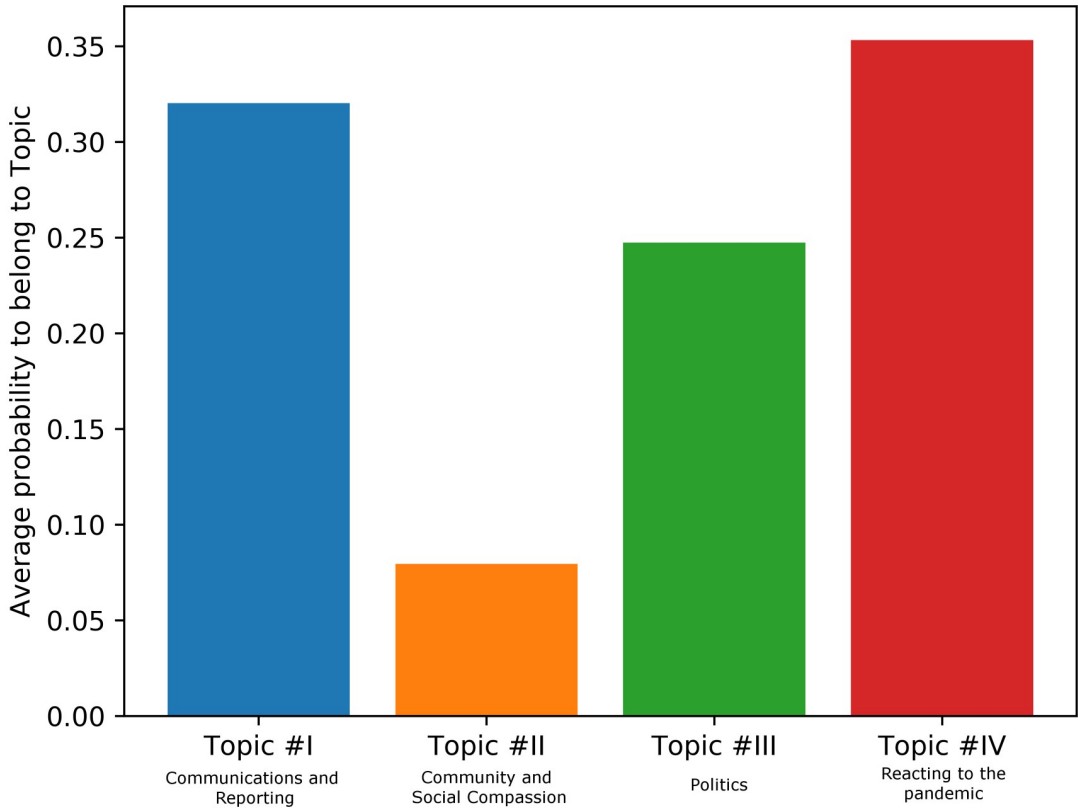

**Fig 4. LDA-predicted average probability of WAR term contributing to one of 4 topics.**

## Discussion

The results show that 5.32% of all tweets contain war-related terms and are therefore likely to frame the discourse around Covid-19 metaphorically, in terms of a literal war. While it is hard to evaluate in absolute terms the impact that this frame has on the overall discourse around Covid-19, we show in the next sections how the WAR frame compares to the usage of 3 other figurative frames, as well as to a literal frame.

The specific words within the WAR frame that appeared to be used in the tweets were "fight" "fighting", the very same word "war", "combat", "threat", and "battle". All these words carry a very negative valence and denote aspects of the war that relate to actions and events. This is probably due to the stage of the pandemic that we are in, that is, the emergency situation, and the related urgent need to take action and confront the negative situation. We cannot exclude that this tendency may change, once the pandemic moves into a different stage. In particular, it could be the case that when the emergency has passed, and we will move toward the next phase, in which we will leave the peak of the death and infection rates, the most frequent words used in relation to the WAR frame might relate to the identification of strategies to keep ourselves safe and to defend our community from potential new attacks.

In relation to the topic modelling of the war-related tweets, we showed that tweets that feature war-related terms are most likely to belong to topics IV, I and III, rather than to topic II. Interestingly, topic IV addresses aspects related to the reactions to the epidemic, including the measures proposed by the governments and taken by the people, such as self-isolating, staying at home, protecting our bodies and so forth. Our analysis therefore suggests that using war-

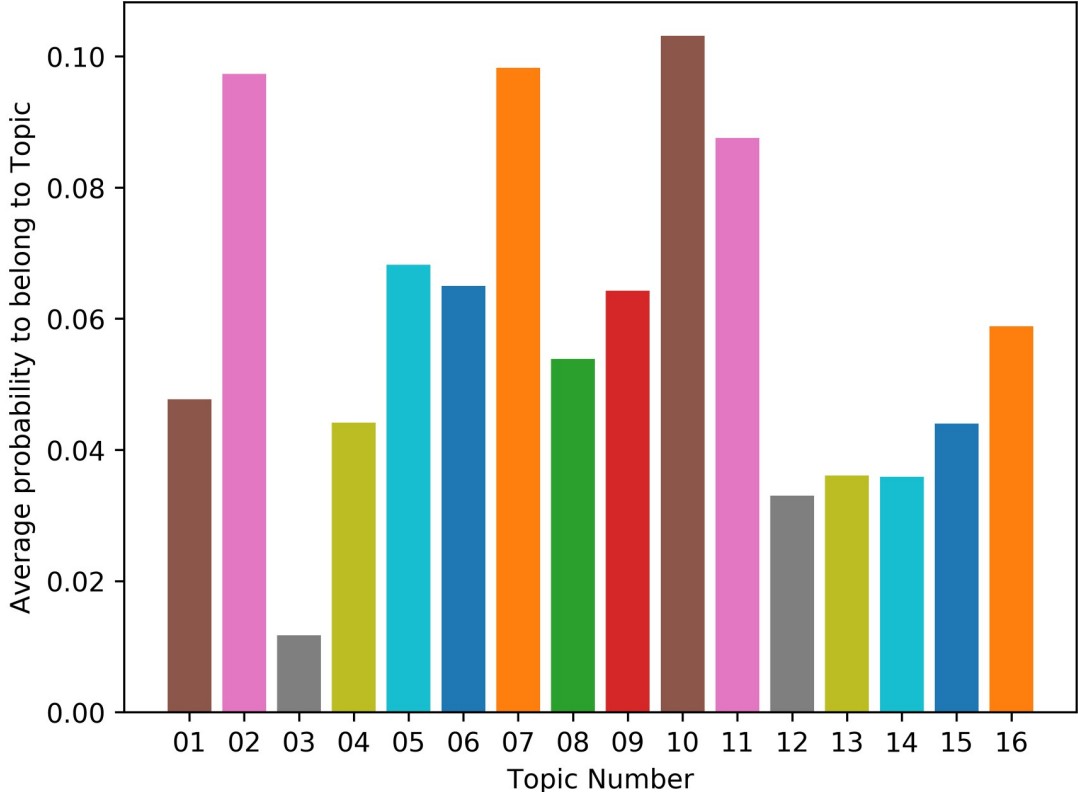

**Fig 5. LDA-predicted average probability of a WAR term contributing to one of 16 topics.**

related words is a communicative phenomenon that we use to express aspects of the Covid-19 epidemic related to the measures needed to oppose (fight!) the virus. Moreover, tweets that feature war-related words are also often classified within the topics I and III, which include the aspects related to communications and reports about the virus, and politics. We interpret these results arguing that public communications and political messages are likely to frame the discourse in the WAR framing. Finally, it might not come as a surprise the fact that topic II, which encompasses aspects of the discourse related to the familiar sphere, the community and the social compassion, does not relate well with the tweets containing war terms.

The fine-grained analysis into 16 topics shows some interesting trends as well. In particular, tweets containing war-related terms are particularly well represented in topics 2, 7, and 10. Topic 2 seems to relate to online learning and education, Topic 7 encompasses aspects related to the treatment of the virus, with words such as "workers", "health", "care", "help", "thank", "need", "support". Similarly, topic 10 relates to the diagnostics and treatment of the virus, with words such as "positive", "death", "cases", "tested", "people", "confirmed". Therefore, as the MetaNet WAR metaphor suggests, and as we described in the Theoretical Background of this paper, it is the discourse around the disease treatment and its diagnostics that are likely to be framed figuratively in terms of a war. Conversely, topic 3, which is characterized by words like "friends", "share", "trying", "family", "time" and therefore addresses intimate social relations and personal affective aspects related to Covid-19, is not related to the WAR frame: tweets addressing these aspects do not employ military lexical units.

We note that an LDA model uses randomness in its training and inference, therefore training a new model with the same parameters will always yield slightly different topic distributions and there are different ways and limitations to analyze those [35].

## Are there alternative figurative frames used to talk about Covid-19 on Twitter?

### Search method for alternative framings and relevant lexical units therein

In order to identify whether or not the war framing is particularly relevant, we explored alternative framings used in discourses on viruses. For this purpose we used the metaphor exploration web services by [36], called *MetaphorMagnet* (http://bonnat.ucd.ie/metaphor-magnet-acl). Using the keywords "virus" and "epidemic" we selected the following alternative frames, which could be in principle used to frame the discourse around Covid-19: STORM, MONSTER and TSUNAMI. These figurative frames have been reported also within the crowd-sourced observations collected by the #ReframeCovid initiative on Twitter. Other possible figurative frames are reported within this initiative too. These are, for example, GAME and the sub-frame SOCCER GAME, used in the Spanish press according to the community of Spanish cognitive linguists. However, lexical units such as "game", "football", "soccer", "game season", and so on, are likely to be used literally in the tweets, to refer to the fact that all sport events and thus all games have been suspended, due to the epidemic. Another frame that has been observed in the press and tagged as #ReframeCovid is the FLOOD frame. However, through a quick search on Metaphor Magnet, we realized that this frame has too many shared lexical units with STORM and TSUNAMI, and was therefore discarded. Moreover, we observed that the wordlists of the frames STORM and TSUNAMI contain shared words. However, dictionary definitions of these two terms suggest that the two phenomena are quite different. For example, the MacMillian online dictionary defines STORM as an occasion when a lot of rain falls very quickly, often with very strong wind or thunder and lightning. Conversely, TSUNAMI is defined as a very large wave or series of waves caused when something such as an earthquake moves a large quantity of water in the sea. Because the two concepts denote different phenomena, the fact that they share a few words does not constitute a redundancy.

In order to select the lexical units within each of the alternative frames, we used the tool *relatedwords*, already used for the WAR frame, for consistency. However, because these alternative frames are arguably less conventionalized, none of them is included in the list of frames on MetaNet. Thus, *relatedwords* was the only tool we used to harvest lexical units for the alternative frames. We created three list of lexical units:

**STORM (57)**: thunderstorm, rain, lightning, snowstorm, blizzard, wind, hurricane, weather, rainstorm, typhoon, tempest, precipitation, beaufort, snow, cyclone, meteorology, hail, hailstorm, windstorm, flooding, thunder, tornado, monsoon, rainfall, rage, force, disaster, ice, storm, atmospheric, disturbance, wildfire, clouds, firestorm, ramp, tornadoes, fog, winds, rains, waves, landfall, thunderhead, duststorm, tides, gusts, floodwaters, wave, cloud, swells, cloudburst, anticyclone, downpour, sandstorm, stormy, whirlwinds, storms, oceanographic.

**MONSTER (51)**: freak, demon, devil, giant, ogre, fiend, zombie, frankenstein, bogeyman, werewolf, horror, mutant, creature, dragon, superhero, goliath, behemoth, monstrosity, colossus, legend, evil, lusus, naturae, mouse, beast, boogeyman, leviathan, dracula, monstrous, teratology, villain, killer, ghost, gigantic, siren, superman, vampire, undead, psycho,

monster, chimera, godzilla, fiction, mythology, mutation, demoniac, manatee, mermaid, monsters, spider, bug.

**TSUNAMI (50)**: earthquake, disaster, tide, oceans, calamity, catastrophe, tragedy, wavelength, wind, period, cataclysm, flood, eruption, tidal, seiche, quake, thucydides, floods, floodwater, cyclone, devastation, ocean, surface, wave, coastlines, typhoon, waves, hurricane, magnitude, aftershock, mudslide, seafloor, richter, seawall, seismic, landslide, tsunamis, aftershocks, flooding, torrential, earthquakes, deepwater, triggering, tsunami, tremors, mudslides, riptide, rains, whirlpool, pacific.

As for the WAR frame, we ran these lists against our corpus and compared the frequency of occurrences within the corpus across the different framings.

## The literal frame of FAMILY used as control

To evaluate the relevance of the figurative frames in the corpus of tweets, we compared the occurrence of the lexical units listed therein with those listed within a frame that we expected to occur in the literal sense: the FAMILY frame. The word list of lexical entries related to this frame encompasses the following words:

**FAMILY (66)**: marriage, household, kin, house, kinfolk, home, lineage, kinship, parent, relative, clan, cousin, children, child, sister, mother, father, uncle, nephew, brother, grandson, son, grandfather, grandmother, kinsfolk, ancestor, consanguinity, tribe, sibling, subfamily, kindred, stepfamily, couple, family, sib, foster, parentage, menage, phratry, folk, daughter, kinsperson, aunt, grandma, granddaughter, grandaunt, stepbrother, niece, stepson, dad, stepdaughter, stepfather, wife, husband, daddy, parents, elder, daughters, mom, siblings, stepmother, grandpa, grandparents, relatives, widow, spouse.

## Alternative framing results

The terms belonging to the frame STORM were found in 3,036 tweets (1.49% of all tweets). The terms in the MONSTER frame were found in 1,382 tweets (0.68% of all tweets). The terms in the TSUNAMI frame were found in 2,304 tweets (1.13% of all tweets). The terms in the literal frame (FAMILY) were found in 24,568 tweets (12.06% of all tweets). The difference between the frequency of occurrence of the frames, and in particular of the sets of words related to each frame, is statistically significant (Cochran's Q test statistic = 47,226.72, df = 4, $p < 0.001$). We then looked at the distribution of the frequencies by which the terms within each framing were used and observed that they all tended to follow Zipf distributions (see Fig 6, where the term "fight" from the WAR frame has more than 3,000 occurrences in the tweets). In other words, within each frame there were few words used very frequently, but many words were rarely used. Moreover, although this is not visible on the plot in Fig 6, in the online repository we stored the full list of lexical units within each frame. Among the top ranked ones for the FAMILY frame we found "home", "family", "house", "children", "parents", "wife", "son", and "mom". For the STORM frame among the most frequently used lexical units we found "force", "disaster", "weather", "ice", "wave", "storm", "cloud", and "rain". For the MONSTER frame among the most frequently used lexical units we find "evil", "horror", "killer", "giant", "monster", "legend", "ghost", "zombie", "devil", "fiction", "bug" and "beast". Finally, for the TSUNAMI frame among the most frequently used lexical units we found "period", "disaster", "wave", "tragedy", "catastrophe", and "waves".

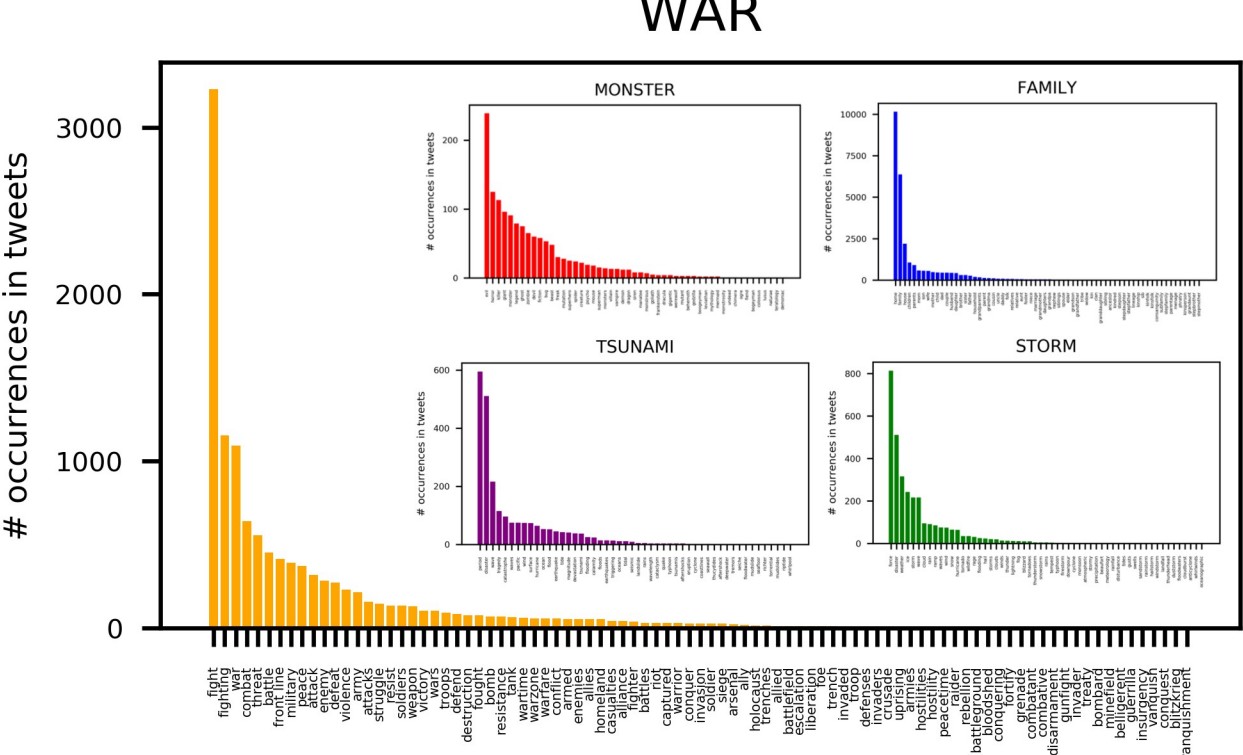

**Fig 6. Five histograms depicting the occurrences of terms for each frame within the corpus.**

Also, the total number of words within each frame was different, with the WAR frame featuring more words than the other figurative frames. In order to compare how frequently the WAR frame was used in Covid-19 discourse, compared to other possible figurative frames (and a literal frame), it was necessary to have lists of lexical units of the same length because longer lists could have yielded larger numbers of tweets in the corpus than shorter lists, in principle. Therefore, we decided to evaluate two subsets of the term lists for each framing, setting two cutoffs at N = 30 and N = 50 terms on each list. In this way, we only considered the top 30 and then 50 most relevant (i.e., most frequently used) terms within each frame. We then compared the number of tweets featuring words from these lists, which were now comparable in length.

Table 2 reports the number of tweets featuring at least one lexical unit related to a frame, and the general percentage of tweets in the corpus that can be related to these frames. Results showed that the literal frame FAMILY is substantially more frequently used in the discourse

**Table 2. Proportions of tweets that contain at least one of the terms from each of the frames with term list size N = 30 and N = 50.**

| Frame | # of tweets with at least 1 word from 30-item list | Percentage of tweets over the whole corpus (30 terms) | # of tweets with at least 1 word from 50-item list | Percentage of tweets over the whole corpus (50 terms) | Total Tweets |
|---|---|---|---|---|---|
| **WAR** | 10,107 | 4.96% | 10,704 | 5.25% | 203,756 |
| **FAMILY** | 24,269 | 11.91% | 24,563 | 12.06% | 203,756 |
| **STORM** | 3,017 | 1.48% | 3,035 | 1.49% | 203,756 |
| **MONSTER** | 1,348 | 0.66% | 1,382 | 0.68% | 203,756 |
| **TSUNAMI** | 2,217 | 1.09% | 2,304 | 1.13% | 203,756 |

on Covid-19 than the figurative frames. However, among figurative frames, the WAR frame covered a higher portion of the tweets in our corpus than the other figurative frames. The table also shows that there is no substantial difference between the coverages of the corpus obtained using the 30 words and the 50 words lists of lexical units for each frame.

## Replication studies

Given the timeliness of this study, our first analysis was based on a corpus of tweets covering 2-weeks' time. During the submission and review process, more data (more tweets) obviously became available. We therefore replicated our analysis in which different frames are compared, with new data. First, we constructed an additional corpus of tweets like the first one, with tweets produced in the two weeks that followed the timeframe of the first corpus. Second, we replicated our study using an external dataset with more than 1.2 million tweets, which became available during the revision process of the current article. The choice of tweets to be collected from an external dataset has been limited by the factors that define our corpus: no retweets, only unique tweeter's tweets, English, from 20.03.2020–20.05.2020 and maximum memory limit (due to hardware constraints). The external corpus included 1,213,420 tweets from Lamsal's Coronavirus Tweets Dataset [25] over two months. Fig 7 presents an overview of the comparison and Table 3 provides the descriptive statistics and results of Cochran Q tests.

As Fig 7 shows, the distribution of the five frames across the first corpus of tweets is very similar to the distributions observed in the replication studies. The increase of >0.22% in the WAR framing from our first corpus (W&B 2 weeks) to the second corpus (W&B 2 months) could be partially explained by new debates entering the discourse, with the development of the epidemic.

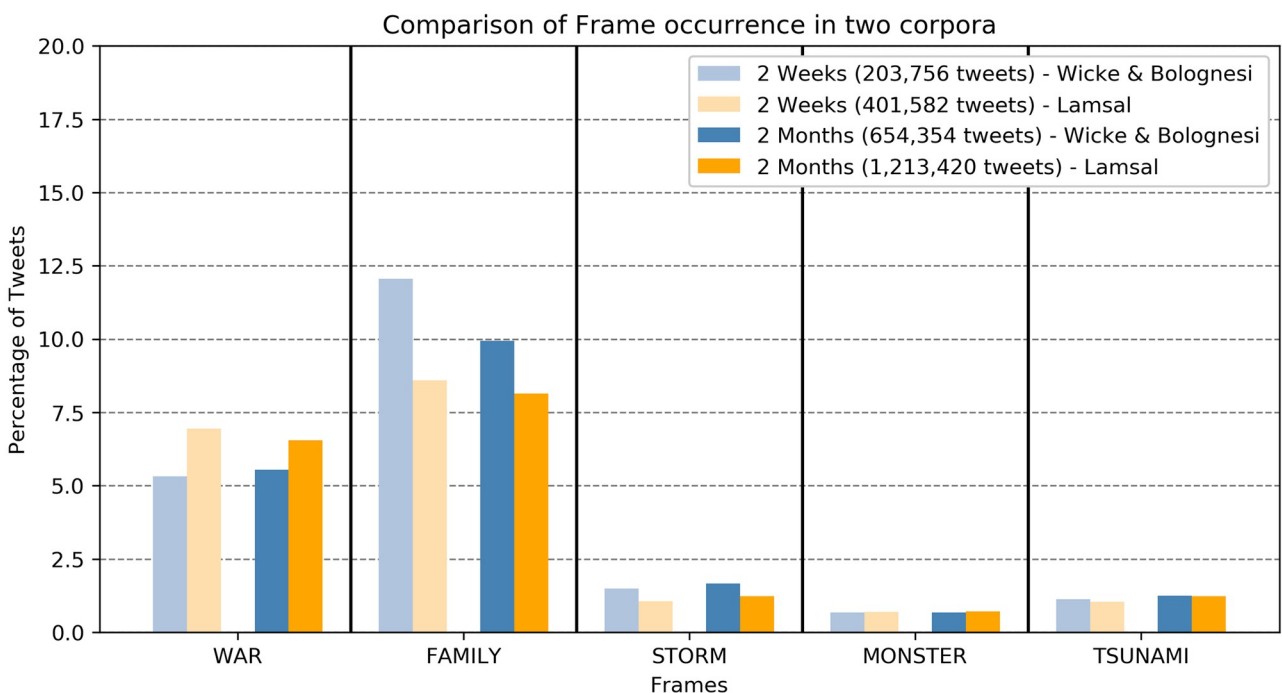

**Fig 7. Comparison of the two corpora for five frames and two time spans (2 weeks, 2 months).** Each bar indicates the percentage of a frame within the respective corpus.

**Table 3. Results of the comparison of the two corpora for five frames and two time spans.**

| | WAR | FAMILY | STORM | MONSTER | TSUNAMI | Tweets Tot. | Cochran Q statistic (df = 4) |
|---|---|---|---|---|---|---|---|
| W&B | 5.32% | 12.06% | 1.49% | 0.68% | 1.13% | 203,756 | Q = 47,226.72 |
| 2 weeks | | | | | | | p<0.0001 |
| Lamsal | 6.94% | 8.6% | 1.06% | 0.70% | 1.05% | 401,582 | Q = 57,159.11 |
| 2 weeks | | | | | | | p<0.0001 |
| W&B | 5.54% | 9.95% | 1.67% | 0.67% | 1.25% | 654,354 | Q = 110,616.87 |
| 2 months | | | | | | | p<0.0001 |
| Lamsal | 6.55% | 8.14% | 1.24% | 0.72% | 1.24% | 1,213,420 | Q = 173,630.43 |
| 2 months | | | | | | | p<0.0001 |

The comparison between our data and the data extracted from the "Coronavirus Tweets Dataset" by Lamsal over the same time-frame shows that overall the relative order of proportions is the same as in our analysis: Family > War > Storm + Tsunami + Monster. The differences in these proportions (Family proportion decreased by 3.46%, War increased by 1.62%) can be explained by different keywords that have been used to acquire the Lamsal dataset. This dataset, in fact, has been constructed using keywords that we did not use to construct our dataset, notably the keyword "Corona". This is arguably a more colloquial expression that we chose to not include in our set of keywords. Moreover, in Lamsal's dataset keywords change multiple times during the process of data mining, while we keep the same set of keywords, day after day.

## Discussion

Our results show that the literal frame used as control (FAMILY) covers a wider portion of the tweets in the corpus while the figurative ones cover substantially less tweets. This is not particularly surprising, as previous literature shows that overall metaphor-related words cover only a percentage of the discourse, and that literal language is still prevalent. Steen and colleagues [37], for example, report that literal language covers 86.4% of the lexical units, while metaphor-related words cover just 13.6% of the lexical units. Their analysis is based on a sub-corpus of the BNC that encompasses 187,570 lexical units extracted from academic texts, conversations, fiction and news texts. All parts of speech are included in their analyses, including function words (such as prepositions and articles). Based on these statistics, we would expect to find around 13% of the lexical units in our corpus to be used metaphorically. This percentage would need to include pervasive metaphorical uses of function words such as prepositions, as well as all words used metaphorically, which can be related to any figurative frame. In this perspective, we believe that the percentage of use of the WAR frame reported in our study suggests that this frame is particularly frequent. In our specific study, based however on a limited number of possible figurative frames, lexical entries related to the WAR frame cover more than one third of all the words attributed to the metaphorical frames hereby investigated.

Within the FAMILY (literal) frame, the top words (i.e., most frequent words) that are used in the tweets denote family members and family relations. Within the STORM frame, words that suggest the most frequently used words seem to denote concrete entities that can be typically observed within a storm scenario. In general, from a qualitative standpoint, it can be observed that the different frames are used to tackle different aspects associated with Covid-19. Words in the STORM and in the TSUNAMI frames seem to relate to events and actions associated with the arrival and spreading of the pandemic (e.g., "wave", "storm", "tide", "tsunami", "disaster", "tornado"). Words within the MONSTER framing, instead, are mostly nouns and

can be arguably used to frame the discourse about the behavior of the virus, in a rather person-ified way, which is loaded with emotional content and extremely negative valence (e.g., "devil", "demon", "horror", "monster", "killer"). This phenomenon, overall, supports the idea that different frames are apt to elaborate the discourse around different aspects, related to a topic, and that therefore multiple frames are more likely to enable the effective description and discussion of different aspects related to the Covid-19 reality.

Finally, the series of replication analyses conducted on new and alternative corpora show that the results hereby reported are similar across corpora and therefore consistent. Small variations between our corpora and the resource provided by Lamsal may be due to the keywords used to construct the datasets. Conversely, differences between the 2-weeks and the 4-weeks corpora may be due to a change in the discourse, reflecting the natural evolvement of the pandemic. In this perspective, future research, which we are currently pursuing, will show in a longitudinal perspective the development of the different topics and figurative frames used in the discourse around Covid-19 week after week.

## General discussion and conclusion

In this study we explored the discourse around Covid-19 in its manifestation on Twitter. We addressed three specific research questions: 1. What are the topics around which the Twitter discourse revolves, in relation to Covid-19; 2. To what extent the WAR framing is used to model the Covid-19 discourse on Twitter, and specifically in relation to which topics does this figurative framing emerge; 3. To what extent does the WAR framing compare to other potentially relevant figurative framings related to the discourse on viruses, and to the literal framing FAMILY.

In general, we found that the topics around which most of the Twitter discourse revolves, in relation to Covid-19, can be labelled as *Communications and Reporting*, *Community and Social Compassion*; *Politics* and *Reacting to the epidemic*. A more fine-grained analysis brings to light topics related to the treatment of the disease, mentioning people involved in this operation such as doctors and nurses, and topics related to the diagnostics of the virus. We also found that these specific topics appear to be those in which the WAR frame is particularly relevant: most lexical units within this frame are found in tweets that get automatically classified within the specific topics of virus treatment and diagnostics. Moreover, in relation to the second research question, we observed that there is a little number of lexical units related to war that are very frequently used, while the majority of war-related words are not used to frame the discourse around Covid-19. The more frequently used words refer to actions and events, such as "fighting", "fight", "battle", and "combat". As we anticipated, this might be a peculiarity of the stage of the pandemic we are currently living, which is the peak of the emergency. We do not exclude that with the development of the pandemic and the passage to the next phase (i.e., leaving the peak) also the most frequent words used within the WAR frame will change, to exploit new aspects of this frame that are relevant to the new situation. Finally, in relation to the third research question, we compared the frequency by which the WAR frame, the FAMILY literal frame and three other figurative frames are used. We found that while the FAMILY literal frame used as control covers a wider portion of the corpus, among the alternative figurative frames analyzed (MONSTER, STORM and TSUNAMI), the WAR frame is the most frequently used to talk about Covid-19, and thus, arguably, the most conventional one, as previous literature also suggests.

It should be mentioned that the current study is based on a corpus of tweets (and then replicated on other corpora of tweets) that has been constructed on the basis of precise methodological criteria. Notably, we dropped retweets from our corpus, and we retained only one tweet

per user, to avoid the bias introduced by super-tweeters, that is, users (sometimes bots) who tweet many times a day and that may have monopolized the sample of tweets used for our analyses. These two operations, which were motivated by methodological requirements, on one hand made our corpus probably more robust, balanced and representative for the phenomena to be investigated, but on the other hand neglected these peculiarities of Twitter. As a matter of fact, Twitter, as a social network typically encompasses retweets (that is, duplicated tweets, which can be retweeted sometimes thousands of times) and it features super-tweeters. Additionally, information propagated by super-tweeters, can "go viral" or gain popularity, measured by likes and retweets. Therefore, a limitation of our study is that our findings may not reflect the actual distribution of topics and figurative frames on Twitter as a social media network per se. Rather, we argue, our findings reflect the way in which a wide selection of American-English speakers conceptualize and talk about Covid-19 on Twitter. In this sense, the approach adopted in the present study is embedded in common practices used in cognitive linguistics, discourse analysis and corpus linguistics. We acknowledge the fact that in scientific fields such as social media monitoring, criteria such as preserving the dynamics that characterize specifically the Twitter platform, such as retweets and super-tweeters, are particularly important. In these fields the construction of the sample of tweets (the corpus) would have been performed differently, to include retweets and without controlling for super-tweeters.

Taken together our results suggest what has been previously argued in discourse analysis, that is the relative pervasiveness of the WAR frame in shaping public discourse. In our study we show that this tendency applies also to the discourse on Covid-19, as previous literature would have predicted, given the frequent use of this frame in discourses on diseases and viruses. However, we have also found that this frame is used to talk about specific aspects of the current epidemic, such as its treatment and diagnostics. Other aspects involved in the epidemic are *not* typically framed within a WAR. This point is particularly important. The WAR frame, like any other frame, is useful and apt to talk about some aspects of the pandemic, such as the treatment of the virus and the operations performed by doctors and nurses in hospitals, but not to talk about other aspects, such as the need to feel our family close to us, while respecting the social distancing measures, or the collaborative efforts that we should undertake in order to #flattenthecurve, that is, diluting the spreading of the virus over a longer period of time, so that hospital ICU departments can work efficiently without getting saturated by incoming patients. In this sense, future studies could focus on the systematic identification of alternative figurative framings actually used in the Covid-19 discourse to tackle different aspects of the epidemic, but could also focus on the generation of additional frames, which can help communities to understand and express aspects of this situation that cannot be expressed by the WAR frame. A collection of different frames and metaphors that tackle different aspects of the current situation, or a *Metaphor Menu* (http://wp.lancs.ac.uk/melc/the-metaphor-menu/), as Semino and colleagues proposed in relation to cancer discourse [15], is arguably the most desirable set of communicative tools that, as language, communication, and computer scientists, we shall aim to construct in these current times, as a service to our communities.

## Acknowledgments

The authors would like to thank all doctors, nurses, health-care workers, grocery store workers and anyone else at the front line of this epidemic.

## Author Contributions

**Conceptualization:** Philipp Wicke, Marianna M. Bolognesi.

**Data curation:** Philipp Wicke, Marianna M. Bolognesi.

**Formal analysis:** Philipp Wicke, Marianna M. Bolognesi.

**Investigation:** Philipp Wicke, Marianna M. Bolognesi.

**Methodology:** Philipp Wicke, Marianna M. Bolognesi.

**Project administration:** Philipp Wicke, Marianna M. Bolognesi.

**Resources:** Philipp Wicke, Marianna M. Bolognesi.

**Software:** Philipp Wicke.

**Supervision:** Marianna M. Bolognesi.

**Validation:** Marianna M. Bolognesi.

**Visualization:** Philipp Wicke.

**Writing – original draft:** Philipp Wicke, Marianna M. Bolognesi.

**Writing – review & editing:** Philipp Wicke, Marianna M. Bolognesi.

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
