## [Decision Letter · Decision Letter 0]

18 May 2020

PONE-D-20-10986

Framing COVID-19 How we conceptualize and discuss the pandemic on Twitter

PLOS ONE

Dear Mr. Wicke,

Thank you for submitting your manuscript to PLOS ONE. After careful consideration, we feel that it has merit but does not fully meet PLOS ONE’s publication criteria as it currently stands. Therefore, we invite you to submit a revised version of the manuscript that addresses the points raised during the review process.

I have benefitted from reviews by five colleagues who are all leading experts in this area. As such, you will be delighted to see that the reviewers have provided incredibly constructive and clear feedback, not only on technical issues relating to methodology, but also on the interpretation of the findings in the grand scheme of things. I invite you to engage thoroughly with every one of the reviewers’ points, as I am sure this will result in a stronger contribution.

We would appreciate receiving your revised manuscript by Jul 02 2020 11:59PM. To enhance the reproducibility of your results, we recommend that if applicable you deposit your laboratory protocols in protocols.io, where a protocol can be assigned its own identifier (DOI) such that it can be cited independently in the future. For instructions see: http://journals.plos.org/plosone/s/submission-guidelines#loc-laboratory-protocols

We look forward to receiving your revised manuscript.

Kind regards,

Panos Athanasopoulos, Ph.D

Academic Editor

PLOS ONE

2. In your Methods section, please include additional information about your dataset and ensure that you have included a statement specifying whether the collection method complied with the terms and conditions for the websites from which you have collected data.

Reviewers' comments:

Reviewer's Responses to Questions

**Comments to the Author**

1. Is the manuscript technically sound, and do the data support the conclusions?

Reviewer #1: Partly

Reviewer #2: Yes

Reviewer #3: Partly

Reviewer #4: Partly

2. Has the statistical analysis been performed appropriately and rigorously? 

Reviewer #1: Yes

Reviewer #2: Yes

Reviewer #3: I Don't Know

Reviewer #4: No

3. Have the authors made all data underlying the findings in their manuscript fully available?

Reviewer #1: Yes

Reviewer #2: Yes

Reviewer #3: Yes

Reviewer #4: Yes

4. Is the manuscript presented in an intelligible fashion and written in standard English?

Reviewer #1: Yes

Reviewer #2: No

Reviewer #3: Yes

Reviewer #4: Yes

5. Review Comments to the Author

Reviewer #1: The paper analyzes Tweets that are used to talk about Covid-19 in three studies. The first identifies topics of these tweets. The second quantifies the prevalence of WAR metaphors in general, and across the topics identified in the first study. The third explores alternative figurative frames for the virus, along with one non-figurative frame (FAMILY). It is a timely study with interesting methods and results. I enjoyed reading it. I have suggestions for a revision, along with some methodological questions detailed below.

Introduction

1. The introduction could be better connected with the methods used in and research questions explored in the studies. Why is it interesting and important to determine what topics are discussed on Twitter in relation to Covid-19 or how frequently WAR and other metaphors are used? For example, could the topic modeling help to improve surveillance and forecasting about the disease? What frequency of WAR metaphors should we expect? How and why were the specific alternative frames chosen? Are there relevant theories in cognitive linguistics that the current work informs?

2. The emphasis and major strength of the paper seems to be the focus on WAR metaphors. The prevalence of WAR metaphors is quantified for an important issue, by topic, and in the context of other metaphoric frames (and a non-metaphoric frame). To highlight these strengths, I would encourage the authors to emphasize the novelty of quantifying the WAR frame using automated methods and real world data. Maybe switching the order of studies 1 and 2 would be helpful? (The results of Study 1 are difficult to interpret on their own).

3. The “Theoretical Background” section describes past work that is certainly interesting and relevant, but it doesn’t really identify theoretically motivated questions that the current work is well-suited to address. In most cases, it emphasizes practical applications and real world issues and/or methods (e.g., the relationship between measuring public sentiment on Twitter and addressing public health issues). Maybe rename this section for clarity.

Smaller points related to the introduction:

4. On p. 3 the authors note that around “16K Tweets are posted by Twitter users every hour, containing a hashtag such as #coronavirus, #Covid-19 or #COVID.” Is this based on the current data collection or a metric that is computed by Twitter? What is the temporal window for this statistic?

5. It seems like there is a typo on p. 4: “Unlike the articles on magazines and journals typically used for corpus analyses of this kind, Twitter does contain messages written by journalists and other experts in mass communication, as most tweets are provided by non-expert communicators.” Maybe “as” should be “although”?

Methods

1. Please clarify the following questions:

a. How were the 25k tweets per day selected? It seems like the algorithm picked up the first 25k tweets that included one of the specified hashtags per day, but that’s not explicitly stated. How long did it typically take to get to 25k tweets?

b. Were retweets included?

c. For a given person who tweeted multiple times in a day about the virus, was it only their first tweet that was included? What was the filtering criteria/method?

d. Were Tweets in languages other than English included?

2. It should be possible to clarify some of the hedging in the sentence, “our corpus arguably encompasses mainly tweets produced by users residing in the USA, where the time of data collection corresponds to awake hours, and the targeted language corresponds to the first language of many (if not most) US residents” (p. 12). I realize that the location data was stripped from the text before it was stored, but the text is initially tagged with the location data, so maybe it is possible to estimate the percentage of tweets from the US. Language use questions are asked on the US census every 10 years and roughly 80% of US residents report speaking English at home. A citation would make the case stronger.

3. How were the topic numbers (n = 4 vs. 16) decided in the first study? I find the results of this study hard to interpret. I think they are more informative when presented in concert with the results of study 2.

4. Categorizing language as metaphorical or not is tricky. There is a fair amount of work on this and the most common approach uses expert coders (see, e.g., Steen et al, 2010, A Method for Linguistic Metaphor Identification). I think the approach taken in the paper is reasonable but I think some limitations should be acknowledged. For example, there would probably be some disagreement by coders over which conceptual metaphors individual instances of “fight” appeal to (war vs. boxing vs. games, etc) and even whether or not particular instances are metaphorical or not.

5. Include some discussion about the relationship between the STORM and TSUNAMI categories. At first glance it seems like all instances of TSUNAMI should also be instances of STORM, but the current approach establishes these categories as mostly (completely?) non-overlapping.

6. I like the comparison of the metaphorical frames to the FAMILY non-metaphorical frame, but the FAMILY frame also seems fairly different in that it is more of a topic than a frame (i.e. in some ways more akin to the topics identified in Study 1). I don’t think this needs to be changed, but it seems worthy of a little discussion.

7. Include a note on how comparisons will be made. No inferential statistics are presented, which is common in cognitive linguistics, but the question about whether the number of cases is meaningfully different by frame or topic type will likely arise for readers.

Results

1. It’s hard to interpret the results of Study 1. Are these topics the ones we would expect? Do they inform theory? Do they inform practice? Could they have come out any other way?

2. What inferences can we draw about the relationship between WAR metaphors and topics from the results of Study 2?

3. If space is an issue, I think the comparison of the 30 vs 50 terms approaches on pp 26-28 could be cut.

4. Small point: There is no General Discussion section, although it is alluded to on p. 24.

5. Small point: Include a citation (and, ideally, a more precise statistic) in the sentence “…as previous literature shows that metaphor-related words cover only a percentage of the discourse, and that literal language is still prevalent” (p. 28). What percentage? Are there other metaphors that might be prevalent?

6. It would be nice to ground some of the qualitative observations noted in the discussion. For example: “Words in the STORM and TSUNAMI frames seem to relate to events and actions associated with the arrival and spreading of the pandemic…” (p. 28).

7. The paper ends by introducing the idea of a Metaphor Menu, which is interesting but it doesn’t logically fall out of the current study in my opinion. Maybe this idea could be discussed a little more.

Reviewer #2: The authors examined the (metaphorical) content of tens of thousands of English tweets surrounding the Covid-19 pandemic, scraped from two recent weeks from largely American twitter users. Topic modeling revealed several common themes (4 and also 16; more on this below), and that war metaphors were somewhat common (~4% prevalence), for some topics more than others, and appeared more frequently than other metaphorical domains. They argue that this is consistent with other empirical and theoretical research on the use of war metaphors in public discourse, but they now provide evidence this extends to everyday lay discourse online.

In general, I thought this was a timely article dealing with an important topic of interest to a variety of scholars, and a nice extension of previous work and theoretical musings on the use and prevalence of war metaphors. I think the methods and analyses were thoughtful and for the most part sound (though, as I detail below, I was confused about some of the details), and the results were solid. That said, I have a variety of comments and concerns that I think the authors should address in a revision before the manuscript is considered for publication.

One overarching concern is that the paper feels like it was rushed to submission and therefore the writing and overall organization are not quite up to the standards of a publishable manuscript. I understand the authors’ sense of urgency in getting this paper out there while the global pandemic crisis is still at its peak, and that they literally wrote it over the past few weeks, but I think extra care needs to be taken during any revision process to make sure the writing is improved. There were many grammatical and punctuation errors throughout the paper, along with confusing sentences and shifts in tense (if they want to refer to “now,” they should stick with phrasing like “at the time of writing,” which they were not consistent with throughout the paper). At times it was difficult to follow the logic of their thinking or make sense of some of the details of the methods and analyses.

One of the issues is mostly organizational: the authors chose to frame their work as three “studies,” presenting the “methods” for each first, followed by the results for each, etc. I found this structure to be confusing and hard to follow, as I had to jump back and forth between methods and results and discussion sections to remember what was done (and why) as I proceeded. At one point they discuss the topics modeling results, for example, but it had been so long since they had discussed the methods, and then they waited until the subsequent section to actually give the topics meaningful labels (which they never do in the main text for the 16 topics). It was very hard to keep track of everything because of this structure.

As the research itself really strikes me as one single study, not three, but with many analytic components, I think the authors should restructure the paper in a more logical, linear fashion. For example, they could still preview the whole set of big questions and their approach in the introduction, and then the main sections could be each question in turn, with meaningful headings/subheadings rather than traditional “Study 1” and “methods” headings.

So, they could still start by describing the procedures for gathering the data from twitter and the organization of the dataset. A sub-heading in that section could be something like “Themes in the data: Topic modeling” where they go through all of the methods, results, AND discussion and labels (each with their own subheadings…) for the topic modeling. Then they can move on to a section about defining their WAR (AND alternative!) dictionaries and analyses and discussion, and then conclude with their general discussion. I think something like this would help make the flow of the paper clearer and more effective.

Some additional comments:

While the authors reviewed a good amount of research on war metaphors, they neglected to discuss any of the dozens of articles have been written very recently about the war metaphor framing for Covid-19 (and its plusses and minuses), in both mainstream and independent outlets online. I think citing and discussing at least some of these would help situate the article in the present moment, provide additional context, and highlight the importance of the present research. Here are some examples:

https://grist.org/climate/no-more-war-on-coronavirus-in-search-of-better-ways-to-talk-about-a-pandemic/

https://www.vox.com/culture/2020/4/15/21193679/coronavirus-pandemic-war-metaphor-ecology-microbiome

https://time.com/5821430/history-war-language/

https://www.theguardian.com/commentisfree/2020/mar/21/donald-trump-boris-johnson-coronavirus

https://medium.com/@steve.howe_63053/were-at-war-the-language-of-covid-19-e3d4f4a1ae2e

https://www.counterpunch.org/2020/04/24/trump-is-not-a-wartime-president-and-covid-19-is-not-a-war/

https://www.afsc.org/blogs/news-and-commentary/how-to-talk-about-covid-19-pandemic

https://blogs.scientificamerican.com/observations/military-metaphors-distort-the-reality-of-covid-19/

https://theconversation.com/war-metaphors-used-for-covid-19-are-compelling-but-also-dangerous-135406

On Page 8, 167, the authors say “As explained in [1], war metaphors are pervasive in public discourse and span a wide range of topics because they provide a very effective structural framework for communicating and thinking about abstract and complex topics, notably BECAUSE of the emotional valence that these metaphors can convey” (emphasis added). This makes it sound like the emotional valence of WAR is part of its structural framework, but I think this is a bit confused. War provides both a structural schema as a source domain AND it conveys an emotional tone; these points are actually separated in the paper referenced in the sentence. The authors break this down on the following page, but this sentence was unclear. Again, this may be part of the broader need to edit and revise some of the language in the article.

P12, Line 243-4: “…and the targeted language [English] corresponds to the first language of many (if not most) US residents.” Look this up and cite a source instead of speculating.

P11-12. I was terribly confused by the whole data gathering and filtering procedure. It was unclear how many tweets there were vs. individual tweeters vs. used tweets. The table tracks cumulative tweets but didn’t say that, which was confusing. It was not explained how the filtering was done (i.e., how did you choose which tweet to keep from each user that posted multiple tweets? Did the same tweeters post on multiple days and how was that dealt with?). I think this whole section could be streamlined and made much clearer.

Lines 270-72: The authors note they expected to find broader and more generic topics when they included 4 as compared to 16 topics. Well, of course, how else could that have turned out? In general, I found the use of two sets of topics to be unnecessarily confusing and did not feel it added much to the overall message in the paper. I suggest the authors stick with one set of topics that have easily identifiable and meaningful labels/clusters of attributes. Perhaps they could split the difference and choose 8 or 10 topics. Whatever makes the most sense for interpreting the metaphor data later is fine. I should also note this was all very exploratory/arbitrary, which is OK, but perhaps should be noted in the text (they could add a footnote explaining that using different numbers of topics doesn’t fundamentally change the pattern of findings).

Lines 309-310: The authors write, “The term list includes the following 79 terms“… but no list was forthcoming yet until the authors discussed their other method for generating terms. Either separate out into two lists (79 + 12) or, better, use one list but BOLD the ones coming from tool two (metaNet), and do not say “the following terms…” until you are actually planning to list the terms.

The authors use FAMILY as their “literal” comparison, but it should be noted that family terms COULD be figurative (and indeed, Lakoff, for example, has written much about the figurative uses of FAMILY in describing governments…). For example, “all Americans are one family.” “the president is the father of the American household,” etc. Is there any way to check to make sure all of the instances of family terms in the dataset are indeed literal and not figurative (and to remove the latter)?

On lines 503-4, the authors note that war words have a “very negative valence, OF COURSE” [emphasis added]. But I am not so sure I agree with that. Some people might get excited and motivated by ‘FIGHTING” the virus (which feels much less negatively valenced than THREAT, for example). Especially in the United States, which comprises many subcultures that glorify guns and wars and the military, I think some of these terms may be quite positively valenced. Maybe draw on some empirical work and use actual ratings of emotional valence of these words (e.g., using Pennebaker’s LIWC or some other database)

Reviewer #3: This paper adopts a topic modelling approach to study a dataset consisting of just over 200,000 tweets about Covid-19 posted in English (and primarily from the USA) in March and April 2020. The approach is employed to: identify the main topics in the data (set at 4 and 16); study the prevalence of a WAR metaphorical framing; compare that framing with three alternative metaphorical framings and a literal topic; and investigate any correlations between the WAR framing and the topics that were automatically identified. The findings are relevant, if somewhat predictable: the WAR framing is more prevalent than the alternative metaphorical framings, and it tends to correlate with discussions of diagnosis and treatment.

Concerning the creation of the dataset, the authors provide some justification for limiting tweets from the same account to one. However, this makes it impossible to capture the actual prevalence of the various framings on Twitter. The consequences of this decision should therefore be explicitly acknowledged.

The labelling of the groups of terms associated with each automatically generated topic imposes more coherence on each set of words than is actually the case, especially in the version of the analysis that only involves four topics. This is typical of this kind of computational approach to discourse analysis, but it should minimally be pointed out as a methodological issue.

As for the alternative metaphorical frames, the terms under TSUNAMI are generally to do with natural disasters, rather than tsunamis specifically.

Finally, it should be acknowledged more explicitly that this kind of analysis cannot shed light on how the WAR framing, or any other framing, are actually used. For example, it cannot distinguish between cases where the WAR framing is adopted and where it is critiqued (as has also been the case on Twitter). Ideally, the subset of tweets that employ WAR-related vocabulary could have been subjected to a more fine-grained analysis, but this usually goes beyond the scope of studies such as this.

Reviewer #4: Thanks for the opportunity to review.

Interesting look at how discussion on Twitter may be framed using frames from the disease literature, and a brief discussion of results of topics models on a limited Twitter dataset. This is certainly a timely thing, so I recommend major revisions. With work I think this could bring value to the public health community as we endeavor to perform contact-tracing and subject to mis- and dis-information around this pandemic.

Main critiques

What I am missing is the theoretical and practical contribution. Specifically, how would the authors answer the "so what" if the tweets are framed like WAR, STORM, etc., and "so what" if they're not? (which, they're not - 90-95% of the posts are not according to the results.)

- Are relative frequencies of frames statistically different from each other, and do they happen often enough to be significant in general? Put another way, does this frame analysis work or matter on Twitter?

- How would the authors characterize the other 90% of the discussion, and why / how is it important? Are there any themes related to mis- or dis-information, or to political polarization?

Second, I have concerns about sampling bias. This amounts to a study of 12 days' worth of tweets, only a few thousand. Line 60 states 16k tweets are posted every hour (do the authors have a citation?), and yet the authors collected 25k tweets per day. This equation does not balance, even when accounting for a 1% sampling rate from the Twitter API. This uncertainty undermines the efficacy of this paper - either the collection has a problem or the statement is false.

Regardless, at the time of data collection multiple datasets of Twitter related to COVID-19 existed. I strongly recommend repeating this analysis in two ways to see if the results change or hold:

- one, now that the authors have been collecting more data for a while,

- and two, perhaps more pressingly, using one of the public open datasets for Twitter with millions of tweets. See e.g. this collection of resources: http://www.socialmediaforpublichealth.org/covid-19/resources/ "Twitter Data"

(This also suggests an opportunity to do temporal analysis, to see if the frames and discussion have changed and if so, how they are changing. This may help with a practical contribution - to answer if discussions are moving in a healthy or helpful direction, or the opposite, and why?

- For example, how often do these topics found happen over time, how often do these frames happen over time, and why is that important? How would we interpret these topics, and why might they be important? How do these frames correspond with hashtags or the literal discussion of the disease?)

Thirdly, please see critiques of the methods, related to LDA and Twitter pre-processing.

Fourthly, I also include more minor points and notes about statistical significance.

I also have concerning methods critiques that may undermine results:

On tweet processing decisions:

- I'm struggling to understand why the authors eliminated all but one tweet per user. This is a limitation. It looks like the methods and results are at the level of a tweet, not at the level of a user. In addition to the sampling bias, the authors could be discarding data that is important to their analysis. If the authors insist on retaining only one tweet per user, how this was performed? Was this random? If not, this could bias one's data again.

- I'm struggling to understand why the authors excluded retweets and mentions. How many retweets and mentions are there? Together, these choices severely limit the amount of analysis possible, to show how often the frame of the discourse is spreading, occurring, or changing. I understand wanting to exclude them initially, but what about repeating the analysis with them included to see how it changes?

On LDA implementation:

- Did the authors use Gibbs sampling or variational inference? Gibbs sampling has been shown to yield vastly superior topics. I'd recommend repeating analysis if used variational inference and see if results hold.

Related:

- how did the authors choose 4 vs 16 topics? why not other numbers? did they check perplexity - what number of topics has the lowest perplexity? (most likely to explain the data)

- How did the authors handle hashtags and URLs and usernames? These may contain information, or not, depending on the design of the study. What happened? These may be useful to report if analyzing the discussion.

On LDA interpretation and results:

- the authors look at significant words in topics, but what about tweets most about those topics? it can make a difference, per the coming citation. I recommend evaluating topics in both ways, as it may affect results of lines 585-596. See https://scholar.google.com/scholar?hl=en&as_sdt=0%2C21&q=reading+tea+leaves+humans+interpret+topic&btnG=

- did the authors interpret all of the 16 topics like they did the 4 topics? (lines 450-453)

- for topic figures, I suggest putting names of topics in the figure axes where possible. without them it's inconvenient to remember which is which

On Twitter pre-processing, I'm worried about the authors' use of general-language tools on tweets which have been shown to use vastly different language structure.

- stopwords from 2012... check/justify that these are up-to-date and apply to Twitter? need to come up with domain-specific ones?

- along the same lines there's a twitter tokenizer (e.g., stanfordNLP, NLTK) that are custom-built for this... what about emoji, how were these handled?

- line 279: better would have been to use tf-idf and leave the common terms in... these would have been reduced by the weighting organically

On literal framing control:

- What about the literal frame "it's a disease"? The authors chose family as a literal frame- this may strongly coincide with incidence or deaths from the disease, which may not be exactly what the authors want to measure.

- In addition... how are the authors controlling by including this? Should this be used for normalization, or testing statistical significance of frequencies or of differences among frame?

On Results and discussion

table 2 - are these results statistically significant? This would give weight to the authors' statement about the relative amount.

lines 512-536, about topics predicting occurrences of frames... are these differences between frequencies statistically significant? are these frequencies high enough to matter?

Continuing down the path about frames vs. topics:

- How often do the family or alternative frames show up in the predicted topics? like likes 383-384 for the WAR frame.

- In addition, how many topics include words in the frames? This may be an indicator if the frames are even worth studying on this domain. (see 90% number and earlier comment about frequencies)

Lines 55-67 do the authors have any citations for any/all of these statements?

6. PLOS authors have the option to publish the peer review history of their article (what does this mean?). If published, this will include your full peer review and any attached files.

Reviewer #1: Yes: Paul Thibodeau

Reviewer #2: No

Reviewer #3: No

Reviewer #4: No

---

## [Author Response · Author response to Decision Letter 0]

14 Jun 2020

Reply to reviewers: Framing COVID-19 How we conceptualize and discuss the pandemic on Twitter PONE-D-20-10986.

We are very thankful to all 5 reviewers for the constructive feedback and comments, which helped us improve our manuscript in many ways. We took onboard all comments and we hereby provide our reply to each and every point raised. We apologize for the length of this document, which matches the length of the actual article.

A note to all reviewers: The matching algorithm used to identify the target terms in the corpus has been improved (notably, it is not case-sensitive anymore, so it can retrieve for example occurrences of “fighting” as well as “FiGHting”: this is quite relevant because on social media capital letters and mixes of small and caps are often used). We have therefore updated the numbers in the “Framing results” with more accurate values (none of which are affecting the topic modeling, interpretation or discussion). Accordingly, Fig. 6 and Table 2 were updated as well.

Reviewer #1:

The paper analyzes Tweets that are used to talk about Covid-19 in three studies. The first identifies topics of these tweets. The second quantifies the prevalence of WAR metaphors in general, and across the topics identified in the first study. The third explores alternative figurative frames for the virus, along with one non-figurative frame (FAMILY). It is a timely study with interesting methods and results. I enjoyed reading it. I have suggestions for a revision, along with some methodological questions detailed below.

Reply: Thank you.

Introduction

1. The introduction could be better connected with the methods used in and research questions explored in the studies. Why is it interesting and important to determine what topics are discussed on Twitter in relation to Covid-19 or how frequently WAR and other metaphors are used? For example, could the topic modeling help to improve surveillance and forecasting about the disease? What frequency of WAR metaphors should we expect? How and why were the specific alternative frames chosen? Are there relevant theories in cognitive linguistics that the current work informs?

Reply: Thank you for this comment. The introduction contains the following information: topic (first paragraph), how we operationalize the topic (through tweets, 2nd paragraph), aim (3rd para) and RQs (4th para). The information required by the reviewer in our opinion does not belong in the introduction, but rather in the Theoretical background, which contains related work in cognitive linguistics, and Methods, where we explain why specific frames were used etc. Moreover, we have already included in the Introduction that we expect to find WAR related metaphors (indicating a very specific expected frequency of metaphors to be found appears to be out of place). We took this comment onboard by adding a very short paragraph at the end of the introduction in which we highlight the relevance of this study, which is then developed in the next section.

2. The emphasis and major strength of the paper seems to be the focus on WAR metaphors. The prevalence of WAR metaphors is quantified for an important issue, by topic, and in the context of other metaphoric frames (and a non-metaphoric frame). To highlight these strengths, I would encourage the authors to emphasize the novelty of quantifying the WAR frame using automated methods and real world data. Maybe switching the order of studies 1 and 2 would be helpful? (The results of Study 1 are difficult to interpret on their own).

Reply: Thank you. We emphasized these strengths in the last paragraph of the introduction. Moreover, following also the suggestion by reviewer 2, we have restructured the paper (methods and analyses), and thus the three studies are now three parts of the same study, so they don’t have to be interpreted by their own. This should improve the readability and solve this comment too.

3. The “Theoretical Background” section describes past work that is certainly interesting and relevant, but it doesn’t really identify theoretically motivated questions that the current work is well-suited to address. In most cases, it emphasizes practical applications and real world issues and/or methods (e.g., the relationship between measuring public sentiment on Twitter and addressing public health issues). Maybe rename this section for clarity.

Reply: Thank you. The Theoretical Background section contains related works (theoretical background) in the following fields, which we bring together in our study: quantitative analyses based on Twitter data related to epidemics (paragraphs 1-4); cognitive linguistic studies of figurative framings and metaphors, including the WAR frame and metaphors that we tackle in our study (paragraphs 5-7) and alternative framings (paragraph 8). This section does contain a selected review of the literature in these two fields of research, which is functional to our argumentation. We therefore stick to the label Theoretical Background.

Smaller points related to the introduction:

4. On p. 3 the authors note that around “16K Tweets are posted by Twitter users every hour, containing a hashtag such as #coronavirus, #Covid-19 or #COVID.” Is this based on the current data collection or a metric that is computed by Twitter? What is the temporal window for this statistic?

Reply: Thank you. We have made this relation clearer by stating that “collecting every tweet containing a hashtag such as #coronavirus, #Covid-19 or #Covid, we accumulated around 16,000 tweets within an hour each day.” Meaning, that the metric is in fact based on the current data collection. This ambiguity has now hopefully been resolved.

5. It seems like there is a typo on p. 4: “Unlike the articles on magazines and journals typically used for corpus analyses of this kind, Twitter does contain messages written by journalists and other experts in mass communication, as most tweets are provided by non-expert communicators.” Maybe “as” should be “although”?

Reply: Thank you. We solved this.

Methods

1. Please clarify the following questions:

a. How were the 25k tweets per day selected? It seems like the algorithm picked up the first 25k tweets that included one of the specified hashtags per day, but that’s not explicitly stated. How long did it typically take to get to 25k tweets?

b. Were retweets included?

c. For a given person who tweeted multiple times in a day about the virus, was it only their first tweet that was included? What was the filtering criteria/method?

d. Were Tweets in languages other than English included?

Reply: 1a) The algorithm sends a query to the Twitter API. Our query asks for 25,000 tweets from one day that contain a list of specified hashtags (#CoronaVirus, #Covid etc.) and are of English language. Twitter restricts the access by firstly not allowing to choose a time of day in the query, therefore we receive the first 25k tweets on that day. Secondly, there is a rate limit to retrieve tweets. To abide by that rate limit, the algorithm has waiting periods. On average it takes about 2hrs and 45min to collect 25k tweets from one day.

Reply: 1b) Retweets were not included (as we explained in the paper).

Reply: 1c) Yes, only the first tweet about the pandemic was retained, for each user. In this way we constructed a balanced corpus that had one tweet per user. By doing so, we avoided having, for example, hundreds of tweets from the same user who, maybe, was particularly keen on using the WAR frame. This would have biased our results. Instead, by keeping 1 tweet per user we have a better overview of how people talk about covid on twitter. The algorithm picked the first tweet per user that it could find, within the timeframe indicated. Collecting all tweets from all users, and then randomly selecting one per each user would have been computationally cumbersome and would have exceeded our database capacity. 

Reply: 1d) see a), we have only selected English tweets through Twitter's API language filter.

2. It should be possible to clarify some of the hedging in the sentence, “our corpus arguably encompasses mainly tweets produced by users residing in the USA, where the time of data collection corresponds to awake hours, and the targeted language corresponds to the first language of many (if not most) US residents” (p. 12). I realize that the location data was stripped from the text before it was stored, but the text is initially tagged with the location data, so maybe it is possible to estimate the percentage of tweets from the US. Language use questions are asked on the US census every 10 years and roughly 80% of US residents report speaking English at home. A citation would make the case stronger.

Reply: In fact, a preliminary analysis of tweets included a statistical analysis of the location data. To the best of our knowledge, the location data provided by the free Twitter API is self-reported by the user. Unfortunately, as the data included a great range of responses from “Anywhere that has liquor” or “Here.” to “México!!!” or “Universe”, we found no feasible way as to manually interpret the sample. We concluded that most of the locations indicated the US. Furthermore, the tweet time and targeted language seemed to be a more reliable estimator to conclude the majority of tweets being from US citizens. We have now added a citation, supporting the claim that most US residents speak English at home.

3. How were the topic numbers (n = 4 vs. 16) decided in the first study? I find the results of this study hard to interpret. I think they are more informative when presented in concert with the results of study 2.

Reply: Thank you. We have added a sentence explaining that the numbers 4 and 16 were chosen on an empirical basis and then backed up by an analysis of the internal coherence of the clusters. We were interested in a small but meaningful number of clusters and opted for 4, because 1-2-3 were in our opinion too limited; then we were also interested in a much more granular solution, so we opted for 16 (4 x 4).

Reply: We have now reported the coherence scores for these clusters for different LDA inference algorithms (as suggested by Reviewer 4). A more detailed discussion of these scores, which is linked to the decision to choose topics 4 and 16 is given in response to Reviewer 4. In short, the methods used to evaluate the internal coherence of the various cluster solutions from 1 cluster (1 topic) to 20 clusters (20 topics), shows that the coherence is particularly good for 4 clusters (according to the elbow method of cluster coherence evaluation) and that the model plateaus at 16 topics, with little interpretable improvement to be expected for more than 16 topics.

Reply: Please note that the structure of the 3 studies, as indicated above, has now changed, and the three studies are 3 phases of one big study. This should improve the interpretability of the results.

4. Categorizing language as metaphorical or not is tricky. There is a fair amount of work on this and the most common approach uses expert coders (see, e.g., Steen et al, 2010, A Method for Linguistic Metaphor Identification). I think the approach taken in the paper is reasonable but I think some limitations should be acknowledged. For example, there would probably be some disagreement by coders over which conceptual metaphors individual instances of “fight” appeal to (war vs. boxing vs. games, etc) and even whether or not particular instances are metaphorical or not.

Reply: Thank you. The MIP (and MIPVU) approaches for metaphor identification in texts (Steen et al. 2010) works around basically the distinction between contextual meanings and basic meanings. If there is a cross-domain comparison between contextual and basic meanings in a given text, in relation to a specific lexical unit, then that specific unit shall be marked as MRW (metaphor related word). The procedure is performed by manual annotations, using dictionaries as tools to check the contextual and basic meanings. Because the dictionaries are the same, it is (rightly) argued that different annotators shall converge to similar annotations of MRWs, and hence the procedure is reliable. In our case, we took a different approach. Firstly, our corpus is much larger than any corpus ever analyzed with the manual annotation MIP. Secondly, there are probably no good dictionaries that list among the contextual meanings of a word, the covid-related uses. It could, however, be argued that we kept somehow the distinction between contextual and basic meanings, in the following way: by selecting tweets that contain the hashtag COVID we set the contextual meanings of the words in the tweet as the pandemic-related domain. Thus, in that domain, any word that has a basic meaning related to WAR, such as all the words used in our lists, retrieved from Lakoff’s database and the web-service related-words (both solidly theoretically motivated) are used metaphorically in the context of covid.

5. Include some discussion about the relationship between the STORM and TSUNAMI categories. At first glance it seems like all instances of TSUNAMI should also be instances of STORM, but the current approach establishes these categories as mostly (completely?) non-overlapping.

Reply: Thank you. The overlap between the two wordlists is about 20%. It should be noted that the two frames STORM and TSUNAMI, relate to quite different concepts. On the Macmillian dictionary: STORM: an occasion when a lot of rain falls very quickly, often with very strong winds or thunder and lightning. TSUNAMI: a very large wave or series of waves caused when something such as an earthquake moves a large quantity of water in the sea. It makes sense that, denoting different natural phenomena, the two frames have quite different sets of related words.

Reply: In general, while deciding for alternative frames, and related word lists, we discarded for example the potential alternative frame FLOOD precisely because it had too much overlap with these two frames, STORM and TSUNAMI. Conversely, STORM and TSUNAMI are substantially different, and therefore both good candidates for alternative frames. Moreover, both these frames, STORM and TSUNAMI, have been indicated as prolific frames that are actually used in the discourse on Covid, by the Reframe Covid initiative (#ReframeCovid: https://docs.google.com/spreadsheets/d/1TZqICUdE2CvKqZrN67LcmKspY51Kug7aU8oGvK5WEbA/edit#gid=268174477). This has been now added to the description of the methods related to the Alternative Frames analyses.

6. I like the comparison of the metaphorical frames to the FAMILY non-metaphorical frame, but the FAMILY frame also seems fairly different in that it is more of a topic than a frame (i.e. in some ways more akin to the topics identified in Study 1). I don’t think this needs to be changed, but it seems worthy of a little discussion.

Reply: Thank you. In traditional framing theory, framing is defined as “select[ing] some aspects of a perceived reality and mak[ing] them more salient in a communicating text, in such a way as to promote a particular problem definition, causal interpretation, moral evaluation, and/or treatment recommendation for the item described” (Entman, 1993,p. 53).

Reply: FAMILY is a frame that, as another reviewer pointed out, can be used to talk about covid because it is an aspect of this reality. It can be used figuratively, as well as literally. Another reviewer mentioned: “family terms COULD be figurative (and indeed, Lakoff, for example, has written much about the figurative uses of FAMILY in describing governments” (for example, within the EU, one could say that Germany is the responsible father of the family, who scolded Italy for her behavior etc). We have chosen FAMILY deliberately to be a frame that is most likely not used figuratively, yet has comparable properties to a frame we expect to be used figuratively (e.g. WAR, STORM or MONSTER). In fact, we have checked qualitatively the tweets that feature one of the FAMILY terms in it, and it appears that they are used literally, not metaphorically. So, for the covid reality, FAMILY is used typically as a literal frame, which carves an aspect of the overall covid reality (the aspect related to family relations and family dynamics and family members).

Reply: Moreover, in communication sciences a frame is typically defined as consisting of two elements (Joris,d’Haenens, & Van Gorp, 2014, p. 609): framing devices which are elements in a text and specific linguistic structures (for example a list of words related to a frame) and reasoning devices which are the (latent) information in a text through which the problem, cause, evaluation, and/or treatment is implied (a more conceptual-communicative dimension of a text). A topic operationalized by topic modelling, corresponds to the first of these two components of a frame (i.e., a list of semantically related words). For example, one can say that sister, father, home, parenthood, brotherly love are all words that together represent the topic of FAMILY, which can be used as a frame in covid discourse: by talking about family aspects one could direct the readers’ attention to the importance of family relations and stress how these may have been disrupted by the pandemic. But a topic can also be used as a frame, if it has the communicative purpose to highlight a specific aspect of a situation to fulfil a specific communicative purpose.

Reply: We have added the definition of framing in the theoretical background, explaining that the FAMILY frame is used literally in this type of discourse (end of Theoretical Background section). We also added a paragraph on the relation between frames and topics (as intended in topic modelling) in the beginning of the section “Identifying topics in Covid-19 discourse on Twitter through Topic Modeling”.

7. Include a note on how comparisons will be made. No inferential statistics are presented, which is common in cognitive linguistics, but the question about whether the number of cases is meaningfully different by frame or topic type will likely arise for readers.

Reply: Thank you. We have now reported the results of a Cochran’s Q test to assess if there are any significant differences between the 5 frames, in the ways they are represented in the corpus of tweets. To do so, we compiled a tweets by frames matrix that displays binary values for presence or absence of a frame-related term in each tweet. Cochran’s Q test statistic = 47,226.72 , df = 4, p < 0.001. The post-hoc pairwise McNemar test is highly significant (p < 0.001) for all permutations (all frame pairs). We can therefore conclude that there are significant differences in the presence of target words between each pair of frames.

Reply: Additionally, we have followed a suggestion by Reviewer #4 and validated our results through a series of replication studies: we have repeated this analysis on a new corpus constructed using another two weeks of tweets; on the overall corpus of tweets encompassing 8 weeks worth of tweets, and on an external, already existing resource containing Twitter data. The results of the replication studies are reported in the section “Replication studies”. Results show that the findings are consistent across the various datasets.

Results

1. It’s hard to interpret the results of Study 1. Are these topics the ones we would expect? Do they inform theory? Do they inform practice? Could they have come out any other way?

Reply: Thank you. As a matter of fact, we have now restructured the paper so that these three studies are three parts of the same study, which is discussed at the end in terms of their relevance for metaphor studies and for communication sciences in general, about how pandemics are understood and discussed by the general public.

2. What inferences can we draw about the relationship between WAR metaphors and topics from the results of Study 2?

Reply: Thank you. Please note that the paper has been restructured. The inferences about the relationship between war metaphors and topics are already elaborated in the section LDA Topic prediction of WAR tweets and discussed in the WAR framing discussion section, from which we report hereby an excerpt: “In relation to the topic modelling of the war-related tweets, we showed that tweets that feature war-related terms are most likely to belong to topics IV, I and III, rather than to topic II. Interestingly, topic IV addresses aspects related to the reactions to the epidemics, including the measures proposed by the governments and taken by the people, such as self-isolating, staying at home, protecting our bodies and so forth. Our analysis therefore suggests that using war-related words is a communicative phenomenon that we use to express aspects of the Covid-19 epidemic related to the measures needed to oppose (fight!) the virus. Moreover, tweets that feature war-related words are also often classified within the topics I and III, which include the aspects related to communications and reports about the virus, and politics. We interpret these results arguing that public communications and political messages are likely to frame the discourse in the WAR framing. Finally, it might not come as a surprise the fact that topic II, which encompasses aspects of the discourse related to the familiar sphere, the community and the social compassion, does not relate well with the tweets containing war terms.”

Reply: Moreover, as mentioned in reply to a previous comment, we have now clarified the definitions and distinctions between frame and topic.

3. If space is an issue, I think the comparison of the 30 vs 50 terms approaches on pp 26-28 could be cut.

Reply: Thank you.

4. Small point: There is no General Discussion section, although it is alluded to on p. 24.

Reply: Thank you. We apologize. We changed the last section (which was General Discussion and Conclusions) into simply Conclusions because of space reasons and forgot to update this sentence in the text. Thank you for pointing this out. We corrected it.

5. Small point: Include a citation (and, ideally, a more precise statistic) in the sentence “…as previous literature shows that metaphor-related words cover only a percentage of the discourse, and that literal language is still prevalent” (p. 28). What percentage? Are there other metaphors that might be prevalent?

Reply: Thank you. We clarified this point and related it to corpus-based literature that supports this claim. In particular, we refer to the analysis performed by Steen and colleagues, who report that literal language covers 86.4% of the lexical units, while metaphor-related words cover just 13.6% of the lexical units. Their analysis is based on a corpus of 187,570 lexical units, manually annotated one by one for metaphoricity in a formal content analysis.

6. It would be nice to ground some of the qualitative observations noted in the discussion. For example: “Words in the STORM and TSUNAMI frames seem to relate to events and actions associated with the arrival and spreading of the pandemic…” (p. 28).

Reply: Thank you. We added examples to ground our arguments.

7. The paper ends by introducing the idea of a Metaphor Menu, which is interesting but it doesn’t logically fall out of the current study in my opinion. Maybe this idea could be discussed a little more.

Reply: Thank you. As we explained with our analysis, the WAR frame appears to be used in relation to specific topics, and thus to frame specific aspects of the discourse around Covid. Metaphors related to war, for example, do not appear to be apt, and therefore used, to frame the covid topics “ such as the need to feel our family close to us, while respecting the social distancing measures, or the collaborative efforts that we should undertake in order to #flattenthecurve, that is, diluting the spreading of the virus over a longer period of time, so that hospitals’ ICU departments can work efficiently without getting saturated by incoming patients.” ( excerpt taken from our Conclusions). Thus, it follows that there are aspects involved in the discourse around covid that may benefit from different figurative frames. In this sense, the war frame and war metaphors alone appear to be not sufficient to discuss all the aspects related to covid. The idea of a metaphor menu, in our argumentation, evolves from this observation, to provide communicative tools that can be used to discuss also the other topics involved in the covid discourse, where war metaphors are not observed.

Reviewer #2

The authors examined the (metaphorical) content of tens of thousands of English tweets surrounding the Covid-19 pandemic, scraped from two recent weeks from largely American twitter users. Topic modeling revealed several common themes (4 and also 16; more on this below), and that war metaphors were somewhat common (~4% prevalence), for some topics more than others, and appeared more frequently than other metaphorical domains. They argue that this is consistent with other empirical and theoretical research on the use of war metaphors in public discourse, but they now provide evidence this extends to everyday lay discourse online.

In general, I thought this was a timely article dealing with an important topic of interest to a variety of scholars, and a nice extension of previous work and theoretical musings on the use and prevalence of war metaphors. I think the methods and analyses were thoughtful and for the most part sound (though, as I detail below, I was confused about some of the details), and the results were solid. That said, I have a variety of comments and concerns that I think the authors should address in a revision before the manuscript is considered for publication.

Reply: Thank you.

One overarching concern is that the paper feels like it was rushed to submission and therefore the writing and overall organization are not quite up to the standards of a publishable manuscript. I understand the authors’ sense of urgency in getting this paper out there while the global pandemic crisis is still at its peak, and that they literally wrote it over the past few weeks, but I think extra care needs to be taken during any revision process to make sure the writing is improved. There were many grammatical and punctuation errors throughout the paper, along with confusing sentences and shifts in tense (if they want to refer to “now,” they should stick with phrasing like “at the time of writing,” which they were not consistent with throughout the paper). At times it was difficult to follow the logic of their thinking or make sense of some of the details of the methods and analyses.

Reply: Thank you. We have now worked on the revised manuscript to improve its readability in various ways: 1) we have restructured the sections (see comment below), 2) we have revised the tenses and 3) we have proofread the manuscript

One of the issues is mostly organizational: the authors chose to frame their work as three “studies,” presenting the “methods” for each first, followed by the results for each, etc. I found this structure to be confusing and hard to follow, as I had to jump back and forth between methods and results and discussion sections to remember what was done (and why) as I proceeded. At one point they discuss the topics modeling results, for example, but it had been so long since they had discussed the methods, and then they waited until the subsequent section to actually give the topics meaningful labels (which they never do in the main text for the 16 topics). It was very hard to keep track of everything because of this structure.

As the research itself really strikes me as one single study, not three, but with many analytic components, I think the authors should restructure the paper in a more logical, linear fashion. For example, they could still preview the whole set of big questions and their approach in the introduction, and then the main sections could be each question in turn, with meaningful headings/subheadings rather than traditional “Study 1” and “methods” headings.

So, they could still start by describing the procedures for gathering the data from twitter and the organization of the dataset. A sub-heading in that section could be something like “Themes in the data: Topic modeling” where they go through all of the methods, results, AND discussion and labels (each with their own subheadings…) for the topic modeling. Then they can move on to a section about defining their WAR (AND alternative!) dictionaries and analyses and discussion, and then conclude with their general discussion. I think something like this would help make the flow of the paper clearer and more effective.

Reply: Thank you. The organization of the empirical part of our paper was presented as follows:

Methods

 General design of the 3 studies

 Constructing the corpus of Covid-19 tweets

 Study 1: Identifying topics in Covid-19 discourse on Twitter through Topic Modeling

 Study 2: Determining lexical units associated with the WAR frame

 Study 3: Search method for alternative framings and relevant lexical units therein

 The literal frame of FAMILY used as control

Results and Discussions

 General Corpus Analytics

 Study 1: Topic Model Analysis

 Analysis of 4 Topics

 Analysis of 16 Topics

 Study 1: Topic Modelling Discussion

 Study 2: WAR Framing Results

 LDA Topic prediction of WAR tweets

 Study 2: WAR Framing Discussion

 Study 3: Alternative Framing Results

 Study 3: Alternative Framing Discussion

 Conclusion.

Reply: Indeed, we provided in a clear and structured way the overall set of methods, and then the section with all results and discussions. The reviewer asks to restructure the paper as one big study with several analytic components. Following the reviewer’s suggestions we have restructured the paper as follows:

Experimental design

 Constructing the corpus of Covid-19 tweets

 General Corpus Analytics

What type of topics are discussed on Twitter, in relation to Covid-19?

 Identifying topics in Covid-19 discourse on Twitter through Topic Modeling

 Topic Model Analysis

 Discussion

To what extent is the WAR figurative frame used to talk about Covid-19 on Twitter?

 Determining lexical units associated with the WAR frame

 WAR Framing Results

 LDA Topic prediction of WAR tweets

 Discussion

Are there alternative figurative frames used to talk about Covid-19 on Twitter?

 Search method for alternative framings and relevant lexical units therein

 The literal frame of FAMILY used as control

 Alternative Framing Results

 Replication studies

 Discussion

General Discussion and Conclusion.

Some additional comments:

While the authors reviewed a good amount of research on war metaphors, they neglected to discuss any of the dozens of articles have been written very recently about the war metaphor framing for Covid-19 (and its plusses and minuses), in both mainstream and independent outlets online. I think citing and discussing at least some of these would help situate the article in the present moment, provide additional context, and highlight the importance of the present research. Here are some examples:

https://grist.org/climate/no-more-war-on-coronavirus-in-search-of-better-ways-to-talk-about-a-pandemic/

https://www.vox.com/culture/2020/4/15/21193679/coronavirus-pandemic-war-metaphor-ecology-microbiome

https://time.com/5821430/history-war-language/

https://www.theguardian.com/commentisfree/2020/mar/21/donald-trump-boris-johnson-coronavirus

https://medium.com/@steve.howe_63053/were-at-war-the-language-of-covid-19-e3d4f4a1ae2e

https://www.counterpunch.org/2020/04/24/trump-is-not-a-wartime-president-and-covid-19-is-not-a-war/

https://www.afsc.org/blogs/news-and-commentary/how-to-talk-about-covid-19-pandemic

https://blogs.scientificamerican.com/observations/military-metaphors-distort-the-reality-of-covid-19/

https://theconversation.com/war-metaphors-used-for-covid-19-are-compelling-but-also-dangerous-135406

Reply: Thank you. All these blog posts and articles are non-academic, addressed to the large public, and most of them have been published online after our submission to PlosOne. These are the reasons why they were not included in the discussion. In fact, there has been an exponential growth of non-academic posts and articles for the large public on the language of covid, especially in relation to the recent #ReframeCovid initiative, which however gained momentum after we submitted the current paper. We gladly related to some of these more recent press materials in our academic paper (although we have already mentioned the rise of the ReframeCovid initiative), for the purpose of framing our article within the current debate, as the reviewer mentions. We added a long paragraph on this matter toward the end of the Theoretical Background section.

On Page 8, 167, the authors say “As explained in [1], war metaphors are pervasive in public discourse and span a wide range of topics because they provide a very effective structural framework for communicating and thinking about abstract and complex topics, notably BECAUSE of the emotional valence that these metaphors can convey” (emphasis added). This makes it sound like the emotional valence of WAR is part of its structural framework, but I think this is a bit confused. War provides both a structural schema as a source domain AND it conveys an emotional tone; these points are actually separated in the paper referenced in the sentence. The authors break this down on the following page, but this sentence was unclear. Again, this may be part of the broader need to edit and revise some of the language in the article.

Reply: Thank you, we rephrased this sentence as the reviewer suggested (as well as others throughout the manuscript, to improve readability).

P12, Line 243-4: “…and the targeted language [English] corresponds to the first language of many (if not most) US residents.” Look this up and cite a source instead of speculating.

Reply: Thank you. Corrected. We cited the American Community Survey (ACS), which reports this information.

P11-12. I was terribly confused by the whole data gathering and filtering procedure. It was unclear how many tweets there were vs. individual tweeters vs. used tweets. The table tracks cumulative tweets but didn’t say that, which was confusing. It was not explained how the filtering was done (i.e., how did you choose which tweet to keep from each user that posted multiple tweets? Did the same tweeters post on multiple days and how was that dealt with?). I think this whole section could be streamlined and made much clearer.

Reply: Thank you. We have edited the paragraph including an additional discussion about the individual tweeter filtering and improved the description of the table. We have simplified the interpretation by explaining that all 203,756 tweets are from individual tweeters, as each day tweets are filtered against the list of previous tweeters.

Lines 270-72: The authors note they expected to find broader and more generic topics when they included 4 as compared to 16 topics. Well, of course, how else could that have turned out? In general, I found the use of two sets of topics to be unnecessarily confusing and did not feel it added much to the overall message in the paper. I suggest the authors stick with one set of topics that have easily identifiable and meaningful labels/clusters of attributes. Perhaps they could split the difference and choose 8 or 10 topics. Whatever makes the most sense for interpreting the metaphor data later is fine. I should also note this was all very exploratory/arbitrary, which is OK, but perhaps should be noted in the text (they could add a footnote explaining that using different numbers of topics doesn’t fundamentally change the pattern of findings).

Reply: Thank you. We looked at a more compressed analysis that includes just 4 topics, and at a fine-grained analysis that encompasses 16 topics and we were interested in observing what type of information is captured by these topics. The choice for these two numbers specifically is motivated by theoretical as well as technical reasons that we have now explained (as requested by reviewer #4). Considering the topic modelling as a clustering task, the 4 ways and 16 ways solutions appear to be coherent cluster solutions. From a theoretical perspective, we were interested in exploring whether the more fine-grained solution (which delivers many smaller and more internally coherent clusters) provided intra-class distinctions and semantic specifications that could not emerge when looking at the condensed solution (which delivers a few large and less internally coherent clusters). In fact, we provided some qualitative observations in this respect, in the discussion of these analyses. It should be noted that labelling topics in LDA is common practice, observed for example in Lazard et al, and in Tran and Lee, and in Miller and colleagues (all these works are reviewed in the Theoretical Background section of our paper). Finally, again in the section Discussion of the Topic modelling analysis, we provide a discussion of the 16-way topic analysis and describe which of these appear to be related to the use of the WAR figurative frame. Therefore, and in line with the feedback received from the other reviewers, we are going to keep both the 4-way and the 16-way analyses.

Lines 309-310: The authors write, “The term list includes the following 79 terms“… but no list was forthcoming yet until the authors discussed their other method for generating terms. Either separate out into two lists (79 + 12) or, better, use one list but BOLD the ones coming from tool two (metaNet), and do not say “the following terms…” until you are actually planning to list the terms.

Reply: Thank you. We fixed this wording. However, we have not bolded the terms coming from metaNet or separated the two lists because it did not add value to our argumentation, and because some of the words appeared in both resources and therefore attributing them only to the first resource would have been misleading.

The authors use FAMILY as their “literal” comparison, but it should be noted that family terms COULD be figurative (and indeed, Lakoff, for example, has written much about the figurative uses of FAMILY in describing governments…). For example, “all Americans are one family.” “the president is the father of the American household,” etc. Is there any way to check to make sure all of the instances of family terms in the dataset are indeed literal and not figurative (and to remove the latter)?

Reply: Thank you. We manually checked the data in our corpus and indeed, as we expected, here the FAMILY related words that are used to construct the family frame are used in their literal sense. It makes sense because people on Twitter mention their personal situations in relation to Covid-19, which involves their family members, family-dynamics affected by the pandemic and the lock down and so on. So, family-related words are used literally. We have also mentioned that indeed this frame can be used metaphorically in other contexts (e.g., to talk about nations and governments, as Lakoff extensively demonstrated), at the end of the Theoretical Background section.

On lines 503-4, the authors note that war words have a “very negative valence, OF COURSE” [emphasis added]. But I am not so sure I agree with that. Some people might get excited and motivated by ‘FIGHTING” the virus (which feels much less negatively valenced than THREAT, for example). Especially in the United States, which comprises many subcultures that glorify guns and wars and the military, I think some of these terms may be quite positively valenced. Maybe draw on some empirical work and use actual ratings of emotional valence of these words (e.g., using Pennebaker’s LIWC or some other database)

Reply: We are grateful for the remark, nonetheless we can find our claim confirmed using emotional valence ratings. Analyzing our WAR term list using Pennebaker’s LIWC (for Social Media, including Twitter) results in a score of 28.9 for negative emotions on our terms, where the average texts have a score of 2.10 (the higher the score, the more negative the emotions). Our terms have a positive emotion score of 1.1, where the average texts have a score of 4.57. This indicates that war words have indeed a very negative valence.

Reviewer #3

This paper adopts a topic modelling approach to study a dataset consisting of just over 200,000 tweets about Covid-19 posted in English (and primarily from the USA) in March and April 2020. The approach is employed to: identify the main topics in the data (set at 4 and 16); study the prevalence of a WAR metaphorical framing; compare that framing with three alternative metaphorical framings and a literal topic; and investigate any correlations between the WAR framing and the topics that were automatically identified. The findings are relevant, if somewhat predictable: the WAR framing is more prevalent than the alternative metaphorical framings, and it tends to correlate with discussions of diagnosis and treatment.

Reply: Thank you. Although the reviewer mentions that the results may appear to be somewhat predictable, there was no quantitative empirical evidence published in support of this intuition. Therefore, our paper provides empirical support for this intuition (in addition to several innovative aspects, such as the relation between the WAR related terms and specific topics of this pandemic).

Concerning the creation of the dataset, the authors provide some justification for limiting tweets from the same account to one. However, this makes it impossible to capture the actual prevalence of the various framings on Twitter. The consequences of this decision should therefore be explicitly acknowledged.

Reply: Thank you. We limited the tweets to one tweet per user, to construct our balanced and representative corpus, precisely because we wanted to explore to what extent Twitter users use the various framings. By doing so we limited the bias of having for instance many tweets by the same user who, arguably, uses (or does not use) a specific frame because it is a frame that he/she finds apt and appropriate, or because he/she particularly likes it. For example, imagine that one specific Twitter user is very fond of sci-fi and monsters-related issues, he/she might use the MONSTER framing very frequently in their tweets, also about Covid-19. If we kept all the tweets by this user, this might have biased the distribution frequency of the MONSTER related words. Instead, by limiting one tweet per user, and dropping the retweets, we kept our corpus relatively manageable from a technical perspective (not too heavy) as well as representative and balanced from a theoretical perspective. In other words, we are not interested in the “actual prevalence” (the reviewer means the absolute prevalence?) of the various frames on Twitter, but the relative coverage of the different framings used to talk about Covid on Twitter. We have explicitly acknowledged these matters in the section “Constructing the corpus of Covid-19 tweets”.

The labelling of the groups of terms associated with each automatically generated topic imposes more coherence on each set of words than is actually the case, especially in the version of the analysis that only involves four topics. This is typical of this kind of computational approach to discourse analysis, but it should minimally be pointed out as a methodological issue.

Reply: Thank you. In topic modelling this is common practice. The clusters of words returned by the topic model are unlabeled, and are typically interpreted and labelled by the analysts (as in much of the literature mentioned in the Theoretical Background section: in Lazard et al, and in Tran and Lee, and in Miller and colleagues.) We have now explained, when introducing topic modelling, that this is the case, in the beginning of the section “Identifying topics in Covid-19 discourse on Twitter through Topic Modeling”.

As for the alternative metaphorical frames, the terms under TSUNAMI are generally to do with natural disasters, rather than tsunamis specifically.

Reply: Thank you. The word lists were constructed using two tools widely used and acknowledged in the scientific literature: the repository of metaphors and frames released by Lakoff and colleagues (hosted at Berkley) and the RelatedWords web service. We have extensively explained both in the manuscript. The specific way in which word lists were drafted is clearly and transparently explained in the manuscript, and backed up by solid scientific literature, not by personal intuitions.

Finally, it should be acknowledged more explicitly that this kind of analysis cannot shed light on how the WAR framing, or any other framing, are actually used. For example, it cannot distinguish between cases where the WAR framing is adopted and where it is critiqued (as has also been the case on Twitter). Ideally, the subset of tweets that employ WAR-related vocabulary could have been subjected to a more fine-grained analysis, but this usually goes beyond the scope of studies such as this.

Reply: Thank you. Indeed, as the reviewer concludes, a qualitative analysis of the subset of tweets that use the war-related words goes beyond the scope of this study. As a matter of fact, here we are talking about 9,502 tweets. We speculate that, because the WAR frame is particularly conventionalized, most of the WAR related words are used subconsciously and therefore adopted, rather than deliberately and consciously opposed and critiqued. But this intuition implies a different set of research questions, empirical analyses and hypotheses, as also the reviewer points out.

Reviewer #4

Thanks for the opportunity to review.

Interesting look at how discussion on Twitter may be framed using frames from the disease literature, and a brief discussion of results of topics models on a limited Twitter dataset. This is certainly a timely thing, so I recommend major revisions. With work I think this could bring value to the public health community as we endeavor to perform contact-tracing and subject to mis- and dis-information around this pandemic.

Reply: Thank you.

Main critiques

What I am missing is the theoretical and practical contribution. Specifically, how would the authors answer the "so what" if the tweets are framed like WAR, STORM, etc., and "so what" if they're not? (which, they're not - 90-95% of the posts are not according to the results.)

- Are relative frequencies of frames statistically different from each other, and do they happen often enough to be significant in general? Put another way, does this frame analysis work or matter on Twitter?

- How would the authors characterize the other 90% of the discussion, and why / how is it important? Are there any themes related to mis- or dis-information, or to political polarization?

Reply: Thank you. We made explicit our goal at the end of the Introduction: “This paper reports and discusses a series of corpus-based quantitative analyses on the figurative framings used in pandemic-related discourse. In particular, the WAR frame, which previous studies have identified as pervasive in many crisis-related texts, is investigated by means of automated methods (topic modelling) applied to real-world data. By answering the research questions outlined above we claim that our data can help to improve surveillance and forecasting about the disease, as previous research on pandemics has shown (see next section).”

Reply: Thank you. We have now reported the results of a Cochran’s Q test to assess if there are any significant differences between the 5 frames, in the way they are represented in the corpus of tweets. To do so, we compiled a tweets-by-frames matrix that displays binary values for presence or absence of a frame-related term in each tweet. Cochran’s Q test statistic = 47,226.72 , df=4, p < 0.001. The post-hoc pairwise McNemar test is highly significant (p < 0.001) for all permutations (all frame pairs). We can therefore conclude that there are significant differences in the presence of target words between each pair of frames.

Reply: Additionally, we have replicated our analyses on an updated corpus of tweets collected over two months with the same criteria that we used to collect the original corpus, and we replicated our analysis on a new corpus, extracted from an existing dataset “Coronavirus Tweets Dataset” (Rabindra Lamsal. (2020). Coronavirus (COVID-19) Tweets Dataset. IEEE Dataport. http://dx.doi.org/10.21227/781w-ef42). We could replicate similar proportions across the same timeframe in the alternative dataset and in our updated corpus. We have included these results in the section “Replication studies”. The distributions of the frames is very similar across the three resources, with the literal (control) frame being the most frequent, the WAR frame being the most frequently used frame, among the various metaphorical frames considered, followed by the STORM, the TSUNAMI and finally the MONSTER frame.

Reply: The rest of the covid tweets on Twitter are characterized by other frames, which can be literal or metaphorical. Arguably, they are mostly literal, because the WAR frame has been indicated as a pervasive metaphorical frame to talk about diseases. Our analysis is the first contribution that tests this qualitative claim, by running an analysis on an extensive corpus of tweets, about covid. This is new quantitative information, in cognitive linguistics and metaphor studies, which wasn’t available before. As we now mentioned in the paper, in the Discussion to the alternative frames section, Steen and colleagues, for example, reported that literal language covers 86.4% of the lexical units in a corpus of 187,570 lexical units analyzed by them (a subcorpus of the BNC), while metaphor-related words cover just 13.6% of the lexical units. Their analysis encompasses all parts of speech, including for example prepositions that are very commonly used metaphorically (e.g., IN, whenever referred to a non-spatial relation). Assuming that this percentage can be applied to any text (around 13% of words used metaphorically, the rest literally) our findings provide an argument for the actual pervasiveness of the WAR frame, which covers more than one third of ALL the metaphorical language used in texts.

Reply: Disinformation and political polarization are themes that are not related to the scope of this paper.

Second, I have concerns about sampling bias. This amounts to a study of 12 days' worth of tweets, only a few thousand. Line 60 states 16k tweets are posted every hour (do the authors have a citation?), and yet the authors collected 25k tweets per day. This equation does not balance, even when accounting for a 1% sampling rate from the Twitter API. This uncertainty undermines the efficacy of this paper - either the collection has a problem or the statement is false.

Reply: We could not collect every tweet that relates to the covid-19 discourse on every day, since those data sets that did, have collected between 0.5 and 3.5 million tweets on a single day (depending on different keywords), quickly accumulating unmanageable amounts of data exceeding the scope of our technical limits. “16k tweets are posted every hour (do the authors have a citation?), and yet the authors collected 25k tweets per day. This equation does not balance” should be elaborated further: We have queried 25k tweets on each day. This query collects every tweet with the related keywords in chronological appearance and stops when 25k of tweets have been reached. As the collection finished after about 1.5hrs, we can infer that around 16k tweets are posted every hour (relating to the covid19 pandemic). We have rewritten this section to make this clear.

Reply: Regarding the sampling bias, we can now refer to the results from our updated corpus and the alternative existing corpus (see next response). Notably, another great influence on the amount of tweets in our sample are the restrictions (only individual tweeters, etc.) which we have discussed in the manuscript and in further responses to reviewer #2. For example, we have collected 6x as many tweets from every day for the two weeks from an alternative dataset (Lamsal - see next response), yet after applying the same restrictions we ended up with a corpus of just 2x the size. This implies that our sample is naturally much smaller than the available data that consists of retweets and tweets from multiple users - tweets that are of no use for our analysis.

Regardless, at the time of data collection multiple datasets of Twitter related to COVID-19 existed. I strongly recommend repeating this analysis in two ways to see if the results change or hold:

- one, now that the authors have been collecting more data for a while,

- and two, perhaps more pressingly, using one of the public open datasets for Twitter with millions of tweets. See e.g. this collection of resources: http://www.socialmediaforpublichealth.org/covid-19/resources/ "Twitter Data"

Reply: We have included a paragraph in the revised manuscript that shows how our results about the impact of WAR terms hold when we look at our updated corpus (59 days of collection) of 654,354 tweets (Section Replication studies). Analyzing all tweets from the updated database, 5.54% tweets contained at least one term from the WAR framing, which previously was 5.32%. We argue that the increase of >0.22% in the WAR framing can be partially explained by new debates entering the discourse. For example, we could identify the topic of increased domestic violence (as a consequence of the lockdown), increased cyber criminality with “attacks” exploiting the anxiety during the pandemic and the increased involvement of the military in supporting and restricting the public. It will be highly interesting to separate those topics and observe them in another topic modeling analysis, yet we do not include such analysis as it would change the nature of the paper, which is focused on the early stage of the discourse.

Reply: We agree that the size of our initial dataset was limited and therefore it is crucial to replicate the results using more data. Yet, our criteria (individual tweeter, English only and the specific keywords) are very specifically tailored and embedded in our theoretical considerations. Nonetheless, we replicated our analysis on a new corpus, extracted from an existing dataset “Coronavirus Tweets Dataset” (Rabindra Lamsal. (2020). Coronavirus (COVID-19) Tweets Dataset. IEEE Dataport. http://dx.doi.org/10.21227/781w-ef42). A comparison of the Lamsal dataset with our dataset over the same time-frame can be seen in the Figure below. Notably, the relative order of proportions is the same as in our analysis: Family > War > Storm + Tsunami + Monster. Differences (Family proportion decreased by 3.46%, War increased by 1.62%) can be explained by different keywords that have been used to acquire the Lamsal dataset, e.g. it includes “Corona”, which is arguably a more colloquial expression that we did not use to select our corpus. Moreover, their keywords change multiple times during the process of data mining, while we kept the same set of keywords. Whereas the suggested dataset from the online repository at www.socialmediaforpublichealth.org/covid-19/resources/ "Twitter Data" includes all languages, which made it infeasible to apply language identification on such a large dataset.

Reply: Additionally, we have compared our updated corpus (two months of tweets) with the Lamsal corpus covering two months of tweets (also depicted in the Figure present in the docx of this response). We can observe that the distributions of tweets within each frame follows the same trends across the various corpora: the data collected on the 2 weeks corpora (ours and Lamsal’s) are very similar (compare bars with bright colors to bars with pastel colors). This figure along with the numerical results and a brief discussion have been added to the updated manuscript, in the section Replication studies.

(This also suggests an opportunity to do temporal analysis, to see if the frames and discussion have changed and if so, how they are changing. This may help with a practical contribution - to answer if discussions are moving in a healthy or helpful direction, or the opposite, and why?

- For example, how often do these topics found happen over time, how often do these frames happen over time, and why is that important?

Reply: Thank you. We did not identify a change in the distribution of the figurative frames within the discourse on Covid, within our initial 14 days of observation. With our updated corpus of 2 months of tweets, we also did not see a change of the proportion with which the frames occur. The Figure below (present in the docx of this response) depicts the percentage of tweets containing at least one term from the respective frame over the course of 2 months (dashed lines indicate the projection over 3 missing days, due to technical issues with the Twitter API. Notably, similar issues can be found in all other publicly available datasets). A thorough statistical analysis of these observations and the correlation of this temporal analysis with the progression of government regulation, infection rates and news are beyond the scope of the current paper. But we are currently pursuing this study, for a different contribution that will have a longitudinal perspective and will show how the different topics and frames change across time, week after week.

How would we interpret these topics, and why might they be important? How do these frames correspond with hashtags or the literal discussion of the disease?)

Reply: Thank you. In the manuscript we have provided a description of the topics identified by LDA, which is indeed our interpretation of the topics (including their labelling).

Reply: The only hashtags that we have considered in our analysis were the hashtags used to construct the corpus of tweets (#covid etc.). We did not analyze any hashtag related to the frames, because it was not in the scope of our analysis (see our research questions).

Thirdly, please see critiques of the methods, related to LDA and Twitter pre-processing.

Reply: Thank you. We replied to those.

Fourthly, I also include more minor points and notes about statistical significance.

Reply: Thank you. We replied to those.

I also have concerning methods critiques that may undermine results:

On tweet processing decisions:

- I'm struggling to understand why the authors eliminated all but one tweet per user. This is a limitation. It looks like the methods and results are at the level of a tweet, not at the level of a user. In addition to the sampling bias, the authors could be discarding data that is important to their analysis. If the authors insist on retaining only one tweet per user, how this was performed? Was this random? If not, this could bias one's data again.

- I'm struggling to understand why the authors excluded retweets and mentions. How many retweets and mentions are there? Together, these choices severely limit the amount of analysis possible, to show how often the frame of the discourse is spreading, occurring, or changing. I understand wanting to exclude them initially, but what about repeating the analysis with them included to see how it changes?

Reply: Thank you. We constructed a corpus, which follows specific criteria that we motivated in the paper. A corpus is a balanced and representative collection of texts. As we explained in the corpus, we limited one tweet per user precisely to avoid having a biased corpus where super-tweeters could have biased the percentages of frequency distribution of frames. Imagine for example that one specific Twitter user who uses Twitter many times per day, is very fond of sci-fi and monsters-related issues, he/she might use the MONSTER framing very frequently in their tweets, also about Covid-19. If we kept all the tweets by this user, this might have biased the distribution frequency of the MONSTER related words: we would have observed a high percentage of tweets with this frame, but these are all produced by the same person, who simply happens to tweet very often. This is not representative for the population. Instead, by limiting one tweet per user, we kept our corpus relatively manageable from a technical perspective (not too heavy) as well as representative and balanced from a theoretical perspective. We also dropped the retweets and mentions because these are duplicated texts, so, the same datapoint would have been counted twice. We argue that this is not desirable, from a corpus linguistics perspective. We have explicitly acknowledged these matters in the section “Constructing the corpus of Covid-19 tweets”. To conclude, our choices for the configuration of the corpus are theoretically motivated and clearly explained in the paper. We do not expect the reviewer to necessarily like them, but methodologically speaking, they are motivated and transparent. As explained later, we have added further analyses to replicate our results on an updated corpus constructed with the same criteria, collected by ourselves, and on a corpus constructed on the basis of existing resources (section “Replication studies").

On LDA implementation:

- Did the authors use Gibbs sampling or variational inference? Gibbs sampling has been shown to yield vastly superior topics. I'd recommend repeating analysis if used variational inference and see if results hold.

- how did the authors choose 4 vs 16 topics? why not other numbers? did they check perplexity - what number of topics has the lowest perplexity? (most likely to explain the data)

Reply: In the manuscript we have used an online variational Bayes (VB) algorithm through Gensim’s LDA-Multicore implementation. The topic numbers 4 and 16 have been chosen intentionally based on two factors. 1) We have observed the words in 4, 8 and 16 clusters and identified 4 and 16 to be semantically most informative and insightful. 2) We have compared the coherence values over 1 to 16 topics using Cv coherence (Syed, S., & Spruit, M., 2017). It is fair to assume that the slight increase of coherence after 16 topics is negligible given the fact that more and more topics might be more coherent, yet not more insightful or semantically meaningful. Therefore, we have chosen 16 topics at the higher end of coherence scores and for a small topic number, we have avoided choosing 2 topics (with the worst coherence score).

Reply: Additionally, we have now included a comparison of the coherence score over 1-16 topics using Gibbs sampling (through Mallet’s Gensim LDA wrapper). We are grateful for bringing this to our attention and report that our results hold. Firstly, the coherence score does not greatly change between the two algorithms (see Figure present in the docx of this response). Secondly, we have investigated the actual topics (4 and 16) for both algorithms and although some of them are different (as to be expected), there are no greatly different topics or insights to be expected. We have included the trained LDA model for 4 topics and 16 topics using Gibbs Sampling in the online repository at Models/lda_N4_Gibbs.pkl and Models/lda_N16_Gibbs.pkl. 

- How did the authors handle hashtags and URLs and usernames? These may contain information, or not, depending on the design of the study. What happened? These may be useful to report if analyzing the discussion.

Reply: Thank you for your questions. We now make it more explicit in the paper by stating that “[i]n our study design, we are not interested in the dynamics within the social network and have not collected retweets and did not investigate usernames, hashtags or URLs.” We have simply included usernames, hashtags and URLs in the frame analysis and LDA training data. Which is for example why we have observed @realdonaldtrump in a topic word cloud.

On LDA interpretation and results:

- the authors look at significant words in topics, but what about tweets most about those topics? it can make a difference, per the coming citation. I recommend evaluating topics in both ways, as it may affect results of lines 585-596. See https://scholar.google.com/scholar?hl=en&as_sdt=0%2C21&q=reading+tea+leaves+humans+interpret+topic&btnG=

Reply: Thank you for this reference. Yes, we have looked at most significant words in the topics (similar to word intrusion in the referred paper). In our section “LDA Topic prediction of WAR tweets”, we are in fact looking at those tweets most about the topics. Admittedly, we only do this for tweets that contain a word from the WAR frame and not for the topics themselves in order to validate their cohesion (similar to topic intrusion in the referred paper). Overall, we now include a paragraph in the revised manuscript that will be more critical about the proposed topics, highlighting the limitations and suggested improvement of the modeling approach as it stands (see Discussion of LDA Topic prediction of WAR tweets). Arguably, there are many ways in which we can adopt and improve the topic modeling, yet our interpretation and especially the analysis of the framing within those topics has been done on a word and tweet level.

- did the authors interpret all of the 16 topics like they did the 4 topics? (lines 450-453)

Reply: We have interpreted all of the 16 topics in the sense that we relate them to the 4 topics (lines 454-463). Here, we also interpret novel topics (line 461-463).

- for topic figures, I suggest putting names of topics in the figure axes where possible. without them it's inconvenient to remember which is which

Reply: Thank you. However, because the labels for these topics have been provided by us, rather than being returned by the model, we believe that indicating these labels in the figures would be misleading. In other words, the labels of the topics are our own interpretation of the results of the LDA analysis. The topic model provides numbers for the topics (1,2,3 and 4), and based on the clusters of words within each topic we provided a label.

On Twitter pre-processing, I'm worried about the authors' use of general-language tools on tweets which have been shown to use vastly different language structure. - stopwords from 2012... check/justify that these are up-to-date and apply to Twitter? need to come up with domain-specific ones? - along the same lines there's a twitter tokenizer (e.g., stanfordNLP, NLTK) that are custom-built for this... what about emoji, how were these handled?

Reply: Thank you. This is valuable feedback and we are grateful for pointing out these specifics. With respect to the general-language tools, we have to explain that we have used stopword-removal, lowercasing and tokenization for the pre-processing. Firstly, our stopwords list was an updated version and we did not only use the original list from 2012, we have included domain specific stop “words” such as “http”, “&” and new format stop words e.g. “don’t, don`t, I'm, I`m”. The manuscript is now updated to clearly state that we have used an updated stop word list.

Reply: As for the Twitter tokenizer, we were not aware of a specific Twitter tokenizer especially for the language library (Gensim) that we have built the analysis with. Due to the raised concerns, we have investigated the source code of the NLTK TweetTokenizer (www.nltk.org/_modules/nltk/tokenize/casual.html#TweetTokenizer) in order to understand its possible improvements over our approach and whether or not they are negligible. The TweetTokenizer removes user handles (@). We are actually indifferent in our analysis about those as we have mentioned before. It removes html entities, which we partially remove with our additional stop words. Notably, it removes repeated characters (heeeeey), which we do not. Yet, we filter tokens with a length < 3 (“as”, “aa”, “to”, “yo” etc). It handles emoticons, emoji differently than we do. We agree that a Tweet tokenizer might reduce the number of tokens overall, but investigating our tokens, our pre-processing and stop words can compare with the same relevant tokens. Yet, we fully acknowledge the possible improvement for any future work.

- line 279: better would have been to use tf-idf and leave the common terms in... these would have been reduced by the weighting organically

Reply: Thank you. We are grateful for this advice. Irrespective of the removal of common terms, we did not see any improvement of the LDA w/ or w/o tf-idf, as mentioned in the manuscript.

On literal framing control:

- What about the literal frame "it's a disease"? The authors chose family as a literal frame- this may strongly coincide with incidence or deaths from the disease, which may not be exactly what the authors want to measure.

Reply: Thank you. We have considered several literal frames, including DISEASE and VIRUS. However, these words used within these frames are indeed often related to the figurative frames used for the analysis (fight, defend, attack, for example). In this sense, a frame like DISEASE or VIRUS, are hyper frames, they are at a different level of abstraction, compared to WAR, STORM, FAMILY, etc. Within the DISEASE frame we could still find the frames of FAMILY, WAR etc. We picked FAMILY because, instead, it appears to be more on the same level of abstraction of the figurative frames.

- In addition... how are the authors controlling by including this? Should this be used for normalization, or testing statistical significance of frequencies or of differences among frame?

Reply: Thank you. The literal frame is used as “control” in the sense that it is used to compare the frequency distributions of the figurative frames (metaphorical frames: war, storm, etc). As explained above, however, the statistical distribution of frequencies cannot be not easily calculated. We opted for a couple of replications, to show in a descriptive manner that the percentages are substantially preserved and the ranking of use of the frames too: the most frequent frame that occurs in the corpus (and in the secondo corpus collected for replication, and in the external corpus on which we ran our analysis for comparison) is FAMILY; followed by WAR (the most frequent metaphorical frame), followed by STORM, then TSUNAMI and eventually MONSTER.

On Results and discussion

table 2 - are these results statistically significant? This would give weight to the authors' statement about the relative amount.

Reply: Thank you. In response to Reviewer #1 (Remark #7), we have provided statistical (significant) results for the relative difference of proportions (using Cochran’s Q - a multi-sample extension of the McNemar test).

Reply: We used the Cochran’s Q test to assess if there are any significant differences between the 5 frames, in the ways they are represented in the corpus of tweets. To do so, we compiled a tweets by frames matrix that displays binary values for presence or absence of a frame-related term in each tweet. Cochran’s Q test statistic = 47,226.72 , df=4, p < 0.001. The post-hoc pairwise McNemar test is highly significant (p < 0.001) for all permutations (all frame pairs). We can therefore conclude that there are significant differences in the presence of target words between each pair of frames.

lines 512-536, about topics predicting occurrences of frames... are these differences between frequencies statistically significant? are these frequencies high enough to matter?

Reply: Thank you. We will clarify the statistical significance in this case: the reported frequencies are probability distributions that are being inferred by our LDA model. To the best of our knowledge it is non-trivial to provide a p-value for this prediction, because its accuracy relates to the goodness of the LDA model. The model itself provides a probability for each tweet to belong to a topic. As topics #1, #3 and #4 show a probability of >24% and topic #2 a probability <8%, we have interpreted this prediction with respect to the quality (coherence of our model) and concluded a meaningful difference. If the reviewer is aware of any statistical test that can be applied here, we would greatly appreciate the feedback and include it.

Continuing down the path about frames vs. topics:

- How often do the family or alternative frames show up in the predicted topics? like likes 383-384 for the WAR frame.

Reply: We cannot provide absolute numbers for the occurrence of the family or alternative frames in the predicted topics, because we can only use the LDA model to infer a probability of each tweet (from family terms / alternative framings) to belong to a topic. We have provided a few more probability distributions below to show that the predicted topics are in fact different for the different frames, yet the difference is diluted when observing the 4 topics (see Figure present in the docx of this response). We have decided not to include a discussion of all possible distributions (4 frames x 2 topic sizes (4, 16) = 8 distributions), as it would not contribute (nor contradict) our hypothesis.

- In addition, how many topics include words in the frames? This may be an indicator if the frames are even worth studying on this domain. (see 90% number and earlier comment about frequencies)

Reply: We think there is a conceptual misunderstanding here. A topic does not include a word or not. Each “topic” is a probability distribution of all tokens. That means that a token “trump”, “news” or “family” belongs to a topic with a certain probability. Therefore, all of the topics include words of the frames, some with high probability, some with low probability. To check the probability of a word within a frame one performs an inference of that word. Doing so for all words of a frame is what we have already done with the reply to the previous question. We can look at the probability from two sides, the topics or the frames, but the distribution is the same. Lines 55-67 do the authors have any citations for any/all of these statements?

Reply: Thank you. We do not believe that the text on these lines, informally describing the general situation generated by the current pandemic, require references to scientific literature. This would only disrupt the reading flow.

Reviewer #5

 General comments:

This article provides some interesting first insights into what topics are discussed in relation to the Covid-19 pandemic on Twitter and what metaphor frames are prevalent. It was slightly disappointing that the research questions are purely descriptive, but maybe that is inevitable for a study that must be one of the first on the topic.

Reply: Thank you.

What is headed 'theory section' in fact includes both a literature review on analysing tweets on epidemics and another part on metaphor theory, but the two parts are not linked well. Also, the results are not linked back to the studies featuring in the literature review. 

Reply: Thank you. Indeed, both parts pertain to the Theoretical Background and they are clearly separated because these are the two “ingredients” that we combine in our paper: social media analyses of pandemics and metaphor studies. They are not supposed to be linked well, that is why they are separated. We have, however, explained that in discourse analysis and metaphor studies topics such as the pandemics can be analyzed in terms of figurative framing, and mentioned the relevant literature.

The paper seems to have been written quickly and was clearly not proofread.

Reply: Thank you. We apologize for this limitation, which we have addressed in this revision. Indeed, we were probably too eager to submit, given the timely nature of the topic.

Specific comments: 

• As for the corpus, how can the authors be sure that the corpus only contains tweets by non-experts?

Reply: Thank you. We have not argued that. What we have argued is that while in a corpus based on newspaper articles, for example, the texts have been written by professionals (journalists), on social media and therefore on Twitter, anyone with an account produces texts (the tweets).

• For Study 2, why did the authors decide against lemmatisation? There are good reasons not to use that process, e.g. because different forms of a lemma can express different metaphor scenarios, but the reasons should be spelled out.

Reply: Thank you. We added this specific motivation to the description of the corpus, as the reviewer suggests. Very much appreciated the constructive nature of this comment.

• The #ReframeCovid also includes storm, monster and tsunami source domains, and it does not only include examples from news media. 

Reply: Thank you. Now, yes. We updated this information in the manuscript, in the section “Search method for alternative framings and relevant lexical units therein”. When we submitted the manuscript, the #ReframeCovid initiative had just been proposed as a hashtag on Twitter, and the crowdsourced collection of frames was about to start.

• What are possible labels for the 16 topics?

Reply: Thank you. The 16-way solution is a highly granular solution, generated by the topic model. Because the labels are not provided by the model, these have to be deduced by the analysts in an empirical fashion, by means of intuitions, basically. This is commonly done in topic modelling, but the more clusters the model generates, the more it is hard to name with an individual label each cluster (that is, each topic) provided by the model. For this reason, we discussed and provided labels to name the 4-way topic model, but we did not feel comfortable in labelling each and every cluster within the 16-way analysis. Our labels might have been easily criticized by the reviewers. For this reason, we instead opted for showing the 16 topics returned by the analysis in the form of word clouds, we discussed the 16 topics as sub-categories of the 4 topics, showed which of these 16 topics contain war-related terms (topics 2, 7, and 10), and to what type of arguments the other topics seem to refer.

• Why was family chosen as a literal control frame?

Reply: Thank you for this question, which gave us the opportunity to clarify this aspect, which was also pointed out by two other reviewers.

Reply: In traditional framing theory, framing is defined as “select[ing] some aspects of a perceived reality and mak[ing] them more salient in a communicating text, in such a way as to promote a particular problem definition, causal interpretation, moral evaluation, and/or treatment recommendation for the item described” (Entman, 1993,p. 53).

Reply: FAMILY is a frame that, as another reviewer pointed out, can be used to talk about covid (it is an aspect of this reality) that can be used figuratively, as well as literally. Another reviewer mentioned: “family terms COULD be figurative (and indeed, Lakoff, for example, has written much about the figurative uses of FAMILY in describing governments” (for example, within the EU, one could say that Germany is the responsible father of the family, who scolded Italy for her behavior etc). We have chosen FAMILY deliberately to be a frame that is most likely not used figuratively, yet has comparable properties to a frame we expect to be used figuratively (e.g. WAR, STORM or MONSTER). In fact, we have checked qualitatively the tweets that feature one of the FAMILY terms in it, and it appears that they are used literally, not metaphorically. So, for the covid reality, FAMILY is used typically as a literal frame.

Reply: Moreover, in communication sciences a frame is typically defined as consisting of two elements (Joris,d’Haenens, & Van Gorp, 2014, p. 609): framing devices which are elements in a text and specific linguistic structures (for example a list of words related to a frame) and reasoning devices which are the (latent) information in a text through which the problem, cause, evaluation, and/or treatment is implied (a more conceptual-communicative dimension of a text). A topic operationalized by topic modelling, corresponds to the first of these two components of a frame (i.e., a list of semantically related words). For example, one can say that sister, father, home, parenthood, brotherly love are all words that together represent the topic of FAMILY, which can be used as a frame in covid discourse: by talking about these family aspects of covid, one could direct the readers’ attention to the importance of staying together and helping each other, rather than looking at one another as strangers and potential risks of contagion. But a topic can also be used as a frame, if it has the communicative purpose to highlight a specific aspect of a situation to fulfil a specific communicative purpose.

Reply: Finally, one reviewer suggested to use, as an alternative frame, something like DISEASE. Infact, we have tried (before settling on FAMILY) to check for DISEASE and VIRUS as literal frames. However, the lexical units within these two frames encompass words related to the figurative frames (fight, defenses, attack, etc). We realized that these (DISEASE and VIRUS) are hyper-frames, that is, they are at a higher level of abstraction, compared to the frames WAR, STORM, or the literal FAMILY. We therefore opted for FAMILY because, we argue, is at the same level of abstraction of WAR and the rest of the figurative frames.

Reply: We have added this discussion in the revised manuscript, by introducing the definition of framing in the theoretical background, explaining that the FAMILY frame is used literally in this type of discourse (end of Theoretical Background section), and by explaining the relation between framing and topics (as intended in topic modelling) in the beginning of the section “Identifying topics in Covid-19 discourse on Twitter through Topic Modeling”.

• "Previous literature shows that metaphor-related words cover only a percentage of the discourse": does this refer to Steen et al. (2010)?

Reply: Thank you. Indeed. Yes. We have added this reference and elaborated this aspect further. The edited text is hereby reported:

Reply: “previous literature shows that overall metaphor-related words cover only a percentage of the discourse, and that literal language is still prevalent (e.g., Steen et al. 2010). Steen and colleagues, for example, report that literal language covers 86.4% of the lexical units, while metaphor-related words cover just 13.6% of the lexical units. Their analysis is based on a sub-corpus of the BNC that encompasses 187,570 lexical units extracted from academic texts, conversations, fiction and news texts. All parts of speech are included in their analyses, including function words (such as prepositions and articles). Assuming that this percentage of figurative language compared to literal language use can be applied to the discourse around Covid-19, we would expect to find around 13% of the lexical units in our corpus to be used metaphorically (including pervasive metaphorical uses of function words). Many of these metaphor-related words pertain figurative frames such as the frames analyzed in our paper. In this perspective, the percentage of use of the WAR frame reported in our study suggests that this frame is particularly pervasive, because it covers more than one third of all the metaphorical language typically used in texts.”

---

## [Decision Letter · Decision Letter 1]

31 Jul 2020

PONE-D-20-10986R1

Framing COVID-19 How we conceptualize and discuss the pandemic on Twitter

PLOS ONE

Dear Dr. Wicke,

Thank you for submitting your manuscript to PLOS ONE. After careful consideration, we feel that it has merit but does not fully meet PLOS ONE’s publication criteria as it currently stands. Therefore, we invite you to submit a revised version of the manuscript that addresses the points raised during the review process.

I have heard back from 3 of the original reviewers. As you can see, once again the colleagues have made incredibly helpful suggestions for revisions. I invite you to consider each point carefully. Please pay particular attention to the clarity of language issues highlighted by reviewers 1 and 2. I agree with them that you have your manuscript proof-read prior to submission of the next revision. And please especially focus on the technical issues with the data highlighted by reviewer 3. The reviewer has given no less than 4 possible ways forward, any of which I find entirely reasonable. At the very least, you should acknowledge the limitations of your data processing method, and modify your theoretical claims accordingly, as suggested by the 1st option the reviewer puts forward. I look forward to receiving your revision as per the guidelines below.

We look forward to receiving your revised manuscript.

Kind regards,

Panos Athanasopoulos, Ph.D

Academic Editor

PLOS ONE

Reviewers' comments:

Reviewer's Responses to Questions

**Comments to the Author**

1. If the authors have adequately addressed your comments raised in a previous round of review and you feel that this manuscript is now acceptable for publication, you may indicate that here to bypass the “Comments to the Author” section, enter your conflict of interest statement in the “Confidential to Editor” section, and submit your "Accept" recommendation.

Reviewer #1: (No Response)

Reviewer #2: (No Response)

Reviewer #4: (No Response)

2. Is the manuscript technically sound, and do the data support the conclusions?

Reviewer #1: Partly

Reviewer #2: Yes

Reviewer #4: No

3. Has the statistical analysis been performed appropriately and rigorously? 

Reviewer #1: I Don't Know

Reviewer #2: Yes

Reviewer #4: Yes

4. Have the authors made all data underlying the findings in their manuscript fully available?

Reviewer #1: Yes

Reviewer #2: Yes

Reviewer #4: Yes

5. Is the manuscript presented in an intelligible fashion and written in standard English?

Reviewer #1: Yes

Reviewer #2: No

Reviewer #4: Yes

6. Review Comments to the Author

Reviewer #1: I appreciate the careful work of the authors to address the concerns raised in the previous round of review. The manuscript is clearer as a result and I think it will make a solid contribution to the literature. I have a few remaining comments and concerns that I describe below.

Introduction & Theoretical Background

The introductory sections do a better job of describing and situating the contribution of the current work. The paragraph that explicitly details “The innovative aspect of this paper…” (p. 5) is particularly helpful.

There is still an issue with the sentence I highlighted in my previous review, which now reads, “Unlike the articles on magazines and journals typically used for corpus analyses of this kind, Twitter contains messages written by journalists and other experts in mass communication: most tweets are provide by non-expert communicators” (p. 4). Maybe: “Although Twitter contains messages written by journalists and other experts in mass communication, most tweets are provided by non-expert communicators…”

I still find the heading “Theoretical Background” misleading. While that section certainly does discuss relevant research that helps situate the current work, it doesn’t discuss theoretical background per se (to my eye). But if the authors feel strongly that it does, that’s ok with me.

I feel more strongly that the heading “Experimental design” should be changed because the study is not an experiment. Maybe “Study design.”

Methods

I appreciate the revised structure of the methods and results, which makes the paper more readable. I still find the first section difficult to interpret. I think it would be useful to say explicitly in the set up or discussion of the topic modeling that the categories will be useful for understanding how the war frame is being used.

It’s not clear what it means to say that, “The two numbers of clusters were chosen on an empirical basis” (p. 17). What was the empirical method / criteria? How was it evaluated? Adding the LDA coherence measure is useful but seems different from what the authors mean when they say that the “clusters were chosen on an empirical basis.”

I appreciate the discussion of the MIP and MIPVU in the response letter, but I think some of these issues should be noted in the manuscript itself. Namely, this method of categorizing and quantifying metaphor is new, different from alternatives, and has its own benefits and limitations (as does any method of coding language). To be clear, I think the method being used is interesting and worthwhile. But it likely misses some instances of metaphor and it likely counts some words as metaphorical that are being used in a literal sense. Maybe, for example, someone tweeted about how COVID was spreading on the aircraft carrier the USS Theodore Roosevelt, which is a war ship. Given that one of the main contributions of the paper is the method, I think some attention should be given to the strengths and limitations of the method. Of note, a major strength of the method is that it can handle a lot of data. This point could also be made more explicitly. [The comment below also relates to a limitation of the coding method.]

I agree that the concept of a STORM is different from the concept of a TSUNAMI. However, if the same words (e.g., wave, flood, disaster) are being used to index both concepts, how do we know which concept people are drawing on? For example, the word “disaster” is the second most frequent marker of a TSUNAMI in the corpus and also the second most frequent marker of a STORM in the corpus. The word “wave” is also frequent in both. This means that some of the tweets are double counted as metaphorical in a sense, no? Do the analyses assume that the categories are mutually exclusive and exhaustive? It would be useful to report the percentage of tweets that included multiple categories of metaphor.

I understand and appreciate how the cluster of words related to FAMILY are being used, as a point of comparison with the figurative frames (WAR, STORM, TSUNAMI, MONSTER). But I still think there is a qualitative difference between this category and the others. The language of WAR (STORM and TSUNAMI and MONSTER) is being used to frame different aspects of the pandemic and response. In many cases, the issues being framed metaphorically could be discussed using an alternative metaphorical frame or in a literal sense. The tweets about FAMILY, on the other hand, seem to be about FAMILY per se. That strikes me as an important difference that should be discussed in the paper.

Results

See above for questions about tweets that were categorized as relating to multiple frames.

Adding the replication study strengthens the paper.

I recommend toning down some of the claims like, “…the percentage of use of the WAR frame reported in our study suggests that this frame is particularly pervasive, because it covers more than one third of all the metaphorical language typically used in texts” (pp. 33-34). The study restricts itself to four metaphorical frames of interest; the full range of metaphor typically used in texts is not quantified here.

Reviewer #2: I want to commend the authors on the impressive revisions to this paper. I think they have done just about everything they could to address all of the reviewers concerns, and I feel the paper is much stronger now and will make a nice addition to the literature. Indeed, I could see myself citing the findings in a future talk or article!

The only reason I am selecting "minor revision" rather than "accept" is that PLOS ONE does not copyedit accepted manuscripts, and I still feel the paper could use one more round of proofreading and editing. While the overall structure of the paper is greatly improved now, there were still grammatical errors and awkward phrases sprinkled throughout that hindered readability.

For example, lines 75-77 contained the sentence "Unlike the articles on magazines and journals typically used for corpus analyses of this kind, Twitter contains messages written by journalists and other experts in mass communications: most tweets are provided by non-expert communicators" This is very confusing to me as it seems to be saying two different things. A more mild example, but still one that should be edited for clarity, comes on lines 179-180: "The military metaphor thanks to which we frame diseases such as cancer is a very common one to be found in public discourse." And in the next sentence, the authors use the article "the" before Time Magazine, which is not necessary.

I am not going to go into every example of confusing writing, but I do want to recommend the authors go over the paper again and update the prose as necessary to enhance clarity. At that time, I will be prepared to accept!

Reviewer #4: Thank you to the authors for drastically improving many points about the paper. I enjoyed reading this interesting research even more this time. It is much cleaner to read and understand, including contributions, backing statistical analyses, and replication on other datasets. Other reviewers also brought up great points that I did not think of.

However, as is written, the study seems to have unacknowledged limitations and concerns that render contributions too broad about understanding general discourse on Twitter that I would recommend addressing before accepting this paper.

My concern stems from ambiguity in terms of unit of analysis that biases the representativeness of the data, which causes unacknowledged impact on results and contributions. Claims cannot be made about general discourse on Twitter, but they *can* be made about a biased yet useful subset - those tweets that originate from less-frequent tweeters.

Generally, any study that performs social monitoring needs to:

- make clear what its unit of analysis is,

- then accordingly determine what it means to be representative of these units in its sampling frame,

- gather units to analyze and analyze those same units

- acknowledge any limitations of representativeness

- and limit contributions accordingly

At minimum in this particular study,

- it should be made clear and consistent through data collection and analyses whether this particular study is analyzing use of frames by users, or use of frames in tweets, (I'm pretty sure it's tweets, right?)

- the limitation of a bias against super-tweeters should be made more explicit, and I would accordingly recommend explicitly walking back the study's claims, research questions, and contributions, because claims of general representativeness on Twitter are not defendable.

Specifically, if it is chosen to omit the vast majority of tweets by keeping one tweet per user and omitting retweets, claims cannot be made about how the general Twitter discourse discusses and frames COVID-19 (or to put it another way, the study's data is not representative of all tweets). Claims instead can be made about how those tweets *originating from less-frequent tweeters* discuss and frame COVID-19.

Either that, or I would recommend at least one of:

- switching to analysis at the level of a user

- more representatively sample from tweeters (as one tweet per user is usually not seen as representative of a user's content, whether they are among the top tweeters or not)

Please find details below - perhaps these can make the issue more clear if it helps.

First, the study has a bias against retweeters and sharers, in favor of original tweeters, as is stated. However, many users on Twitter retweet or share someone else's tweets - this is seen as a proxy for behavior and opinion, as usually people share things they agree with. Therefore, more accurate counts of frames might be *including* those frames that are retweeted. At the very least, one may analyze with retweets and without to see any differences. It's fine that the study disregards retweets, but this should be acknowledged as a limitation and claims about discourse should be modified accordingly.

And second, to paraphrase, data is being "limited to one tweet per user because of technical limitations and to make the corpus balanced and representative". I would reject the study's claim that the resulting corpus is representative - because the study is not clear about *how* it is representative. As designed, it seems like the study is confused about the unit of analysis that it is trying to represent: the tweet or the user.

- If the study wishes to accurately represent tweets, then by Twitter's nature there are super-tweeters, and frequencies of frames will be biased by super-tweeters who use frames more. This is an accurate representation of the Twitter discourse.

- If the study wishes to accurately represent users, then it should state this, and tweets should be aggregated or sampled representatively per user, and and all subsequent analyses should be at the level of the user, not the level of the tweet.

However, as currently designed, the study starts at the level of a user by "retaining one tweet per user", but then moves back to analyze at the level of a tweet. This results in a bias against super-tweeters with "retaining one tweet per user". Therefore, all subsequent analysis at the level of a tweet omits the vast majority of tweets that compose Twitter discourse - exactly what it seems to want to analyze.

- Minor point: The study should address exactly how it "retained only one tweet per user". This was asked in the review. Was the earliest tweet retained? the most recent? the one with the most framing words? Is it simple random sampling? I ask to elucidate, not to be flippant, as ambiguity does not lead to reproducibility - it leads to concern. A brief mention would suffice.

I can recommend four courses of action, although the authors would know which would be more appropriate for their desired contributions:

1. Keep current data collection, methods, and results that seem to analyze at the level of a tweet, and acknowledge explicit bias against super-tweeters, to favor those users who tweet less often. This seems to require revisions to contributions and research questions, as the overall Twitter discourse is not being studied for framing, but the Twitter discourse being studied is biased towards discourse by less-frequent users. Is there a theoretical motivation for studying in this way?

2. Keep current data collection, analyze at the level of a user, and acknowledge explicit bias against super-tweeters, to favor those users who tweet less often. This would require more extensive revisions to analyses, but would result in a different contribution: instead of studying the entirety of framing discourse on Twitter, studying how *users*, with emphasis on less-frequent users, frame the discussion.

3. Decreasing bias against super-tweeters by gathering a representative amount of tweets per user, and analyze at the level of a user. This would result in a proper representative analysis of discourse on Twitter at the level of the user, and contributions on how users discuss and frame would follow.

4. Decreasing bias against super-tweeters by gathering a representative amount of tweets per user, and analyze at the level of a tweet. This would result in a proper representative analysis of discourse on Twitter at the level of the tweet, and defendable contributions on how tweets generally discuss and frame would follow.

7. PLOS authors have the option to publish the peer review history of their article (what does this mean?). If published, this will include your full peer review and any attached files.

Reviewer #1: **Yes: **Paul Thibodeau

Reviewer #2: No

Reviewer #4: No

---

## [Author Response · Author response to Decision Letter 1]

17 Aug 2020

To the Editor and Reviewers,

Once again, we are very thankful to the reviewers, for taking their time to provide constructive and helpful comments to our paper, which has now significantly improved in quality. In this letter we have addressed the remaining comments and indicated what we changed in our manuscript. We have then submitted the paper to a professional proofreader and editor, native speaker of American English, to proofread the text. We hope that this iteration has cleared any remaining language issues highlighted by Reviewer 1 and 2.

As for the technical issues addressed by Reviewer 4, we have provided a lengthy response and several adaptations to our paper. We are grateful for the detailed feedback, yet we have to acknowledge that some of the criticism will remain, which we attribute not to a wrong but different methodology arising from different research areas (Corpus Linguistics vs. Social Media Monitoring). Find our detailed response below.

REVIEWER 1

Introduction & Theoretical Background

The introductory sections do a better job of describing and situating the contribution of the current work. The paragraph that explicitly details “The innovative aspect of this paper…” (p. 5) is particularly helpful.

Reply: Thank you.

There is still an issue with the sentence I highlighted in my previous review, which now reads, “Unlike the articles on magazines and journals typically used for corpus analyses of this kind, Twitter contains messages written by journalists and other experts in mass communication: most tweets are provide by non-expert communicators” (p. 4). Maybe: “Although Twitter contains messages written by journalists and other experts in mass communication, most tweets are provided by non-expert communicators…”

Reply: Indeed, we apologize, this was a typo that we didn’t catch because of the track changes, thank you for pointing this out, we corrected it.

I still find the heading “Theoretical Background” misleading. While that section certainly does discuss relevant research that helps situate the current work, it doesn’t discuss theoretical background per se (to my eye). But if the authors feel strongly that it does, that’s ok with me.

I feel more strongly that the heading “Experimental design” should be changed because the study is not an experiment. Maybe “Study design.”

Reply: Thank you. As explained in the previous letter, we do believe that the Theoretical Background discusses literature related to the topic addressed in the paper: quantitative analyses based on Twitter data related to epidemics (paragraphs 1-4); cognitive linguistic studies of figurative framings and metaphors, including the WAR frame and metaphors that we tackle in our study (paragraphs 5-7) and alternative framings (paragraph 8). This section does contain a selected review of the literature in these two fields of research, which is functional to our argumentation. We therefore remain with this heading. Thank you for your collaboration.

Reply: We changed the heading “Experimental design” into “Study design” as suggested by the reviewer.

Methods

I appreciate the revised structure of the methods and results, which makes the paper more readable. I still find the first section difficult to interpret. I think it would be useful to say explicitly in the set up or discussion of the topic modeling that the categories will be useful for understanding how the war frame is being used.

Reply: Thank you. In the setup of the study (Study Design) we explicitly mentioned: “To address our three research questions, first we explored the range of topics addressed in the discourse on Covid-19 on Twitter using a topic modelling technique. Consequently, we explored the actual usage of the WAR frame, and in which topics related to Covid-19 is the WAR frame more frequently used.” We have now made explicit in that paragraph the fact that the identification of the topics is functional to the investigation of the distribution of the war-related terms.

It’s not clear what it means to say that, “The two numbers of clusters were chosen on an empirical basis” (p. 17). What was the empirical method / criteria? How was it evaluated? Adding the LDA coherence measure is useful but seems different from what the authors mean when they say that the “clusters were chosen on an empirical basis.”

Reply: With the term “empirical basis” we mean that these numbers were initially chosen by looking at the data and then backed up by post hoc testing. Since 2 and 3 seemed intuitively to be a too small number of clusters that would not allow much observation to emerge, we opted for multitudes of 2, e.g. 4, 8, 12, 16. Then we tested with a post hoc analysis the coherence of the various cluster solutions obtained, and based on these measures, we confirmed that 4 and 16 clusters appeared to be the best choice, given our goal to have a less granular (4 clusters) and a more granular (16 clusters) solutions. Clusters in between 4 and 16 did not provide any meaningful semantic insight, which had not been generalized by 4 clusters or specified by the 16 clusters. The post hoc coherence analysis was conducted by evaluating the coherence score calculated for each cluster solution. The cluster coherence score that we adopted is typically used in topic modelling and with the LDA algorithm, and it is called “Cv measure”. This score tells us how coherent a cluster solution (that is, a topic modelling analysis) based on a number of clusters (=topics) is. The score is based on a one-set segmentation of the most important words, a sliding window and an indirect confirmation measure. The latter uses normalized pointwise mutual information and cosine similarity (for more information on this, see: M. Röder, A. Both, and A. Hinneburg: Exploring the Space of Topic Coherence Measures. In Proceedings of the eighth International Conference on Web Search and Data Mining, 2015.)

I appreciate the discussion of the MIP and MIPVU in the response letter, but I think some of these issues should be noted in the manuscript itself. Namely, this method of categorizing and quantifying metaphor is new, different from alternatives, and has its own benefits and limitations (as does any method of coding language). To be clear, I think the method being used is interesting and worthwhile. But it likely misses some instances of metaphor and it likely counts some words as metaphorical that are being used in a literal sense. Maybe, for example, someone tweeted about how COVID was spreading on the aircraft carrier the USS Theodore Roosevelt, which is a war ship. Given that one of the main contributions of the paper is the method, I think some attention should be given to the strengths and limitations of the method. Of note, a major strength of the method is that it can handle a lot of data. This point could also be made more explicitly. [The comment below also relates to a limitation of the coding method.]

Reply: Thank you, we agree and we have now acknowledged these strengths and limitations of our method in the paper, comparing our way to identify automatically metaphor-related (frame-related) words in a very large corpus of data, to manual methods such as MIPVU, as suggested by the reviewer. We added a paragraph explaining this trade off at the end of the section “Determining lexical units associated with the WAR frame”.

I agree that the concept of a STORM is different from the concept of a TSUNAMI. However, if the same words (e.g., wave, flood, disaster) are being used to index both concepts, how do we know which concept people are drawing on? For example, the word “disaster” is the second most frequent marker of a TSUNAMI in the corpus and also the second most frequent marker of a STORM in the corpus. The word “wave” is also frequent in both. This means that some of the tweets are double counted as metaphorical in a sense, no? Do the analyses assume that the categories are mutually exclusive and exhaustive? It would be useful to report the percentage of tweets that included multiple categories of metaphor.

Reply: Thank you for this comment. The fact that there are lexical entries that appear in both domains tells us that the two domains are indeed semantically related, and therefore the same lexical entries can be used in both frames. This is semantically correct, and also, to be expected. The frames do not need to contain mutually exclusive sets of words: in fact, the more two frames are semantically close to one another, or similar to one another, the more they will share lexical entries. From an ontological perspective, the idea cannot be defended that words belong to one and only one frame. Our algorithm operates in a way that is motivated by these assumptions, and therefore as the reviewer correctly observes, can attribute a tweet to both frames, the STORM and the TSUNAMI, because the calculation is based on the frequency count of a lexical entry that may be listed in both frames, such as the word “disaster”. In this sense, each framing is measured independently from other frames, and we agree that there might be overlaps. This is a sustainable perspective in terms of its ecological validity: we cannot really distinguish whether in the mind of the tweeter there was specifically a STORM frame or a TSUNAMI frame, when they tweeted. It would be unrealistic to argue otherwise. Our analyses do not assume that the lists of lexical entries used for each figurative frame are mutually exclusive and exhaustive for that specific frame. They rather assume that the lists of lexical entries for each frame are theoretically motivated , meaningful and representative for the frame they stand for. Please note that the statistical test we applied in our analyses to compare the percentages by which the different frames are used takes into account the fact that a lexical entry may belong to more than one frame (in this sense, these categories are not mutually exclusive; the presence/absence of a lexical entry in a tweet, however, is a binary, mutually exclusive property: either the word is there, or it is not).

Reply: To clarify this further, our algorithm counted 1 hit whenever a lexical unit (a word from a list) was found in a tweet. Then, the numbers were added up within each frame. In this sense, our unit of analysis is the lexical units in the word lists, not the tweets. A tweet can be counted as many times as it has words related to a frame. For example a tweet such as “This wave of #covid is hard to fight” will count 1 hit for the TSUNAMI frame (wave), 1 hit for the STORM frame (wave) and 1 hit for the WAR frame (fight). Therefore, reporting the percentage of tweets that were counted more than once does not seem to be useful. We have however reported, in the previous and in the current version of the paper, the number of tweets that encompass more than one war-related term (N=1253).

I understand and appreciate how the cluster of words related to FAMILY are being used, as a point of comparison with the figurative frames (WAR, STORM, TSUNAMI, MONSTER). But I still think there is a qualitative difference between this category and the others. The language of WAR (STORM and TSUNAMI and MONSTER) is being used to frame different aspects of the pandemic and response. In many cases, the issues being framed metaphorically could be discussed using an alternative metaphorical frame or in a literal sense. The tweets about FAMILY, on the other hand, seem to be about FAMILY per se. That strikes me as an important difference that should be discussed in the paper.

Reply: Thank you. Indeed, these are qualitatively different frames because WAR, STORM, TSUNAMI, and MONSTER are figurative, while FAMILY is literal. The reviewer mentions: “ The language of WAR (STORM and TSUNAMI and MONSTER) is being used to frame different aspects of the pandemic and response. [...] The tweets about FAMILY, on the other hand, seem to be about FAMILY per se”. This is precisely because the first are used metaphorically (and therefore with a specific effect, generated by the metaphor used, which can be replaced by using another metaphor, or by using literal language instead). The latter is literal language, thus it is used to talk about the literal referents. We have chosen FAMILY deliberately to be a frame that is most likely not used figuratively, yet has comparable properties to a frame we expect to be used figuratively (e.g. WAR, STORM or MONSTER). In fact, we have checked qualitatively the tweets that feature one of the FAMILY terms in it, and it appears that they are used literally, not

metaphorically. So, for the covid reality, FAMILY is used typically as a literal frame. In any tweet that uses a term from FAMILY the context is still the pandemic due to the nature of the dataset.

Results

See above for questions about tweets that were categorized as relating to multiple frames.

Reply: Thank you, as explained above our analyses do not require the lexical entries to belong to mutually exclusive lists. For this reason we applied the Cochran’s Q test statistic in our analyses to compare the percentages by which the different frames are used takes into account the fact that a lexical entry may belong to more than one frame (in this sense, these categories are not mutually exclusive; the presence/absence of a lexical entry in a tweet, however, is a binary, mutually exclusive property: either the word is there, or it is not). Cochran's Q test requires that there only be a binary response (e.g. word is present/word is not present or 1/0) and that there be more than 2 groups of the same size (e.g. WAR, FAMILY, STORM, MONSTER, TSUNAMI). The binary response does not need to be mutually exclusive for the groups.

Adding the replication study strengthens the paper.

Reply: Thank you.

I recommend toning down some of the claims like, “…the percentage of use of the WAR frame reported in our study suggests that this frame is particularly pervasive, because it covers more than one third of all the metaphorical language typically used in texts” (pp. 33-34). The study restricts itself to four metaphorical frames of interest; the full range of metaphor typically used in texts is not quantified here.

Reply: Thank you, we toned down the argument and clarified it further, in relation to the percentages of metaphorical language reported by Steen and colleagues in their corpus analyses.

REVIEWER 2

I want to commend the authors on the impressive revisions to this paper. I think they have done just about everything they could to address all of the reviewers concerns, and I feel the paper is much stronger now and will make a nice addition to the literature. Indeed, I could see myself citing the findings in a future talk or article!

Reply: Thank you, this is great news and very much appreciated.

The only reason I am selecting "minor revision" rather than "accept" is that PLOS ONE does not copyedit accepted manuscripts, and I still feel the paper could use one more round of proofreading and editing. While the overall structure of the paper is greatly improved now, there were still grammatical errors and awkward phrases sprinkled throughout that hindered readability.

Reply: Thank you, this is very helpful. In fact, we have now had a (paid) professional proofreader, American English native speaker, proofreading this version of the manuscript.

For example, lines 75-77 contained the sentence "Unlike the articles on magazines and journals typically used for corpus analyses of this kind, Twitter contains messages written by journalists and other experts in mass communications: most tweets are provided by non-expert communicators" This is very confusing to me as it seems to be saying two different things.

Reply: This is indeed a typo due to the heavy editing and track changes. The revised statement now reads: “Although Twitter contains messages written by journalists and other experts in mass communication, most tweets are provided by non-expert communicators”.

A more mild example, but still one that should be edited for clarity, comes on lines 179-180: "The military metaphor thanks to which we frame diseases such as cancer is a very common one to be found in public discourse." And in the next sentence, the authors use the article "the" before Time Magazine, which is not necessary.

Reply: Thank you for your collaboration, we acknowledge these typos and mistakes, and we corrected these specific cases even before sending the manuscript to the proofreader. Then, the proofreader took care of the whole document. Very much appreciated.

I am not going to go into every example of confusing writing, but I do want to recommend the authors go over the paper again and update the prose as necessary to enhance clarity. At that time, I will be prepared to accept!

Reply: Thank you.

REVIEWER 4

Thank you to the authors for drastically improving many points about the paper. I enjoyed reading this interesting research even more this time. It is much cleaner to read and understand, including contributions, backing statistical analyses, and replication on other datasets. Other reviewers also brought up great points that I did not think of.

However, as is written, the study seems to have unacknowledged limitations and concerns that render contributions too broad about understanding general discourse on Twitter that I would recommend addressing before accepting this paper.

My concern stems from ambiguity in terms of unit of analysis that biases the representativeness of the data, which causes unacknowledged impact on results and contributions. Claims cannot be made about general discourse on Twitter, but they *can* be made about a biased yet useful subset - those tweets that originate from less-frequent tweeters.

Reply: Thank you. This comment is related to our methodological choice to select one tweet per user and drop retweets, which we motivated in the previous reply to reviewers. We are here reporting our previous reply: “We constructed a corpus, which follows specific criteria that we motivated in the paper. A corpus is a balanced and representative collection of texts. As we explained, we limited one tweet per user precisely to avoid having a biased corpus where super-tweeters could have biased the percentages of frequency distribution of frames. Imagine for example that one specific Twitter user who uses Twitter many times per day, is very fond of sci-fi and monsters-related issues, he/she might use the MONSTER framing very frequently in their tweets, also about Covid-19. If we kept all the tweets by this user, this might have biased the distribution frequency of the MONSTER related words: we would have observed a high percentage of tweets with this frame, but these are all produced by the same person, who simply happens to tweet very often. This is not representative for the population. Instead, by limiting one tweet per user, we kept our corpus relatively manageable from a technical perspective (not too heavy) as well as representative and balanced from a theoretical perspective. We also dropped the retweets and mentions because these are duplicated texts, so, the same datapoint would have been counted twice. We argue that this is not desirable, from a corpus linguistics perspective. We have explicitly acknowledged these matters in the section “Constructing the corpus of Covid-19 tweets”. To conclude, our choices for the configuration of the corpus are theoretically motivated and clearly explained in the paper. We do not expect the reviewer to necessarily like them, but methodologically speaking, they are motivated and transparent. As explained later, we have added further analyses to replicate our results on an updated corpus constructed with the same criteria, collected by ourselves, and on a corpus constructed on the basis of existing resources (section “Replication studies").”

Reply: The reviewer has not commented on our reply, which provides a rational motivation for our methodological choices. In this sense, the comment by the reviewer stating that our analyses are based on “a biased yet useful subset - those tweets that originate from less-frequent tweeters” is not fully correct. Our analyses are based on a corpus (and then replicated on another corpus) of tweets. Decades of published research reporting empirical studies in corpus linguistics are based on corpora that are carefully constructed, based on principles such as representativeness and balance of the texts included therein. We therefore believe that our corpus of tweets, which we constructed based on a few theoretically motivated and clearly explained criteria, is representative for the way in which Twitter users conceptualize and talk about Covid on Twitter: we constructed a corpus that represents the language that people use on Twitter, as it is common practice in corpus linguistics and cognitive linguistics, as well as in many disciplines within Language sciences. Please note that we do not only consider “tweets that originate from less-frequent tweeters” as the reviewer mentions, but we rather consider one tweet per tweeter, thus taking into account also super tweeters, by including one of their tweets in the sample.

Generally, any study that performs social monitoring needs to:

- make clear what its unit of analysis is,

- then accordingly determine what it means to be representative of these units in its sampling frame,

- gather units to analyze and analyze those same units

- acknowledge any limitations of representativeness

- and limit contributions accordingly

Reply: We are frankly unfamiliar with the term “social monitoring”. We have found a definition for “social media monitoring”, which appears to be the active monitoring of social media channels for information about a company or organization, usually tracking of various social media content in general as a way to determine the volume and sentiment of online conversation about a brand or topic. This is not what we present in our study.

Reply: The information requested by the reviewer has been provided in the paper as follows:

- units of analysis: the first study is based on topic modelling, and adopts the methods described by the LDA algorithm; the other two studies are based on the lexical entries that belong to the various frames, as clearly stated multiple times in the manuscript. We believe that with units of analysis the reviewer may refer to the words (from the various lists of frame-related words) that we counted in the tweets. To clarify this further, our algorithm counted 1 hit whenever a lexical unit (a word from a list) was found in a tweet. Then, the numbers were added up within each frame. In this sense, our unit of analysis is the lexical units in the word lists, not the tweets. A tweet can be counted as many times as it has words related to a frame. For example a tweet such as “This wave of #covid is hard to fight” will count 1 hit for the TSUNAMI frame (wave), 1 hit for the STORM frame (wave) and 1 hit for the WAR frame (fight). We then reported in the paper the total number of occurrences of the words in the tweets, as this paragraph explains: “Analyzing all tweets from the database, a total of 10,846 tweets contained at least one term from the WAR framing, which is 5.32% of all tweets. Of these, 1,253 tweets had more than one war-related term."

- representativeness: the representativeness of these units is backed up by the sources where they have been retrieved (RelatedWords and the Frames in the Berkeley database of figurative framings).

- gather units and analyze those units: that is how we proceeded, indeed!

- Acknowledge limitations: we have acknowledged the limitation of our analysis, in particular the limitation due to the assumption we made, that the lexical units used for our analysis are used always metaphorically, throughout the corpus. This was necessary, in order to perform an automated analysis. We added a paragraph explaining this trade off at the end of the section “Determining lexical units associated with the WAR frame”.

- we toned down the claims about our contribution, in the General Discussion and Conclusion section.

At minimum in this particular study,

- it should be made clear and consistent through data collection and analyses whether this particular study is analyzing use of frames by users, or use of frames in tweets, (I'm pretty sure it's tweets, right?)

Reply: We analyse the presence of lexical entries related to specific frames in a balanced corpus of tweets that is constructed by taking into account one tweet per user, to avoid the bias of super-tweeters. In this sense, we do not monitor how Twitter works per se, but how people, in our case Twitter users, tend to talk about the pandemic on Twitter. Our study relates to principles and assumptions generally made in fields such as corpus linguistics, discourse analysis and cognitive linguistics, which may be substantially different from those made in the “social media monitoring” field, suggested by the reviewer. As we explained, in line with common practice in the disciplines with which we relate, we started by constructing a balanced and representative corpus of tweets that features as many users as possible to preserve variability within the corpus (one tweet per user), that avoids the bias introduced by super tweeters (explained above), and that provides a window onto the language used on Twitter to talk about the current epidemic.

- the limitation of a bias against super-tweeters should be made more explicit, and I would accordingly recommend explicitly walking back the study's claims, research questions, and contributions, because claims of general representativeness on Twitter are not defendable.

Specifically, if it is chosen to omit the vast majority of tweets by keeping one tweet per user and omitting retweets, claims cannot be made about how the general Twitter discourse discusses and frames COVID-19 (or to put it another way, the study's data is not representative of all tweets). Claims instead can be made about how those tweets *originating from less-frequent tweeters* discuss and frame COVID-19.

Reply: We have not “chosen to omit the vast majority of tweets by keeping one tweet per user and omitting retweets”, as the reviewer argues. Rather, we have constructed a corpus of tweets that is representative and balanced for the language people use on Twitter to talk about Covid, which is the phenomenon we want to investigate. Note that any type of sampling would indeed “omit the vast majority of tweets”. The criteria used for sampling, in our case, have been clearly explained and motivated, in line with common practices in corpus linguistics, discourse analysis, and cognitive linguistic research.

Either that, or I would recommend at least one of:

- switching to analysis at the level of a user

- more representative sample from tweeters (as one tweet per user is usually not seen as representative of a user's content, whether they are among the top tweeters or not)

Please find details below - perhaps these can make the issue more clear if it helps.

First, the study has a bias against retweeters and sharers, in favor of original tweeters, as is stated. However, many users on Twitter retweet or share someone else's tweets - this is seen as a proxy for behavior and opinion, as usually people share things they agree with.

Reply: As previously mentioned, we constructed a corpus of tweets based on the principles of representativeness and balance, in relation to the language that people use to talk about covid on Twitter. We find the reviewer’s statement “people share things they agree with” quite interesting and possibly controversial, as people may share things they strongly disagree with, by retweeting them and adding a very negative comment on them. But, as non-experts of social media monitoring, we limit our discussion to the linguistic phenomena hereby investigated.

Therefore, more accurate counts of frames might be *including* those frames that are retweeted. At the very least, one may analyze with retweets and without to see any differences. It's fine that the study disregards retweets, but this should be acknowledged as a limitation and claims about discourse should be modified accordingly.

Reply: Indeed, we have acknowledged the fact that we dropped retweets on purpose, and motivated our choice. We have modified accordingly our claims about the discourse around Covid on Twitter, explaining that our findings relate to the corpus of tweets that we constructed, and not to Twitter as a social network, which includes retweets and information propagation that can be strongly affected by super-tweeters. A new paragraph including these limitations to our study has been added to the General Discussion and Conclusion section.

And second, to paraphrase, data is being "limited to one tweet per user because of technical limitations and to make the corpus balanced and representative". I would reject the study's claim that the resulting corpus is representative - because the study is not clear about *how* it is representative. As designed, it seems like the study is confused about the unit of analysis that it is trying to represent: the tweet or the user.

Reply: As argued above, the units of analysis are the lexical entries, that is, the frame-related words, which have been carefully constructed on the basis of existing resources. We investigated how these lexical entries are used in a corpus of tweets about Covid, which has been constructed by selecting tweets that contain specific hashtags about covid, dropping retweets because they are technically repeated texts that are not original, and thus in our opinion not much representative of the variability in semantic content that can be found on Twitter, and keeping one tweet per user to construct a balanced corpus that is not biased toward the texts tweeted by super-tweeters (or even twitter bots, which are usually super tweeters).

- If the study wishes to accurately represent tweets, then by Twitter's nature there are super-tweeters, and frequencies of frames will be biased by super-tweeters who use frames more. This is an accurate representation of the Twitter discourse.

- If the study wishes to accurately represent users, then it should state this, and tweets should be aggregated or sampled representatively per user, and and all subsequent analyses should be at the level of the user, not the level of the tweet.

However, as currently designed, the study starts at the level of a user by "retaining one tweet per user", but then moves back to analyze at the level of a tweet. This results in a bias against super-tweeters with "retaining one tweet per user". Therefore, all subsequent analysis at the level of a tweet omits the vast majority of tweets that compose Twitter discourse - exactly what it seems to want to analyze.

Reply: Thank you. This study, as mentioned above, is rooted in principles and assumptions that are usually starting points in corpus linguistics and cognitive linguistics. It is physically impossible to mine the whole Twitter as it stands, because it is a dynamic archive of texts that increases on a daily basis, and therefore there are no claims that can be true for “Twitter” as a whole, technically speaking. So, we have to sample this archive, and construct a corpus that works well for our purposes. In our case we constructed a corpus that is balanced and representative for the aim of the study. Our study is to mine the linguistic phenomenon of figurative framing, that is: the use of words related to specific frames, which suggests a specific conceptualization related to the topic of Covid. Given this aim, we constructed a corpus that is suitable for this type of linguistic investigation, and that would not distort the statistic of our findings, related to the phenomena investigated.

Reply: Our choice to select one tweet per user does not go in the direction of a user-based analysis. Rather, it is done in order to have a selection of tweets that have been produced by as many different human minds as possible, rather than by a very limited number of minds (supertweeters) which may in fact very likely encompass artificial minds (twitter bots).

Reply: Regarding our methodological choice to remove duplicates (retweets), please note that even in the construction of corpora based on the texts found in the web (e.g., UK WAC, IT WAC etc) duplicate texts are typically dropped, to favor the representativeness of the texts included in the corpus. Our choice for the construction of the corpus of tweets goes in the same direction of these previous studies. In fact, there are several tools implemented precisely to remove duplicates texts for the construction of web-based corpora. Twitter as well, is a web-based resource, where creating duplicates is extremely easy (one click: retweet!), but the motivations behind this operation may be different (notably, agreement vs disagreement with the content of the original tweet) and hard to interpret in an automated manner.

- Minor point: The study should address exactly how it "retained only one tweet per user". This was asked in the review. Was the earliest tweet retained? the most recent? the one with the most framing words? Is it simple random sampling? I ask to elucidate, not to be flippant, as ambiguity does not lead to reproducibility - it leads to concern. A brief mention would suffice.

Reply: Thank you, we added the information that only the first contribution of each user was retained, meaning we have kept the first tweet on that day by users whose tweets we have not collected yet. In the previous response, we have explained that “The algorithm picked the first tweet per user that it could find, within the timeframe indicated.” To respond directly to the questions of the reviewer: Was the earliest tweet retained? Yes, the earliest on that given day. The most recent? No. The one with the most framing words? No. Is it simple random sampling? No, since we stated in the previous response that “[c]ollecting all tweets from all users, and then randomly selecting one per each user would have been computationally cumbersome and would have exceeded our database capacity.”

I can recommend four courses of action, although the authors would know which would be more appropriate for their desired contributions:

1. Keep current data collection, methods, and results that seem to analyze at the level of a tweet, and acknowledge explicit bias against super-tweeters, to favor those users who tweet less often. This seems to require revisions to contributions and research questions, as the overall Twitter discourse is not being studied for framing, but the Twitter discourse being studied is biased towards discourse by less-frequent users. Is there a theoretical motivation for studying in this way?

2. Keep current data collection, analyze at the level of a user, and acknowledge explicit bias against super-tweeters, to favor those users who tweet less often. This would require more extensive revisions to analyses, but would result in a different contribution: instead of studying the entirety of framing discourse on Twitter, studying how *users*, with emphasis on less-frequent users, frame the discussion.

3. Decreasing bias against super-tweeters by gathering a representative amount of tweets per user, and analyze at the level of a user. This would result in a proper representative analysis of discourse on Twitter at the level of the user, and contributions on how users discuss and frame would follow.

4. Decreasing bias against super-tweeters by gathering a representative amount of tweets per user, and analyze at the level of a tweet. This would result in a proper representative analysis of discourse on Twitter at the level of the tweet, and defendable contributions on how tweets generally discuss and frame would follow.

Reply: Thank you, we adopted the first course of action and acknowledged the limitations of our study accordingly, emphasizing our background in cognitive and corpus linguistics and where our methodological choices stem from.

---

## [Decision Letter · Decision Letter 2]

8 Sep 2020

PONE-D-20-10986R2

Framing COVID-19 How we conceptualize and discuss the pandemic on Twitter

PLOS ONE

Dear Dr. Wicke,

Thank you for submitting your manuscript to PLOS ONE. After careful consideration, we feel that it has merit but does not fully meet PLOS ONE’s publication criteria as it currently stands. Therefore, we invite you to submit a revised version of the manuscript that addresses the points raised during the review process.

I am happy to accept your paper pending a few minor modifications as outlined by reviewer 1 (see below). Please submit the final revision with a cover letter showing how you have incorporated these final minor revisions as per the instructions below. I will then process the paper for publication without another round of review. Thank you for your diligent effort to address all of the reviewers' helpful comments throughout. I look forward to receiving the final version of your paper in due course.

We look forward to receiving your revised manuscript.

Kind regards,

Panos Athanasopoulos, Ph.D

Academic Editor

PLOS ONE

Reviewers' comments:

Reviewer's Responses to Questions

**Comments to the Author**

1. If the authors have adequately addressed your comments raised in a previous round of review and you feel that this manuscript is now acceptable for publication, you may indicate that here to bypass the “Comments to the Author” section, enter your conflict of interest statement in the “Confidential to Editor” section, and submit your "Accept" recommendation.

Reviewer #1: All comments have been addressed

Reviewer #2: All comments have been addressed

2. Is the manuscript technically sound, and do the data support the conclusions?

Reviewer #1: Yes

Reviewer #2: Yes

3. Has the statistical analysis been performed appropriately and rigorously? 

Reviewer #1: Yes

Reviewer #2: Yes

4. Have the authors made all data underlying the findings in their manuscript fully available?

Reviewer #1: Yes

Reviewer #2: Yes

5. Is the manuscript presented in an intelligible fashion and written in standard English?

Reviewer #1: Yes

Reviewer #2: Yes

6. Review Comments to the Author

Reviewer #1: The manuscript, which explores metaphors used to frame COVID-19 on Twitter, has been revised to address concerns raised in review. Overall, I think the revision is very successful. The argument is much clearer, and I think the paper is nearly ready for publication. I have a few final comments that should be straightforward to address. I am recommending that the paper be accepted.

1. I still think the use of “empirical basis” is misleading and confusing (p. 17) because, in many research contexts, this term is used to refer to specific criteria or quantitative methods of comparison. Since the current approach is more informal, why not say in the paper what is said in the response letter? Namely, something like, “These two numbers of clusters chosen by looking at the data and are backed up by post hoc testing.”

2. Please include some details about the criteria that Lamsal used to create their database of COVID-19-related Tweets.

3. I was a little confused by the term “individual” in “individual tweeters” on page 13. I think “unique tweeters” would be clearer, assuming that is the intended meaning of “individual tweeters.”

4. Using the Topic labels (e.g., “Communications and Reporting”), rather than the numeric codes (e.g., “Topic III”), would make the figures more readable.

Once again, I found this research really interesting and think that it will make a nice contribution to the literature. I appreciate all the careful work the authors have done throughout the review process.

Reviewer #2: I am satisfied the authors have addressed all critical reviewer points. The current manuscript is significantly stronger than the initial submission and makes a nice contribution to the literature.

7. PLOS authors have the option to publish the peer review history of their article (what does this mean?). If published, this will include your full peer review and any attached files.

Reviewer #1: No

Reviewer #2: No

---

## [Author Response · Author response to Decision Letter 2]

8 Sep 2020

REVIEWER 1

The manuscript, which explores metaphors used to frame COVID-19 on Twitter, has been revised to address concerns raised in review. Overall, I think the revision is very successful. The argument is much clearer, and I think the paper is nearly ready for publication. I have a few final comments that should be straightforward to address. I am recommending that the paper be accepted.

Reply: Thank you. We appreciate the constructive feedback throughout the review process and such successful revision would not have been possible without such useful and detailed reviews.

1. I still think the use of “empirical basis” is misleading and confusing (p. 17) because, in many research contexts, this term is used to refer to specific criteria or quantitative methods of comparison. Since the current approach is more informal, why not say in the paper what is said in the response letter? Namely, something like, “These two numbers of clusters chosen by looking at the data and are backed up by post hoc testing.”

Reply: Thank you. We have now changed the use of “empirical basis” and refer to the selection of clusters as mentioned by the reviewer.

2. Please include some details about the criteria that Lamsal used to create their database of COVID-19-related Tweets.

Reply: Thank you. We have now included the following criteria in the paper: Lamsal’s dataset is a constantly updated repository of twitter IDs. The collection of those IDs is based on English tweets that include 90+ hashtags and keywords that are commonly used while referencing the pandemic.

3. I was a little confused by the term “individual” in “individual tweeters” on page 13. I think “unique tweeters” would be clearer, assuming that is the intended meaning of “individual tweeters.”

Reply: Thank you. We have now changed the mention of “individual tweeters” with “unique tweeters”.

4. Using the Topic labels (e.g., “Communications and Reporting”), rather than the numeric codes (e.g., “Topic III”), would make the figures more readable.

Reply: Thank you. We have now updated the figure with the topic labels for the 4 topics as requested by the reviewer.

Once again, I found this research really interesting and think that it will make a nice contribution to the literature. I appreciate all the careful work the authors have done throughout the review process.

Reply: Thank you, we appreciate this comment and all the careful responses from the reviewer.

REVIEWER 2

I am satisfied the authors have addressed all critical reviewer points. The current manuscript is significantly stronger than the initial submission and makes a nice contribution to the literature.

Reply: Thank you. We appreciate the constructive feedback throughout the review process and such successful revision would not have been possible without such useful and detailed reviews.

---

## [Editor Report · Decision Letter 3]

18 Sep 2020

Framing COVID-19 How we conceptualize and discuss the pandemic on Twitter

PONE-D-20-10986R3

Dear Dr. Wicke,

We’re pleased to inform you that your manuscript has been judged scientifically suitable for publication and will be formally accepted for publication once it meets all outstanding technical requirements.

Kind regards,

Panos Athanasopoulos, Ph.D

Academic Editor

PLOS ONE
---

## [Editor Report · Acceptance letter]

22 Sep 2020

PONE-D-20-10986R3

Framing COVID-19:
How we conceptualize and discuss the pandemic on Twitter

Dear Dr. Wicke:

I'm pleased to inform you that your manuscript has been deemed suitable for publication in PLOS ONE. Congratulations! Your manuscript is now with our production department.

Kind regards,

on behalf of

Professor Panos Athanasopoulos 

Academic Editor

PLOS ONE